# Probing plant signal processing optogenetically by two channelrhodopsins

Meiqi Ding[1,5], Yang Zhou[2,4,5], Dirk Becker[1,5], Shang Yang[2,5], Markus Krischke[3], Sönke Scherzer[1], Jing Yu-Strzelczyk[2], Martin J. Mueller[3], Rainer Hedrich[1✉], Georg Nagel[2✉], Shiqiang Gao[2✉] & Kai R. Konrad[1✉]

Early plant responses to different stress situations often encompass cytosolic $Ca^{2+}$ increases, plasma membrane depolarization and the generation of reactive oxygen species[1–3]. However, the mechanisms by which these signalling elements are translated into defined physiological outcomes are poorly understood. Here, to study the basis for encoding of specificity in plant signal processing, we used light-gated ion channels (channelrhodopsins). We developed a genetically engineered channelrhodopsin variant called XXM 2.0 with high $Ca^{2+}$ conductance that enabled triggering cytosolic $Ca^{2+}$ elevations in planta. Plant responses to light-induced $Ca^{2+}$ influx through XXM 2.0 were studied side by side with effects caused by an anion efflux through the light-gated anion channelrhodopsin ACR1 2.0[4]. Although both tools triggered membrane depolarizations, their activation led to distinct plant stress responses: XXM 2.0-induced $Ca^{2+}$ signals stimulated production of reactive oxygen species and defence mechanisms; ACR1 2.0-mediated anion efflux triggered drought stress responses. Our findings imply that discrete $Ca^{2+}$ signals and anion efflux serve as triggers for specific metabolic and transcriptional reprogramming enabling plants to adapt to particular stress situations. Our optogenetics approach unveiled that within plant leaves, distinct physiological responses are triggered by specific ion fluxes, which are accompanied by similar electrical signals.

When threatened, sessile plants respond rapidly with metabolic switches and genetic reprogramming to environmental cues[5–7]. Electrical, cytosolic $Ca^{2+}$ concentration ($[Ca^{2+}]_{cyt}$) and reactive oxygen species (ROS) signals are among the earliest plant reactions observed with stressors as diverse as wounding, pathogen attack, water and salt stress, and are suggested to be intertwined[1–3]. After pathogen attack or wounding, jasmonic acid (JA) is rapidly synthesized, playing a crucial role in balancing plant growth and defence. In this trade-off scenario, JA and salicylic acid (SA) act antagonistically in the control of immunity and programmed cell death (PCD)[8,9]. A ROS burst precedes PCD playing a crucial role in pathogen defence[10]. Conversely, abscisic acid (ABA) governs drought and salt stress by regulating the plant water status when turgor pressure declines. This simplified view of hormonal control is in fact much more complicated. Mutual control of the phytohormones and second messengers involved is challenging to dissect. This optogenetics study aimed to investigate the role of $[Ca^{2+}]_{cyt}$- and anion-efflux-induced electrical signals in encoding specificity in plant processes by individually triggering them by means of light-gated ion channels.

Microbial rhodopsins are light-sensitive proteins and powerful tools for minimally invasive manipulation of cells by light (optogenetics)[11,12]. These optogenetic tools have recently been made available for use in plants[4,13]. ACR1 2.0 triggers defined membrane depolarizations by anion efflux and guides pollen tubes when stimulated locally[4] or initiates stomatal closure through depolarization-synchronized anion and cation efflux[14]. So far, the role of $[Ca^{2+}]_{cyt}$ signals in plants has been studied with loss-of-function approaches of $Ca^{2+}$-signalling elements or $Ca^{2+}$-permeable channels[15,16]. Here we established an approach equivalent to a gain-of-function strategy, a light-gated channel with on–off features allowing defined $[Ca^{2+}]_{cyt}$ modifications.

## A light-gated $Ca^{2+}$-permeable channel

Channelrhodopsins are light-gated cation channels with a broad selectivity for cations[17,18]. Previously, we engineered a channelrhodopsin variant with extra expression and medium long open time (XXM)[19] and pronounced $Ca^{2+}$ permeability[20]. Here we screened a set of XXM mutants and identified an extra H134Q substitution in XXM (XXM 1.1) that augments photocurrents and $Ca^{2+}$ conductance (Fig. 1a–c and Extended Data Fig. 1a–d). Endoplasmic reticulum-export and plasma membrane-targeting signal peptides in XXM 1.2 and XXM 1.3 further improved membrane targeting and photocurrents in *Xenopus laevis* oocytes, and an N-terminal 11 amino acid truncation in XXM 1.3 led to the final XXM 2.0 version (Fig. 1c and Extended Data Fig. 1a,e).

XXM 2.0 was cloned into a plant expression vector providing for in planta production of all-*trans* retinal, the essential chromophore of

[1]Molecular Plant Physiology and Biophysics, Julius-von-Sachs-Institute, University of Wuerzburg, Würzburg, Germany. [2]Department of Neurophysiology, Physiological Institute, University of Wuerzburg, Würzburg, Germany. [3]Pharmaceutical Biology, Julius-von-Sachs-Institute, University of Wuerzburg, Würzburg, Germany. [4]Present address: School of Life Sciences, Zhengzhou University, Zhengzhou, China. [5]These authors contributed equally: Meiqi Ding, Yang Zhou, Dirk Becker, Shang Yang. ✉e-mail: rainer.hedrich@uni-wuerzburg.de; nagel@uni-wuerzburg.de; gao.shiqiang@uni-wuerzburg.de; kai.konrad@uni-wuerzburg.de

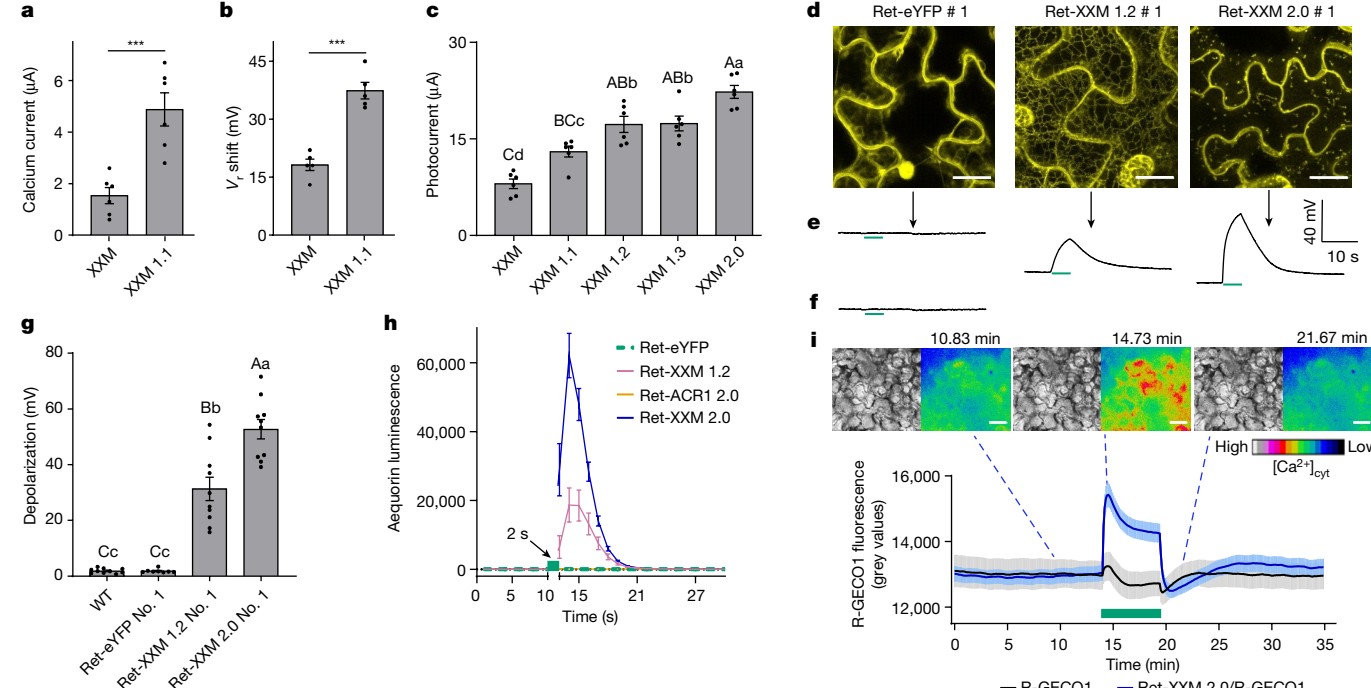

**Fig. 1 | Functional characterization of XXM 2.0, a channelrhodopsin variant with enhanced Ca²⁺ conductance. a,b,** Blue-light (473 nm, 3 mW mm⁻²)-activated calcium current (**a**) and reversal potential ($V_r$) shift (**b**) of *X. laevis* oocytes expressing XXM and XXM 1.1. Error bars show s.e.m., $n$ = 6 (**a**) and 5 (**b**) cells of 2 oocyte batches. Significance was determined by two-sided Student's $t$-test. ***$P \le 0.001$. **c,** Blue-light-induced photocurrents of XXM variants in *X. laevis* oocytes. Error bars show s.e.m., $n$ = 6 cells of 2 oocyte batches. Significance was determined by one-way analysis of variance (ANOVA) followed by a Tukey post hoc test. Different letters indicate significant differences among samples (capital letters: $P \le 0.01$ and lowercase letters: $P \le 0.05$). **d–f,** Confocal images of leaf epidermis (**d**), and mesophyll cell membrane voltage of transgenic Ret-eYFP #1, Ret-XXM 1.2 #1 and Ret-XXM 2.0 #1 (**e**) and WT (**f**) *N. tabacum* leaves. Scale bars, 20 μm (**d**). $n$ = 6 leaves of 2 batches of *N. tabacum* plants. Green bars indicate green light application (532 nm, 180 μW mm⁻²). **g,** Membrane potential changes of WT or transgenic *N. tabacum* leaves during green light irradiation (532 nm, 180 μW mm⁻²). Error bars show s.e.m., $n$ = 10, 8, 10 and 10 leaves from 2 batches of *N. tabacum* plants. One-way ANOVA followed by a Dunnett T3 post hoc test was used to determine significance. **h,** Aequorin-luminescence recordings in *N. benthamiana* leaves following green light (520 nm, 50 μW mm⁻²) illumination. Error bars show s.e.m., $n$ = 9, 7, 8 and 10 leaves from 2 batches of *N. benthamiana* plants. **i,** R-GECO1-based [Ca²⁺]$_{cyt}$ changes in *N. benthamiana* mesophyll cells transiently expressing the denoted constructs following local green light (532 nm, 180 μW mm⁻²) illumination. Error bars show s.e.m., $n$ = 16, 25 leaves from 5 batches of *N. benthamiana* plants. Scale bars, 50 μm.

rhodopsin[4] (Extended Data Fig. 1a). Retinal synthesis did not affect carotenoid content or plant growth under non-stimulating red light conditions when compared to wild-type (WT) plants (Extended Data Fig. 2a–f). The broad XXM 2.0 action spectrum (Extended Data Fig. 1f) permits green light stimulation, minimizing interference with plant photoreceptor signalling[21,22]. Compared to Ret-XXM 1.2 or 1.3, Ret-XXM 2.0 exhibited improved plasma membrane targeting and stronger membrane depolarizations when stimulated with green light (Fig. 1d–g and Extended Data Fig. 2g–i). This probably resulted from pronounced cation influx and Ca²⁺-dependent alteration of endogenous ion transport[23]. No notable light response was induced in WT or control plants expressing the retinal-producing enzyme and soluble enhanced yellow fluorescent protein (Ret-eYFP; Fig. 1e–g and Extended Data Fig. 1a).

In contrast to Ret-eYFP or Ret-ACR1 2.0, Ret-XXM 2.0 elicited a substantial increase in [Ca²⁺]$_{cyt}$ in *Nicotiana benthamiana* leaves expressing the aequorin Ca²⁺ sensor[24] (Fig. 1h). Using the red fluorescent Ca²⁺ sensor R-GECO1[25] and the pH reporter pHuji[26], we observed sustained [Ca²⁺]$_{cyt}$ elevations but only minor, short-lived pH deflections with Ret-XXM 2.0 at cellular resolution (Fig. 1i, Extended Data Fig. 3a,b and Supplementary Video 1). Simultaneous electrical recordings and [Ca²⁺]$_{cyt}$ imaging revealed that light-induced Ret-XXM 2.0-dependent [Ca²⁺]$_{cyt}$ signals were accompanied by reproducible membrane depolarizations and both could be fine-tuned by light intensity or duration (Extended Data Fig. 3c–i). Ret-ACR1 2.0 triggered membrane depolarizations too, but no sustained [Ca²⁺]$_{cyt}$ increases (Extended Data Fig. 4a). For physiological investigations we finally developed a global light-application protocol

(520 nm, 9 μW mm⁻²) to induce membrane depolarizations or Ca²⁺ influx in plant leaf cells stably expressing Ret-ACR1 2.0 or Ret-XXM 2.0 (hereafter referred to as ACR1 and XXM; Fig. 2a,b and Extended Data Fig. 4b).

## XXM and ACR1 elicit distinct phenotypes

When exposed to continuous global low-intensity green light illumination, XXM plants developed necrotic spots after 24 h (Fig. 2c and Extended Data Fig. 5a,b). In support of a pathogen-associated PCD[27] response governed by Ca²⁺ signalling, the necrotic phenotype was suppressed by chelating extracellular Ca²⁺ (Extended Data Fig. 5c,d). A nonlinear increase in apoplastic conductivity further indicated that PCD develops about 4–8 h after XXM activation (Fig. 2d). By contrast, ACR1 triggered steady anion release from mesophyll cells, resulting in a linear increase in ion leakage towards the apoplast and plant wilting after 4 h continuous low-intensity green light illumination (Fig. 2c,d and Extended Data Figs. 5e–j and 6a). The rapid recovery of leaf turgor within 20 min after ACR1 shut-off (Supplementary Video 2) supports the idea that the wilting phenotype results from reversible leaf cell water loss initiated by ACR1-driven anion efflux and concomitant potassium release through depolarization-activated K⁺ channels[14].

Watering plants with 35% polyethylene glycol 6000 (PEG) mimicked the wilting time course observed in ACR1 plants (Extended Data Fig. 5e–j) and provided suitable experimental conditions for analysing ACR1 responses in a physiological context. Similarly, infection with

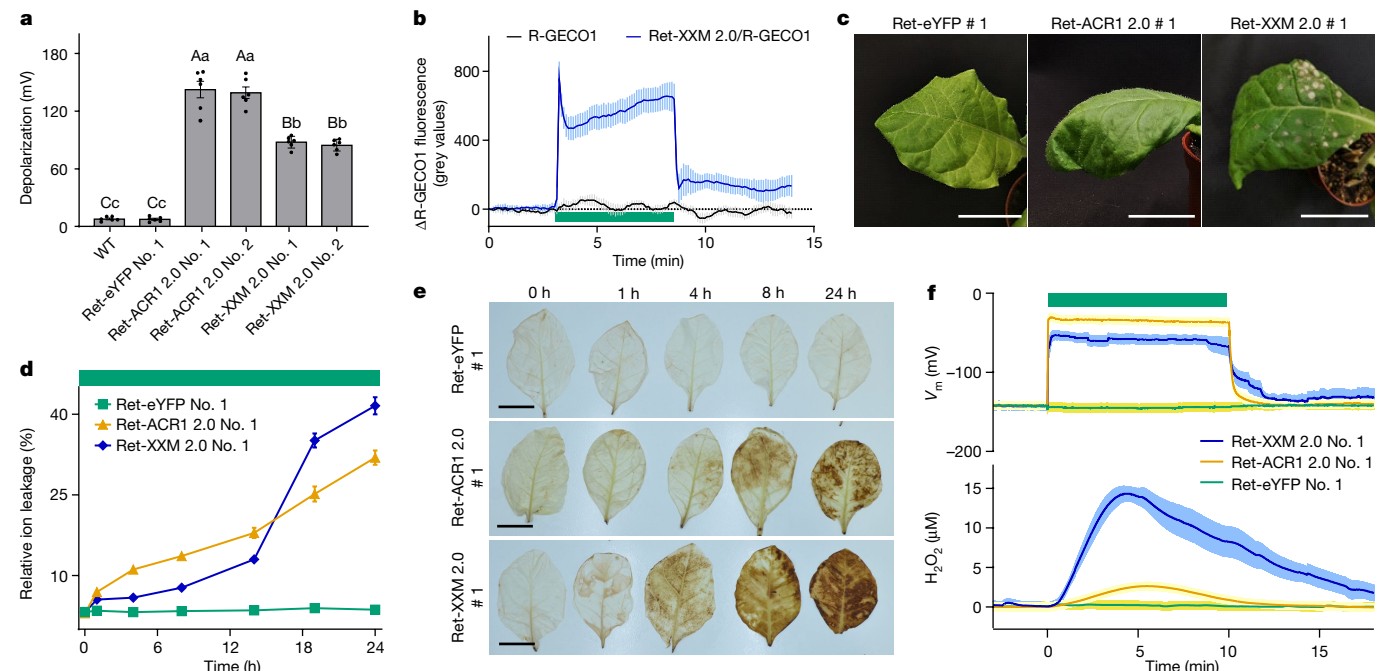

**Fig. 2 | Distinct plant stress responses induced by ACR1 2.0 and XXM 2.0 stimulation. a**, Mesophyll cell depolarization induced by 60 s global green light illumination (520 nm, 9 μW mm$^{-2}$) in WT, a transgenic Ret-eYFP line or in two Ret-ACR1 2.0 and two Ret-XXM 2.0 transgene plant lines. Error bars show s.e.m., $n$ = 6 leaves of 2 batches of *N. tabacum* plant. One-way ANOVA followed by a Dunnett T3 post hoc test was used to determine significance. **b**, Mean R-GECO1 fluorescence change in transgenic *N. tabacum* mesophyll cells following global green light illumination (green bar). Error bars show s.e.m., $n$ = 8 and 7 leaves from 2 batches of *N. tabacum* plants. **c**, Phenotypes of transgenic *N. tabacum* leaves after 24 h global green light treatment. Scale bars, 5 cm. $n$ = 6 leaves of 2 batches of *N. tabacum* plants. **d**, Relative ion leakage from leaf tissue at different time points following global green light treatment. Error bars show s.e.m., $n$ = 6 leaves from 2 batches of *N. tabacum* plants. **e**, ROS detection in *N. tabacum* leaves by diaminobenzidine staining. *N. tabacum* leaves were collected at indicated time points after global green light illumination. Scale bar, 5 cm. $n$ = 5 leaves from 2 batches of *N. tabacum* plants. **f**, Simultaneous amperometric quantification of hydrogen peroxide (H$_2$O$_2$) dynamics and membrane potential ($V_m$) in transgenic *N. tabacum* leaves following global green light illumination. Error bars show s.e.m., $n$ = 7, 8 and 10 leaves from 2 batches of *N. tabacum* plants.

*Pseudomonas syringae* pv. *tomato* strain DC3000 (*Pst*) was utilized as a suitable physiological control, reproducing the phenotypic characteristics observed with XXM (Extended Data Fig. 5a,b).

*Nicotiana tabacum* leaves inoculated with *Pst* triggered robust [Ca$^{2+}$]$_{cyt}$ increases (Extended Data Fig. 6b,c). As Ca$^{2+}$-dependent ROS production is key for plant immunity and abiotic stress signalling[10,28,29], we monitored ROS production. We observed strong and progressive ROS generation in leaves 4 h after XXM stimulation as well as *Pst* inoculation. ROS production, however, was significantly lower in leaves of ACR1 or PEG-treated plants compared to XXM leaves and was barely detectable in control leaves (Fig. 2e and Extended Data Fig. 6d–g). In vivo quantitative real-time amperometric H$_2$O$_2$ measurements revealed a rapid H$_2$O$_2$ transient, reaching peak values of 15 μM within 4 min following light stimulation in XXM leaves, that levelled down to sustained steady-state values of 4–5 μM and returned to control levels upon light off (Fig. 2f and Extended Data Fig. 6f). The H$_2$O$_2$ signal lagging about 1 min behind the XXM-induced [Ca$^{2+}$]$_{cyt}$ increase and depolarization (Extended Data Fig. 6h–j) supports voltage- or Ca$^{2+}$-dependent ROS production. Despite comparable amplitudes of electrical signals triggered by both optotools, ACR1 provoked only a small ROS rise of about 1 μM after 1 h stimulation (Extended Data Fig. 6f). Thus, our data support Ca$^{2+}$-dependent profound H$_2$O$_2$ production in plants[30].

## Metabolic rearrangement by XXM and ACR1

The observed phenotypes indicated that XXM and ACR1 address signalling pathways related to immune responses and osmotic stress, respectively. The phytohormone ABA is central to drought adaptation whereas the interplay of JA isoleucine (JA-Ile) and SA orchestrates defence signalling pathways[29,31]. In general, JA is rapidly synthesized through its intermediate *cis*-(+)-12-oxo phytodienoic acid (OPDA) and is subsequently conjugated with L-isoleucine to its active variant JA-Ile[30,32]. In this context, OPDA is discussed as a signalling molecule that regulates diverse biological processes in a JA-independent manner[33,34].

We used a targeted metabolomic approach to quantify the aforementioned marker metabolites. After 1 h, 4 h and 24 h green light treatment, ABA and proline levels increased strongly in ACR1 and PEG-treated control plants, whereas no significant changes were detected in XXM stimulated plants and *Pst*-treated plants or red light control plants (Fig. 3a–d and Extended Data Fig. 7a–c). OPDA, JA, JA-Ile and SA remained at low levels in PEG-treated plants or ACR1 stimulated plants and remained at basal levels under control conditions (Fig. 3e–l and Extended Data Fig. 7d–o). By contrast, in XXM plants JA-Ile, JA and OPDA strongly accumulated already after 1 h, returning to control levels within 4 h. In comparison, the onset of the SA transient was delayed and returned to basal levels after about 24 h (Fig. 3f–h,j–l and Extended Data Fig. 7i–k,m–o). Following *Pst* leaf inoculation, levels of JA-Ile, JA and OPDA were comparable to those of control plants, whereas SA levels increased significantly but less than with XXM stimulation (Fig. 3e,i–l and Extended Data Fig. 7h,l). These data corroborate the phenotypes as well as the kinetics of ROS generation and ion leakage (Extended Data Figs. 5a,b and 6a,d,e). In conclusion, spray inoculation and optotool activation seem to act on different timescales. The slow, successive *Pst* infection process[35,36] contrasts with the immediate impact of XXM stimulation, which affects all leaf cells simultaneously. This difference is evident in the unchanged SA levels observed 2 h after *Pst* infection, compared to the tenfold increase in SA levels observed at the same time point following XXM stimulation (Fig. 3i versus Fig. 3j inset). The minor transient increase of JA-Ile and JA observed after control spray

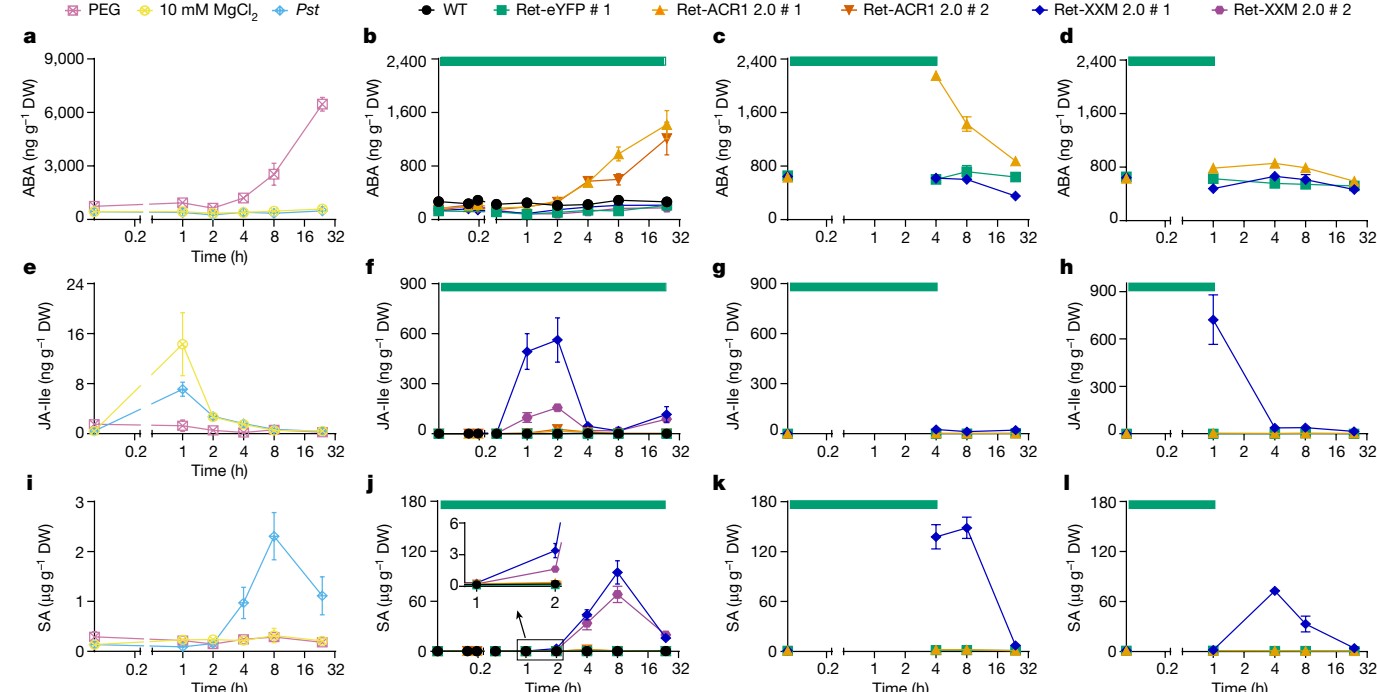

**Fig. 3 | XXM 2.0 and ACR1 2.0 activation trigger distinct metabolite and hormone patterns. a**, Quantification of ABA in Ret-eYFP #1 transgenic *N. tabacum* plants at indicated time points after watering with 35% PEG, spray inoculation with *Pst* or 10 mM MgCl₂ as control. Error bars show s.e.m., *n* = 5 and 6 leaves from 2 batches of *N. tabacum* plants. **b**–**d**, ABA content in WT or transgenic *N. tabacum* plants at indicated time points upon constant (**b**), 4 h (**c**) or 1 h (**d**) global green light illumination (520 nm, 9 μW mm⁻²). Error bars show s.e.m., *n* = 4, 5, 6 and 7 leaves from 2 batches of *N. tabacum* plants. **e**, JA-Ile content in Ret-eYFP #1 transgenic *N. tabacum* plants following PEG, *Pst* or MgCl₂ treatment. Error bars show s.e.m., *n* = 4, 5 and 6 leaves from 2 batches of *N. tabacum* plants. **f**–**h**, JA-Ile content in WT or transgenic *N. tabacum* plants at

different time points in response to constant (**f**), 4 h (**g**) or 1 h (**h**) global green light illumination. Error bars show s.e.m., *n* = 4, 5, 6 and 7 leaves from 2 batches of *N. tabacum* plants. **i**, SA content in Ret-eYFP #1 transgenic *N. tabacum* plants following PEG, *Pst* or MgCl₂ treatment. Error bars show s.e.m., *n* = 4, 5 and 6 leaves from 2 batches of *N. tabacum* plants. **j**–**l**, SA content in WT or transgenic *N. tabacum* plants at indicated time points in response to constant (**j**), 4 h (**k**) or 1 h (**l**) global green light illumination, with the inset in **j** showing a magnified view of the indicated time points. Error bars show s.e.m., *n* = 4, 5, 6 and 7 leaves from 2 batches of *N. tabacum* plants. The exact numbers of samples in **a**–**l** are listed in Supplementary Table 1.

inoculation probably resulted from mechanical cues[37]. However, the possibility of a rapid, transient increase in JA-Ile and its precursors being missed during *Pst* treatment cannot be excluded. Overall, our results probably indicate that the two different optogenetic tools addressed distinct metabolic pathways in leaves.

## XXM and ACR1 control transcript profiles

To substantiate the notion that the optogenetic tools specifically address pathogen and wounding responses with XXM and drought signalling with ACR1, we complemented our metabolite profiling with transcriptomics. For this purpose, we subjected ACR1 and XXM plants to 1 h or 4 h low-intensity green light episodes and followed transcriptome profiles along a time course of 1, 4 and 8 h (Extended Data Fig. 8a).

Activation of either ACR1 or XXM evoked significant transcriptional reprogramming: more than 1,200 (1 h) and 2,300 (4 h) differentially expressed genes (DEGs) in ACR1 plants, and about 11,000 (1 h) and 13,000 (4 h) DEGs in XXM plants, respectively (Fig. 4a,b and Supplementary Table 2, tabs 1–5). The number of DEGs rapidly declined after the light stimulus ceased, resembling optogenetic on–off switching (Fig. 4a,b). In contrast to the optotools, and in line with a steady stress response manifestation during water stress[38] or pathogen attack[35], DEGs exhibited a gradual increase over time following physiological stress (Fig. 4a–d and Supplementary Table 2, tabs 1–7). About 20% (1 h) and 50% (4 h) of the 1,333 DEGs addressed by drought stress (8 h PEG treated) were shared with ACR1, respectively. Similarly, 73% and 77% of 1,638 DEGs addressed by biotic stress (8 h *Pst* inoculation) were covered by the XXM dataset (Extended Data Fig. 8b,c).

Gene ontology (GO)-based annotation of DEGs (plant GOSlim) revealed that abiotic (PEG8h) and biotic (*Pst*8h) stress responses matched best to those obtained with the 1 h stimulated ACR1 and XXM plants, respectively (Extended Data Fig. 8d–g and Supplementary Table 2, tab 8). Accordingly, at the level of gene function, the upregulated GO terms related to ABA signalling and water transport were enriched following ACR1 activation and 8 h PEG treatment. Conversely, biological processes related to transport and signalling of auxin were downregulated in ACR1 plants (Extended Data Fig. 8f,h,i and Supplementary Table 2, tabs 9–14). These results indicate that after activation, ACR1 plants face water deprivation and respond correspondingly by activating ABA signalling and biosynthesis and shutting down auxin-controlled plant growth (Extended Data Fig. 8h and Supplementary Table 2, tabs 9–12).

Biological processes associated with hormonal responses, cell recognition, immune responses and wounding were upregulated in XXM and *Pst*-inoculated plants, whereas GO terms related to photosynthesis were downregulated (Extended Data Fig. 8g,j,k and Supplementary Table 2, tabs 15–20). Chlorophyll fluorescence measurements corroborated a rapid shutdown of photosynthesis in light-stimulated XXM plants and to a lesser extend in *Pst*-inoculated leaves (Extended Data Fig. 9a). Compared to that for *Pst* inoculation, the six to eight times higher number of XXM-addressed DEGs highlights the numerous genes specifically targeted by XXM. Functional classification (plant GOSlim) of these genes again identified signal transduction, stress responses and cell death as enriched upregulated biological processes whereas photosynthesis as well as growth and development were downregulated (Supplementary Table 3). The observation that many upregulated

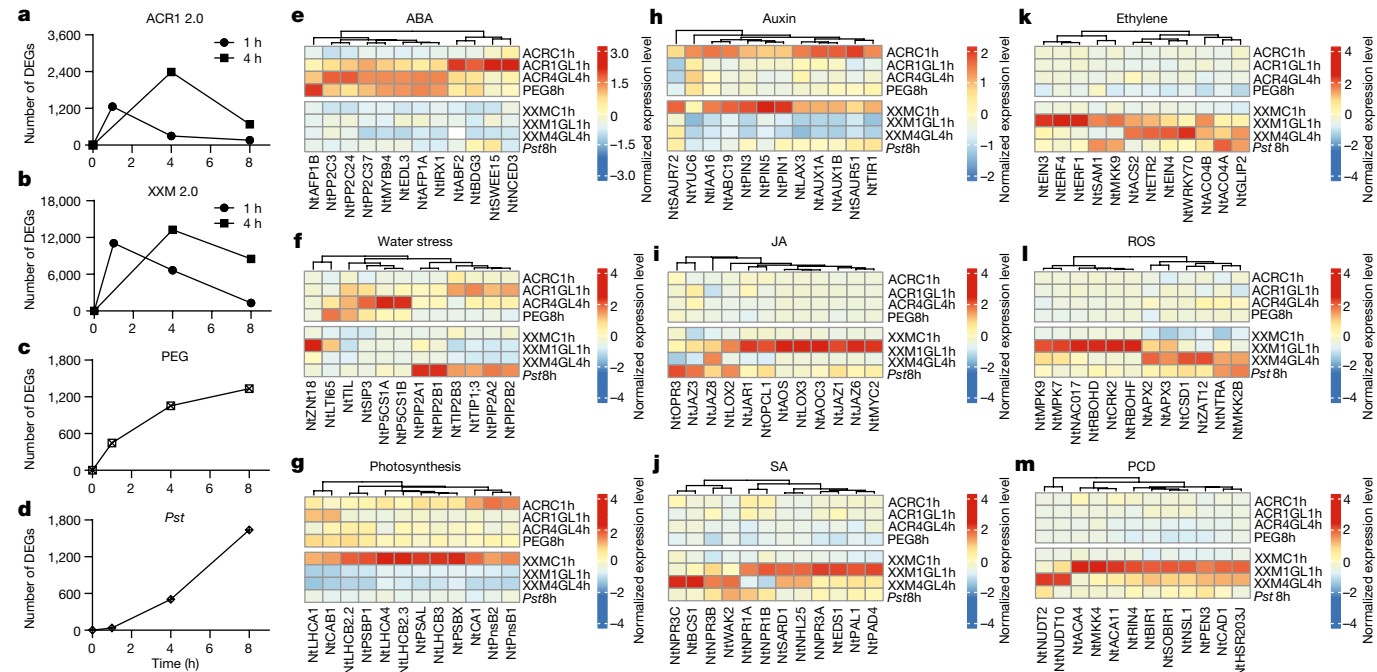

**Fig. 4 | Rapid, reversible and divergent transcriptional reprogramming by optogenetic activation of ACR1 2.0 and XXM 2.0. a–d**, Time course of DEGs at 1, 4 and 8 h after 1 h or 4 h of green light stimulation (520 nm, 9 µW mm$^{-2}$) in Ret-ACR1 2.0 (**a**) and Ret-XXM 2.0 (**b**) transgenic plants, or PEG-watered (**c**) and *Pst*-sprayed (**d**) control plants. **e–m**, Heat maps showing representative transcripts upregulated (red) or downregulated (blue) when ACR1 2.0 plants (upper heat map) or XXM 2.0 plants (lower heat map) were stimulated with green light. ACR1 2.0 and XXM 2.0 plants not stimulated with green light, at the 1 h time point (ACRC1h or XXMC1h), PEG8h or *Pst*8h served as biological controls. Representative DEGs associated with responses or pathways to ABA (**e**), water stress (**f**), photosynthesis (**g**), auxin (**h**), JA (**i**), SA (**j**), ethylene (**k**), ROS (**l**) and PCD (**m**) are shown. The expression levels of the DEGs are represented by *z*-score-normalized colour scales from blue (low expression) to red (high expression).

XXM DEGs are involved in Ca$^{2+}$ signalling (Supplementary Table 4), suggests that—compared to receptor-based *Pst* recognition—the Ca$^{2+}$-permeable optotool activates a 'global' Ca$^{2+}$ response, addressing several Ca$^{2+}$-dependent pathways.

ACR1 and XXM shared common DEGs and GO terms (Extended Data Fig. 9b–e). However, expression profiles of top regulated genes addressed by either ACR1 or XXM activation showed that genes for proline synthesis, water transport, ABA synthesis and signalling were regulated by ACR1 (Fig. 4e,f and Supplementary Table 5). Genes associated with photosynthesis and auxin-dependent growth regulation were repressed with both optotools (Fig. 4g,h). Many genes related to JA, SA, ethylene, PCD and ROS pathways showed specific regulation in XXM plants but remained largely unchanged in ACR1 (Fig. 4i–m). These involved transcriptional master regulators of JA signalling and biosynthesis (Fig. 4i) that fine-tune plant immunity[31]. XXM-induced DEGs also included SA receptors and hub regulators (Fig. 4j) essential for pathogen resistance[39]. Furthermore, XXM activation strongly triggered ethylene-associated plant immune responses[40] (Fig. 4k). The overall picture that XXM, but not ACR1, activates immune responses is also reflected by specific upregulation of genes associated with ROS production and redox regulation or PCD (Fig. 4l,m).

## Tuning plant responses optogenetically

Green light has minor effects on photosynthesis and photomorphogenesis compared to red and blue light[41–43]. It addressed few DEGs in mature *N. tabacum* leaves (Supplementary Table 6), making it a minimally invasive condition for rhodopsin-based optogenetic plant manipulation.

We introduced XXM as a new tool to orchestrate Ca$^{2+}$ signals and downstream signalling pathways in planta by applying tailored light protocols (Figs. 1 and 2 and Extended Data Fig. 3). Of note, XXM additionally conducts Na$^+$ in animal cell systems[20]. In contrast to that in animal cells, the extracellular Na$^+$ concentration in non-salt-stressed plants is about 1 mM (refs. 44,45), not exceeding 40 mM in the cytosol[46]. Given a resting potential in plant leaf cells of about −120 to −160 mV, a small inward-directed Na$^+$ movement through XXM is outcompeted by Ca$^{2+}$, driven by its steep inward gradient (>10,000-fold). Experimental evidence showed that an XXM-mediated necrotic phenotype occurred in Ca$^{2+}$- but not Na$^+$-rich media (Extended Data Fig. 5d). Further, typical sodium stress markers were barely induced by XXM, whereas many more genes related to Ca$^{2+}$ transport and signalling were upregulated (Supplementary Table 4). Thus, in the plant context, the Na$^+$ conductance of XXM can be disregarded, making XXM ideal to study the role of Ca$^{2+}$ signals.

Using XXM and ACR1 side by side, we were able to tackle a long-standing question of how membrane potential changes and/or [Ca$^{2+}$]$_{cyt}$ variations shape plant stress responses. XXM and ACR1 both trigger comparable membrane depolarizations but led to clearly separable plant stress responses (Figs. 1–4). Our study allows the following interpretation: a long-term voltage change may not represent a signalling cue per se to convey specificity for the drought or immune response observed here. Instead, an ACR1-mediated light-induced hydraulic signal, resulting from anion efflux, accompanied by potassium and water efflux, initiated the drought stress response (Figs. 2–4). By contrast, the XXM-mediated Ca$^{2+}$ influx triggered an immune response and suggests [Ca$^{2+}$]$_{cyt}$ to represent the key signal. Further studies should identify turgor sensors or Ca$^{2+}$-dependent immune responses, possibly with the help of optotools. Of note, our studies are based on physiological responses observed in leaf cells, still leaving open the possibility that in distinct cell types—such as guard cells—signalling may be different[47]. Defined oscillations in [Ca$^{2+}$]$_{cyt}$ or membrane voltage are considered to address specific signalling networks and physiological plant responses[48–50]. Precise spatial activation of XXM or ACR1 may help to answer how [Ca$^{2+}$]$_{cyt}$ together with voltage signatures encodes

information to communicate local and long-range stress-related cues in plants (Extended Data Fig. 10). Moreover, considering the observed dynamic kinetics of optogenetically induced physiological responses (Figs. 3 and 4 and Extended Data Fig. 9f–m), switching off optotools could allow for temporally resolved investigation of the underlying restoring mechanisms. Given the wilting phenotype induced by ACR1, there is potential to screen for mutants that show enhanced osmolyte uptake, making them more resistant to salt stress or drought. Overall, experimental approaches in this area would enhance our understanding of plant plasticity dynamics and resilience to changing environmental conditions.

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

## Methods

### Ethics

The laparotomy to obtain oocytes from *X. laevis* was carried out in accordance with the principles of the Basel Declaration and recommendations of Landratsamt Wuerzburg Veterinaeramt. The protocol under License number 70/14 from Landratsamt Wuerzburg, Veterinaeramt, was approved by the responsible veterinarian.

### Molecular cloning

The pGEMHE vector[51] was used for in vitro RNA synthesis and the following expression in *X. aevis* oocytes. The binary pCAMBIA3300 vector with the UBQ10 promoter or the pCAMBIA1300 vector with the CaMV 35S promoter was used for *Agrobacterium* infiltration and stable transformation of plants.

Substitutions were introduced by QuikChange site-directed mutagenesis PCR. The pCambia1300 NES-2×R-GECO1 was cloned using the USER cloning technique[52]. For other constructs, all of the fragments with suitable restriction sites were introduced into the pGEMHE vector or the binary vectors pCAMBIA3300 with T4 DNA ligase (Thermo Fisher Scientific).

Plasmid extractions from *Escherichia coli* cultures were carried out using the MiniPrep 250 Kit (QIAGEN) according to the manufacturer's instructions. All constructs were verified by sequencing (Eurofins Genomics). The sequences of XXM 1.1 and XXM 2.0 are available in the Supplementary Information.

### Molecular engineering and development of XXM 2.0

ChR2-XXM (D156H) was selected as a template to start molecular engineering owing to the high expression level and already enhanced $Ca^{2+}$ permeability[19]. As no specific ion-selective filter is identified in ChR2, we anticipated that the three (extracellular, centre and intracellular) molecular gates (ECG, CG and ICG) might contribute to modulating ion-selective properties of ChR2, in addition to its gating function. Considering known substitution sites of ChR2 variants with modified ion selectivity in relation to the crystal structure indeed suggests a possibility to modulate ion selectivity using the ChR2 structure generated by PyMOL[53] (Extended Data Fig. 1b). Within the three-dimensional structure, the internal cavities were calculated using HOLLOW, which is a published python script[54]. Some gating residues (H134 and E83 in ICG, and Q117 and R120 in ECG) and residues located in close proximity to the gate (E101 near ECG) were therefore substituted (Extended Data Fig. 1b). S63 (when combined with D156H) and E90 substitutions in CG have been reported to decrease $Ca^{2+}$ conductance[55,56], and therefore were not tested here. The resulting variants were first tested by two-electrode voltage-clamp analysis with *X. laevis* oocytes for comparison of photocurrent amplitude (Extended Data Fig. 1c). Variants exhibiting high photocurrent amplitude were next studied to determine the ion selectivity (reversal potential shift comparison) by changing extracellular ion concentrations (Extended Data Fig. 1d, see details in the following section) for selecting the superior candidate with high $Ca^{2+}$ conductance (Fig. 1). To further enhance its membrane trafficking, N-terminal truncation of ChR2 and addition of signal peptides as recently reported[4] were explored and examined by fluorescence imaging along with a comparison of photocurrents in *X. laevis* oocytes.

### Two-electrode voltage-clamp analysis with *X. laevis* oocytes

The AmpliCap-MaxT7 High Yield Message Maker Kit (Epicentre Biotechnologies) was used to synthesize XXM, XXM 1.1, XXM 1.2, XXM 1.3 and XXM 2.0 complementary RNA (cRNA). All of the cRNAs were stored in nuclease-free water at −20 °C. Oocytes were injected with 30 ng cRNA and incubated in ND96 solution (96 mM NaCl, 2 mM KCl, 1 mM $CaCl_2$, 1 mM $MgCl_2$, 10 mM 4-(2-hydroxyethyl)-1-piperazineethanesulfonic acid (HEPES), pH 7.4) at 16 °C for 2 days. Two-electrode voltage-clamp recordings were carried out at room temperature with a two-electrode

voltage-clamp amplifier (TURBO TEC-03X, NPI Electronic). Electrode capillaries (diameter 1.5 mm, wall thickness 0.178 mm; Hilgenberg) were pulled by a vertical puller (PC-10, Narishige) and filled with 3 M KCl, with tip resistance of 0.4–1 MΩ. A USB-6221 DAQ interface (National Instruments) and WinWCP V5.3.4 software (University of Strathclyde, UK) were used for data acquisition. Blue light illumination was supplied by a 473-nm laser (Changchun New Industries Optoelectronics Tech).

Photocurrents of different XXM versions in response to blue light illumination (473 nm, 3 mW $mm^{-2}$) were compared at a holding potential of −90 mV in a standard recording solution (96 mM NaCl, 2 mM KCl, 1 mM $BaCl_2$, 1 mM $MgCl_2$, 10 mM HEPES, pH 7.6). For the reversal potential shift (mV) comparison, photocurrents of XXM and XXM 1.1 were recorded in both buffer A (119 mM *N*-methyl-D-glucamine, 0.8 mM $BaCl_2$, 5 mM HEPES, pH 7.6) and buffer B (80 mM $BaCl_2$, 5 mM HEPES, pH 7.6). Blue light (473 nm, 3 mW $mm^{-2}$) was applied to activate different XXM variants. In the experiment for the comparison of calcium currents between XXM and XXM 1.1, 10 mM 1,2-bis(*o*-aminophenoxy) ethane-*N*,*N*,*N′N′*-tetraacetic acid (BAPTA) was injected to block the endogenous $Ca^{2+}$-activated chloride channels of oocytes. The $Ca^{2+}$ currents triggered by 3 mW $mm^{-2}$ blue light were measured in the bath solution containing 80 mM $CaCl_2$ and 10 mM 3-(cyclohexylamino)-2-hydroxy-1-propanesulfonic acid (CAPSO) at pH 9 and a holding potential of −90 mV.

The action spectrum for XXM 2.0 was detected with light of different wavelengths. Light of different wavelengths (399.3 nm, 422 nm, 440.7 nm, 456 nm, 479.5 nm, 496 nm, 516 nm, 541 nm, 562 nm and 595 nm) was obtained by narrow bandwidth interference filters (Edmund Optics) together with a PhotoFluor II light source (89 North). Equal photon flux was set for each wavelength. Photocurrents were detected when treated with light of distinct wavelength and normalized to the maximal stationary currents triggered by blue light (456 nm). The light intensities were measured with a Plus 2 Power & Energy Meter (Laserpoint).

### *Agrobacterium* transformation

The *Agrobacterium tumefaciens* strain GV3101 was collected by centrifugation after 1 day of culture and washed twice with sterile distilled water. All of the plasmids described for plant expression were transformed into *A. tumefaciens* by an electroporation protocol[57]. Monoclonal colonies were selected from lysogeny broth (LB)–agar plates with 100 μg $ml^{-1}$ kanamycin, 25 μg $ml^{-1}$ gentamycin and 10 μg $ml^{-1}$ rifampicin at 28 °C and cultured in LB medium with 100 μg $ml^{-1}$ kanamycin, 25 μg $ml^{-1}$ gentamycin and 10 μg $ml^{-1}$ rifampicin at 28 °C. The colonies were confirmed by PCR.

### *Agrobacterium* infiltration of *N. benthamiana* leaves

For *Agrobacterium* infiltration, 30–37-day-old *N. benthamiana* plants grown in the greenhouse (40–60 kilolux light irradiation from 08:00–20:00, 24–26 °C) were used. Transient transformation of *N. benthamiana* plants was carried out according to the protocol of ref. 58. Briefly, agrobacteria were cultured in LB medium with 150 μM acetosyringone at 28 °C for about 16–18 h, collected by centrifugation and washed twice with infiltration buffer (10 mM $MgCl_2$, 10 mM 2-morpholinoethanesulfonic acid (MES) (pH was adjusted to 5.6 by KOH), 150 μM acetosyringone). The final concentration was adjusted to 0.4 at an optical density of 600 nm ($OD_{600nm}$) in infiltration buffer. A 1-ml syringe was used to infiltrate the resuspended agrobacteria into the leaves through the abaxial epidermis. The infiltrated plants were grown in 650 nm red light (light intensity of about 30 μW $mm^{-2}$; cycles of 14 h light at 26 °C/10 h dark at 16 °C).

### Stable transformation of *N. tabacum* plants

*N. tabacum* (*N. tabacum* cultivar Petit Havana SR1) seeds sterilized by 6% NaOCl were germinated and grown in 500-ml sterile plastic boxes on agar plates (Murashige and Skoog medium including vitamins and

MES (Duchefa Biochemie), 3% sucrose, 0.8% Gelzan (Sigma-Aldrich), pH 5.8 with KOH) in stable culture conditions of cycles of 14 h light at 26 °C/10 h dark at 16 °C. The *A. tumefaciens* strain GV3101 harbouring the pCAMBIA3300 vector with BASTA resistance or the pCAMBIA1300 vector with hygromycin resistance was used for *N. tabacum* transformation as described previously[59] with some minor modifications. Agrobacteria were collected and washed twice with sterilized MS solution (Murashige and Skoog medium including vitamins and MES (Duchefa Biochemie), 3% sucrose, pH 5.8 with KOH). The final concentration was adjusted to $OD_{600nm}$ = 0.1. Sterilized leaves were cut into pieces of about 2 $cm^2$ and soaked in the resuspended agrobacteria solution for 20 min. The wet leaf pieces were dried, placed on plant growth medium and transferred after 3 days to a callus-inducing medium (Murashige and Skoog medium including vitamins and MES (Duchefa Biochemie), 3% sucrose, 0.8% Gelzan (Sigma-Aldrich), pH 5.8 with KOH, 20 μg $ml^{-1}$ DL-phosphinothricin (Duchefa Biochemie) or 30 mg $l^{-1}$ hygromycin B (Thermo Fisher Scientific), 500 μg $ml^{-1}$ ticarcillin disodium (Duchefa Biochemie), 100 mg $l^{-1}$ myo-inositol, 1 mg $l^{-1}$ thiamine hydrochloride, 1 mg $l^{-1}$ 6-benzylaminopurine and 100 μg $l^{-1}$ 1-naphthaleneacetic acid (Sigma-Aldrich)) and cultured under 650-nm light-emitting diode (LED) red light with light intensity of about 30 μW $mm^{-2}$. The pieces of leaves, explants and calli were transferred to new callus-inducing medium every 2 weeks. Generated shoots were decapitated and moved onto rooting medium (Murashige and Skoog medium including vitamins and MES (Duchefa Biochemie), 3% sucrose, 0.8% Gelzan (Sigma-Aldrich), pH 5.8 with KOH, 20 μg $ml^{-1}$ DL-phosphinothricin (Duchefa Biochemie) or 30 mg $l^{-1}$ hygromycin B (Thermo Fisher Scientific), 500 μg $ml^{-1}$ ticarcillin disodium (Duchefa Biochemie), 100 mg $l^{-1}$ myo-inositol, 1 mg $l^{-1}$ thiamine hydrochloride)) and, after root formation, were grown on soil in 650 nm red light (about 30 μW $mm^{-2}$, cycles of 14 h light at 26 °C/10 h dark at 16 °C).

The transgenic plants were verified by eYFP fluorescence or R-GECO1 fluorescence in leaves. Seeds of individual plants were collected and selected for BASTA or hygromycin resistance using selection medium (Murashige and Skoog medium including vitamins and MES (Duchefa Biochemie), 3% sucrose, 0.8% Gelzan, pH 5.8 with KOH, 20 μg $ml^{-1}$ DL-phosphinothricin (Duchefa Biochemie) or 30 mg $l^{-1}$ hygromycin B (Thermo Fisher Scientific)). Homozygous lines were used for experimental studies.

### Confocal microscopy and image processing

A confocal laser scanning microscope (Leica SP5, Leica Microsystems CMS) controlled by Leica LAS AF (Version 2.7.3.9723, Leica Microsystems) was used to subcellularly localize the rhodopsin–eYFP fusions in plant cells or *X. laevis* oocytes. The yellow fluorescence was observed with a dipping 25× HCX IRAPO 925/0.95 objective in *N. benthamiana* leaves 3 days post *Agrobacterium* infiltration, in *N. tabacum* leaves after 45 days grown in red light (light intensity of about 30 μW $mm^{-2}$; cycles of 14 h light at 26 °C/10 h dark at 16 °C) and in *X. laevis* oocytes 2 days post injection. eYFP was excited at 496 nm and fluorescence was captured between 520 and 580 nm. *N. benthamiana* and *N. tabacum* leaf discs were placed upside down for yellow fluorescence detection. FIJI IMAGEJ-win64 software[60] was used for image processing.

### Aequorin-based cytoplasmic free $Ca^{2+}$ measurements

Co-infiltration of agrobacteria with 10 μM of coelenterazine (PJK Biotech) was carried out as described in the '*Agrobacterium* infiltration of *N. benthamiana* leaves' section under red light conditions. After 2 days in the red light growth room, aequorin luminescence from the infiltrated leaves was measured by a homemade luminometer. Luminescence was detected by a photomultiplier (Photo Counting Module MP 1983 RS CPM, Perkin Elmer) controlled by IGI-MPRS232 (IGIsystems). Labview 14.0.0 (National Instruments) was used to control the shutter and LEDs. A 520-nm green light LED (from WINGER, WEPGN3-S1) with a light intensity of 50 μW $mm^{-2}$ was used to activate rhodopsins.

To prevent the LED light from being detected by the photomultiplier, an additional shutter (Uniblitz, VCM-D1) was installed.

### Live-cell imaging and all-optical physiology measurements

Live-cell imaging experiments were carried out using transiently transformed *N. benthamiana* leaves, transgenic *N. tabacum* leaves (Ret-XXM 2.0 with R-GECO1, Ret-ACR1 2.0 with R-GECO1, R-GECO1, Ret-XXM 2.0 with pHuji and pHuji) or transiently transformed *N. benthamiana* mesophyll protoplasts (R-GECO1 and Ret-XXM 2.0 with R-GECO1). The *A. tumefaciens* strain GV3101 harbouring the corresponding pCAMBIA vectors and *A. tumefaciens* strain K19 were cultured at 28 °C overnight. The infiltration solution for co-expression of Ret-XXM 2.0 with R-GECO1 contained *A. tumefaciens* strain K19 ($OD_{600nm}$ = 0.3) and *A. tumefaciens* strain GV3101 harbouring the pCAMBIA plasmids (pCAMBIA3300 vector carrying Ret-XXM 2.0, pCAMBIA1300 vector carrying NES-2× R-GECO1; $OD_{600nm}$ = 0.4 for both). Control plants were infiltrated with infiltration solution containing *A. tumefaciens* strain K19 ($OD_{600nm}$ = 0.3) and *A. tumefaciens* strain GV3101 harbouring the pCAMBIA1300 vector carrying NES-2× R-GECO1 ($OD_{600nm}$ = 0.4). $[Ca^{2+}]_{cyt}$ measurements were carried out 3 days post infiltration. Mesophyll protoplasts were prepared from the transiently transformed *N. benthamiana* leaves from which the abaxial epidermis was peeled off. The leaf pieces without the main vein were incubated in enzyme solution (1% BSA, 0.05% pectolyase Y23, 0.5% cellulase R-10, 0.5% macerozym R-10, 1 mM $CaCl_2$, 10 mM MES, 500 mM D-sorbitol, pH 5.6 with Tris) for 2 h. Following enzymatic digestion, cells were filtered through a 100-μm mesh. Protoplasts were collected by low-speed centrifugation (80*g*) without acceleration at 4 °C. Protoplasts were washed twice using precooled wash solution (1 mM $CaCl_2$, 500 mM D-sorbitol, pH 5.6 with Tris) and finally resuspended in precooled wash solution and stored on ice until use. Leaf disc samples with a diameter of 5 mm were prepared by peeling the abaxial epidermis off and gluing leaf discs upside down with medical adhesive (ULRICH Swiss) on custom-made recording chambers. The samples were allowed to recover in the dark in bath solution (1 mM KCl, 1 mM $CaCl_2$, 10 mM MES, and 1,3-bis(tris(hydroxymethyl)methylamino) propane (BTP), pH 6.0) at room temperature (about 25 °C) overnight before R-GECO1 or pHuji fluorescence measurement and all-optical experiments were carried out.

The microscope setup to carry out live-cell imaging is described in detail elsewhere[61]. R-GECO1[25] and pHuji[26] were excited with 570 nm excitation light. VisiView software (Version 2.1.1) was used to simultaneously control R-GECO1 imaging and triggering of local green light (532 nm, 180 μW $mm^{-2}$) illumination by a solid-state laser (Changchun New Industries Optoelectronics Tech) or global green light (520 nm, 9 μW $mm^{-2}$) illumination by a homemade LED device (LED from WINGER, WEPGN3-S1). XXM 2.0 activation by green light was carried out in the 5 s interval time during R-GECO1 or pHuji imaging. Green light illumination was terminated more than 1 s before R-GECO1 was excited to avoid photoswitching effects of R-GECO1 during optogenetic stimulation as reported previously[62]. A dichroic mirror (HC593 (F38-593), AHF Analysetechnik) combined with a high-speed filter wheel equipped with bandpass filters for R-GECO1 or pHuji (ET 624/20 nm) was used to detect the red fluorescence. During the detection of $[Ca^{2+}]_{cyt}$ signal triggered by XXM 2.0 stimulation when a different light condition was used, a 2 s interval time was set during R-GECO1 illuminations. It should be noted that the light pulse protocols used to set defined $[Ca^{2+}]_{cyt}$ signatures must be customized for each particular cell system or plant line and cannot act as a blueprint as features such as the expression level and cell type used or the $Ca^{2+}$ homeostasis will probably influence the signature.

To avoid undesirable XXM 2.0 or ACR1 2.0 activations during bright-field imaging, the microscope white light source was covered by a primary red filter (Lee filter 106). Simultaneous plasma membrane potential and R-GECO1 fluorescence recordings were carried out in R-GECO1, Ret-XXM 2.0 with R-GECO1, Ret-ACR1 2.0 with R-GECO1 samples. Current-clamp-based voltage recordings were carried out by

microelectrode impalement as described elsewhere[61]. Glass microelectrodes filled with 300 mM KCl were connected to the microelectrode amplifier (TEC-05X; NPI Electronic) equipped with head stages of more than $10^{13}$ Ω input impedance. The reference electrodes were filled with 300 mM KCl. A piezo-driven micromanipulator (Sensapex) was used to direct the glass electrode. The current-clamp protocols were applied by WinWCP V5.3.4 software (University of Strathclyde, UK). R-GECO1 fluorescence intensities were recorded using FIJI IMAGEJ-win64 software[60].

Ca$^{2+}$ signals were detected in *N. tabacum* leaves utilizing the R-GECO1 reporter following *Pst* treatment using a perfusion system (780 µl min$^{-1}$), which prevents motion and touch-induced imaging artefacts[47,63–65]. *Pst* inoculated from LB–agar plates containing 10 µg ml$^{-1}$ rifampicin was cultured in LB medium with 10 µg ml$^{-1}$ rifampicin for 1.5 days (28 °C, 200 r.p.m.) and subsequently subcultured in 500 ml LB medium containing 10 µg ml$^{-1}$ rifampicin for 16 h. *Pst* cells were washed twice using sterile deionized water and suspended in bath solution (1 mM KCl, 1 mM CaCl$_2$, 10 mM MES and BTP, pH 6.0), resulting in a final perfusion solution with an OD$_{600nm}$ of 0.5. Leaf disc samples from R-GECO1 transgenic *N. tabacum* plants without abaxial epidermis were glued with the adaxial side down to the coverslip in custom-made chambers using Medical Adhesive B (Ulrich Swiss) and allowed to recover in bath solution overnight before Ca$^{2+}$ imaging.

## Membrane voltage recordings in mesophyll cells

Microelectrodes for mesophyll cell impalement were pulled from borosilicate glass capillaries (inner diameter 0.58 mm, outer diameter 1.0 mm, Hilgenberg) using a horizontal laser puller (P2000, Sutter). Microelectrodes filled with 300 mM KCl having an electrode resistance of 60–110 MΩ were connected by Ag/AgCl wires to the microelectrode amplifier (Axon geneclamp 500 or VF-102; BioLogic). Reference electrodes were filled with 300 mM KCl, and plugged with 2% agar in 300 mM KCl. The NA USB-6221 interface (National Instruments) was used to digitalize data. Cells were impaled by an electronic micromanipulator (NC-30, Kleindiek Nanotechnik) and current-clamp protocols were applied with the WinWCP V5.3.4 software (University of Strathclyde, UK).

## Plant growth conditions and sample collection

All of the WT and transgenic *N. tabacum* plants were grown under constant red light (650 nm, 30 µW mm$^{-2}$, 26 °C) for 45 days. For ACR1 2.0 and XXM 2.0 stimulation, 0 h, 1 h, 4 h and 24 h additional global green light (520 nm, 9 µW mm$^{-2}$) were applied for the experimental groups. For osmotic stress treatment, 35% PEG was used to water Ret-eYFP #1 transgenic *N. tabacum* plants. This PEG concentration was selected after initial experiments with different PEG concentrations that would cause wilting phenotypes in a similar time frame (after 4–5 h) to that for the green-light-treated Ret-ACR1 2.0 plants. For *Pst* treatment, the *Pst* was washed twice with sterile deionized water and suspended in 10 mM MgCl$_2$ containing 0.04% Silwet L-77 with a final OD$_{600nm}$ of 0.5 and sprayed on the entire Ret-eYFP #1 transgenic *N. tabacum* plants. *Pst* treatment was applied by spray inoculation to prevent wounding effects taking place that would unequivocally occur during infiltration by a syringe. The same amount (about 25 ml for each plant) of 10 mM MgCl$_2$ containing 0.04% Silwet L-77 was sprayed on Ret-eYFP #1 transgenic *N. tabacum* as the negative control. At $t = 0$ h before the additional green light illumination, PEG, *Pst* or 10 mM MgCl$_2$ was applied. For plants treated with 0 h, 1 h and 4 h green light illumination, the fifth leaves of *N. tabacum* were collected at different time points ($t = 0$ h, 1 h, 4 h and 8 h) and frozen in liquid nitrogen quickly for metabolite measurement and transcriptomics analysis. For plants treated with 24 h global green light, PEG, *Pst* or 10 mM MgCl$_2$, the fifth leaves from *N. tabacum* at different time points ($t = 0$ min, 3 min, 10 min, 0.5 h, 1 h, 2 h, 4 h, 8 h and 24 h) were used for detection and quantification of ROS, electrolyte leakage estimations, chlorophyll fluorescence detection, metabolite measurement and transcriptomics analysis.

## Detection of necrosis in leaf discs

A tissue puncher (Stiefel Disposable biopsy punch, diameter of 6 mm) was used to prepare leaf discs from the fifth leaf of 45-day-old *N. tabacum* plants. Leaf discs were washed twice with deionized water and transferred into 24-well plates containing 0.4 ml ultrapure H$_2$O that contained 1 mM CaCl$_2$, 10 mM CaCl$_2$, 5 mM EGTA (pH 7.0, KOH), 5 mM K$_4$BAPTA or 5 mM K$_4$BAPTA plus 10 mM NaCl as indicated in Extended Data Fig. 5c,d. Samples were placed in the dark for 1 h before exposing them to the light condition (growth chamber with constant red light (650 nm, 30 µW mm$^{-2}$) plus constant green light (520 nm, 9 µW mm$^{-2}$, 26 °C)). Images were captured after 24 h treatment.

## Chlorophyll fluorescence measurements

*N. tabacum* plants treated as described in the 'Plant growth conditions and sample collection' section were used to quantify photosynthesis performance with a pulse-amplitude modulation fluorometer. The fifth leaf was fixed and monitored with a Maxi pulse-amplitude modulation fluorometer (AVT 033); chlorophyll fluorescence measurements were recorded with IMAGING WIN v.2.41a FW MULTI RGB (Walz). The dark-adapted *N. tabacum* leaf was exposed to actinic light with intensity 7 (photosynthetically active radiation as 146 µmol m$^{-2}$ s$^{-1}$). The maximal fluorescence yield of a dark-adapted sample (Fm) and dark-level fluorescence yield (Fo) were detected. The quantum yield of the dark-adapted leaf samples is a measure of the potential quantum yield of the samples, which was calculated according to the equation: Yield = (Fm − Fo)/Fm.

## Electrolyte leakage estimations

The membrane integrity of *N. tabacum* leaf cells was estimated by electrolyte leakage of leaf samples. Ion conductivity was measured as described previously[66]. Seven leaf discs (5 mm diameter) were detached from the fifth leaves of 45-day-old plants and equilibrated together in 0.3 ml of ultrapure H$_2$O after washing twice with ultrapure H$_2$O. Ion conductivity was quantified 20 min after leaf disc equilibration in ultrapure H$_2$O (EC1) using a LAQUAtwin EC-11 conductivity meter (Horiba). The samples were then heated at 99 °C for 1 h to measure the final electrical conductivity (EC2) when the samples reached room temperature again. The relative electrolyte leakage was calculated, as a percentage, by the formula: EL = EC1/EC2 × 100.

## Detection of ROS

Chemical detection of ROS in green light (520 nm, 9 µW mm$^{-2}$)-treated *N. tabacum* leaves was carried out by 3,3-diaminobenzidine (DAB, Sigma-Aldrich) staining as described previously[67]. The fifth leaves of *N. tabacum* plants were stained with fresh DAB staining solution (10 mM Na$_2$HPO$_4$, 1 mg ml$^{-1}$ DAB, 0.05% Tween 20, pH 3.0) by application of negative pressure for 5 min in dark. After 5 h incubation (shaking speed of 80–100 r.p.m.), the stained leaves were moved into fresh chlorophyll destaining solution (ethanol/acetic acid/glycerol, 3:1:1) and bathed in hot water (about 90–95 °C) for 15 min. Finally, the stained leaves were put into cold fresh chlorophyll destaining solution for 30 min. Images were taken with a plain white background under uniform lighting.

The chemiluminescent 'superoxide probe' luminol can be applied to indicate the ROS production[68]. Superoxide released from leaf tissues was detected by the luminescence of luminol with Skanlt software (Version 6.1) according to the method described previously[69] with minor modifications. Leaf discs were prepared from the fifth leaves of 45-day-old *N. tabacum* plants using a tissue puncher (Stiefel Disposable biopsy punch, diameter of 6 mm). Leaf discs were washed with deionized water twice and transferred into the 96-well assay plate (black plate, clear bottom with lid, Corning) and incubated in the dark overnight to recover. Water was replaced with 200 µl of luminol–peroxidase working solution (30 mg l$^{-1}$ luminol (Sigma) and 20 mg l$^{-1}$ horseradish peroxidase (Sigma)) in each well containing leaf discs. Samples were kept in the dark for 1 h before measurement. Luminescence was measured in

a microplate reader (Luminoskan Ascent, Thermo Labsystems) and 5 min global constant green light (9 μW mm$^{-2}$) illumination was applied during the rest periods.

The method for in vivo measuring the production of $H_2O_2$ amperometrically from mesophyll cells in parallel with intracellular membrane potential recordings was described previously[70]. Measurements were carried out in standard bath solution (1 mM KCl, 1 mM CaCl$_2$, 10 mM MES, and BTP, pH 6.0) with a platinum–iridium electrode (MicroProbes) cut back to an active (uninsulated) area of about 1 mm length. ROS detection was carried out by Patch-Master software V2x90 (HEKA). The platinum–iridium disc was gently placed in close proximity to mesophyll cells and held at a constant voltage of 600 mV with an amperometry amplifier (VA 10X, NPI Electronic). Oxidation of $H_2O_2$ at the active microelectrode surface resulted in a positive current signal, which was low-pass-filtered at 1 Hz and recorded with Patch-Master software V2x90 (HEKA). The electrode was calibrated in freshly prepared bath solutions with defined $H_2O_2$ concentrations. Green light (520 nm, about 9 μW mm$^{-2}$) illumination on the N. tabacum leaves was carried out by green LEDs (WINGER, WEPGN3-S1).

## All-*trans* retinal and carotenoid measurements

All-*trans* retinal and carotenoids were measured according to a protocol published elsewhere[4]. The fifth leaf of transgenic N. tabacum plants grown for 45 days in red light was triturated in liquid nitrogen and 200 mg leaf material was extracted with 500 μl of chloroform. The extract was centrifuged for 5 min at 18,400g and 50 μl of the organic phase was evaporated in a SpeedVac at 40 °C and dissolved in 50 μl of a 1:1 ethanol and chloroform mixture. A 5 μl volume of dissolved solution was analysed by ultrahigh-performance liquid chromatography (UPLC) combined with ultraviolet and tandem mass spectrometry detection using a Waters Acquity UPLC system coupled to a Waters Quattro Premier triple-quadrupole mass spectrometer equipped with an electrospray interface. Ten plants were used for retinal and 12 plants were used for carotenoid quantification.

## Phytohormone measurement

All of the samples were prepared as described in the 'Plant growth conditions and sample collection' section. Ground samples (150 mg) were lyophilized in a laboratory freeze dryer (CHRIST, Laboratory freeze dryer Alpha 1-2) and subsequently used for phytohormone extraction. The extraction and chromatographic separation was carried out as described previously[71], using 5 ng of dihydro-JA, JA–norvaline, [$^{18}O_2$] OPDA, [D$_4$]SA and [D$_6$]ABA as phytohormone internal standard. The extraction solution contained ethylacetate (p.a.) and formic acid (p.a.) (99:1 in volume) to which 5 ng phytohormone internal standard was added. All samples were fixed on a TissueLyser with shaking for 3 min at a speed of 23 Hz. After that, samples were centrifuged and the supernatant was dried in a SpeedVac at 45 °C and finally dissolved in 40 μl liquid (acetonitrile (for high-performance liquid chromatography))/water (MilliQ), 1:1 (v/v)). Phytohormones were analysed by UPLC–electrospray interface–tandem mass spectrometry using a Waters Acquity I-Class UPLC system coupled to an AB Sciex 6500+ QTRAP tandem mass spectrometer (AB Sciex), operated in negative ionization mode as described elsewhere[72,73]. Analyst (Version 1.6.3) software and MultiQuant (Version 3.0.2) software from Sciex were used for mass spectrometry detection of hormones and metabolites.

## Proline quantification

The proline content of N. tabacum leaves was measured according to the spectroscopic method of ref. 74. Samples were prepared as described in the section 'Plant growth conditions and sample collection'. Ground samples (150 mg) were lyophilized in a laboratory freeze dryer (CHRIST, Laboratory freeze dryer Alpha 1-2). The dry samples were mixed in 40% ethanol and incubated at 4 °C overnight. The supernatant was collected after centrifugation at 13,500g for 5 min. A 500 μl volume

of the ethanol extraction or 100 μl standard solution was mixed with 1,000 μl reaction mix (1% ninhydrin (w/v) in 60% acetic acid (v/v) and 20% ethanol (v/v)). After incubation at 95 °C for 20 min and subsequent centrifugation for 1 min at 9,000g, the samples (supernatant) were subjected to absorption measurement at 520 nm with a spectrophotometer (Hitachi U-1500). Proline concentration was determined according to the standard curve, and concentrations were calculated on the basis of dry weight.

## Transcriptomics analysis

Three replicates of leaf samples from two batches of N. tabacum plants were collected for RNA sequencing. The experimental design for transcriptomics is shown in Extended Data Fig. 8a. Samples at 0 h, 1 h, 4 h and 8 h from plants growing in red light conditions were used as the biological controls to compare the expression levels of Ret-XXM 2.0 and Ret-ACR1 2.0 transgenic plants during or after the global green light illumination. In these experiments PEG-treated plants and *Pst* inoculation served as possible physiological controls to ACR1 2.0 and XXM 2.0 activation, and spraying leaves with buffer (MgCl$_2$) served as an additional control to *Pst*-sprayed plants (Extended Data Fig. 8a). RNA extraction was carried out using ground samples (150 mg) with the Macherey-Nagel NucleoSpin RNA Plant Kit (https://www.takarabio.com/documents/User%20Manual/UM/UM_TotalRNAPlant_Rev_07.pdf). DNase1 (Thermo Fisher) was used to digest DNA. RNA sequencing was carried out by Novogene (UK) with an Illumina NovaSeq 6000 Sequencing System. Paired-end 150 bp was the read length. Data processing (fastp) and mapping to the N. tabacum genome (kallisto)[75] was carried out using Amalgkit (https://github.com/kfuku52/amalgkit). Functional annotations of N. tabacum genes for subsequent bioinformatic analyses were retrieved from the Dicots PLAZA 5.0 repository[76,77].

Normalization and DEG analysis were carried out employing the DIANE package using DESeq2 and default parameters[77]. DESeq2 uses a Wald test, in which the shrunken estimate of log fold change is divided by the standard error to produce a z-statistic. This z-statistic is then compared against a standard normal distribution[78]. A prefiltering step eliminated genes exhibiting rowMeans over all conditions ≤ 5 counts, reducing the number of input genes from 69,500 to 42,196. Unless stated otherwise, |log$_2$[fold change]| ≥ 2 with a false discovery rate of 0.01 was taken as the cutoff for DEG identification. Only a small number of DEGs were identified when comparing Ret-ACR1 2.0 or Ret-XXM 2.0 transgenes with Ret-eYFP control plants grown under non-stimulating red light conditions (Supplementary Table 2, tab 1), demonstrating that ACR1 and XXM expression have virtually no impact on the transcript profiles of plants and plants growing in red light conditions are proper biological controls. Likewise, GO enrichment analysis on DEGs was carried out using the DIANE suite and corresponding enrichment plots were created using the srplot web interface (https://www.bioinformatics.com.cn/en). Venn diagrams were generated with the GOVenn script of the GOPlot package[79] and subsequent GO analysis on Venn subsets was carried out using gprofiler2[80]. g:Profiler functional enrichment analysis is conducted using the g:GOSt tool that carries out over-representation analysis via the hypergeometric test[80]. Finally, heat maps were generated with the pheatmap R package (version 1.0.12; https://CRAN.R-project.org/package=pheatmap). N. tabacum genes were further annotated manually according to their A. thaliana orthologues[81] and corresponding gene symbols from the Aramemnon database[82].

## Quantitative real-time PCR

Quantification of gene transcripts was carried out by real-time PCR as described elsewhere[83]. The samples were prepared as described in the section 'Plant growth conditions and sample collection'. Ground samples (100 mg) were used for RNA extraction by the NucleoSpin RNA Plant Kit (Macherey-Nagel). cDNA was synthesized from 2.5 g of total RNA using oligo(dT) primer (Thermo Fisher Scientific) and M-MLV Reverse Transcriptase (Promega). All quantitative real-time

PCR reactions were carried out with the Eppendorf Mastercycler ep realplex 2 system and Eppendorf Mastercycler ep realplex (Version 2.2) software, in a 20 µl reaction volume containing 2 µl diluted cDNA, 0.8 µM primer pairs and 10 µl ABsolute qPCR SYBR Green Capillary Mix (Thermo Scientific). Information on the genes and primers used is provided in Supplementary Table 7. Transcripts were normalized to that of 10,000 molecules of actin.

## Surface potential recording on *N. tabacum* leaves

The design for long-range electrical signal measurements in *N. tabacum* leaves is shown in a diagram in Extended Data Fig. 10a. The surface potential recordings were carried out on 6–7-week-old *N. tabacum* leaves according to a previously described protocol[84] with minor modifications. A USB-6221 interface (National Instruments) was used to digitalize the electrical signals, which were recorded with WinWCP V5.3.4 software (University of Strathclyde, UK). The electrode silver wires (Ag/AgCl) connected to a microelectrode amplifier (Axon geneclamp 500) were wrapped around the petiole of the fifth leaves gently. Electrode gel (Auxynhairol) was used to cover the surface of the wrapped wires to aid connectivity between the electrodes and the petiole. The reference electrode consisting of an Ag/AgCl electrode was placed in a 200-ml pipette tip filled with electrode gel (Auxynhairol), which was inserted in the soil of the pots the *N. tabacum* plants grew in. Nine hours after mounting the electrodes, the surface potential was recorded when applying a 600-ms green light (532 nm, 5.3 mW mm$^{-2}$) pulse at the main vein. A popular Technology Enhanced Clad Silica multimode optical fibre (diameter of 1,500 µm, 0.39 NA, Thorlabs) was placed directly on the top of main vein for light application. To prevent scattering of light and to guarantee local green light application, the optical fibre was covered by a non-transparent black plastic pipe up to the tip. The electrical signals were monitored at exactly 5 cm away from the illumination spot.

## Significance analysis

Student's *t*-test or ANOVA was used to analyse significant differences between groups. Significance analysis among more than three groups was carried out with one-way ANOVA using IBM SPSS statistics (version 26.0). For the post hoc multiple comparisons, the homogeneity of variances was tested first. If the variances were homogeneous ($P > 0.05$), the Tukey test was used for significance analysis. If the variances were not homogeneous ($P < 0.05$), either the Dunnett T3 or Games-Howell test was chosen, depending on whether the sample sizes were equal or not. Different letters indicate significant differences among the samples (lowercase letters indicate *P* values at the 0.05 level and capital letters indicate *P* values at the 0.01 level). Significance analysis among two groups was carried out with a two-sided Student's *t*-test. *$P \leq 0.05$, **$P \leq 0.01$ and ***$P \leq 0.001$. The significance analysis is performed with a 95% confidence of interval. All of the *P* values are listed in Supplementary Table 8.

## Reporting summary

Further information on research design is available in the Nature Portfolio Reporting Summary linked to this article.

## Data availability

All the data generated in this study are available in the paper and the Supplementary Information. The RNA-sequencing data have been deposited in the National Center for Biotechnology Information database (Bioproject ID: PRJNA1108451). Source data are provided with this paper.

## Code availability

No customized code was generated in this study.

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

**Acknowledgements** This study is financed by the Deutsche Forschungsgemeinschaft (DFG) Projektnummer 374031971 TRR 240 A04 and 417451587 granted to G.N. Financial support provided by the Prix Louis-Jeantet to G.N. and by the DFG to K.R.K. (KO3657/2-3) and R.H. (Reinhart Koselleck Grant 415282803) is acknowledged. Y.Z. and M.D. were supported by

the Chinese Scholarship Council and the German Academic Exchange Service (DAAD). In addition M.D. was supported by DFG Projekt 415282803 to R.H. We thank J.P. Rathjen and C. Zipfel for sharing seeds of aequorin-expressing *N. benthamiana* plants. *Pst* isolated by B. Staskawicz was provided by S. Berger. *X. laevis* oocytes were either bought from EcoCyte Bioscience or obtained from frogs in the Julius-von-Sachs-Institute, Wuerzburg University. We thank F. Waller for sharing expertise and equipment for ion conductivity measurements and K. Fukushima for providing the AMALGKIT pipeline and transcript mapping (https://github.com/kfuku52/amalgkit). Open access funding was enabled and organized by the DEAL-Agreement.

**Author contributions** G.N., S.G., M.K., M.J.M., R.H., D.B. and K.R.K. conceived the project. S.G., K.R.K., M.D. and G.N. designed and S.G., K.R.K. and G.N. supervised the experiments. M.D., Y.Z., J.Y.-S. and S.G. carried out the in planta voltage recordings, confocal microscopy and aequorin experiments and analysed the data. Y.Z., S.Y. and M.D. generated all constructs used in this study. M.D. generated the transgenic plants and carried out metabolite analysis, ion leakage measurement, quantitative real-time PCR, Ca²⁺ and pH imaging experiments, and phenotype characterization. S.Y. and S.G. designed the development of XXM 2.0 and S.Y. carried out the functional characterization in *X. laevis* oocyte experiments. M.K. and M.D. carried out the hormone and metabolite analysis. M.D. and S.S. carried out ROS detection and analysed the data. D.B. and M.D. identified the orthologues of the *N. tabacum* genes and designed primers for quantitative real-time PCR. D.B. carried out all bioinformatic analyses. K.R.K., M.D., S.G., D.B., R.H. and G.N. wrote the paper and the initial draft was written by K.R.K. All authors edited and approved the final version of the manuscript to be published.

**Funding** Open access funding provided by Julius-Maximilians-Universität Würzburg.

**Competing interests** The authors declare no competing interests.

**Additional information**
**Correspondence and requests for materials** should be addressed to Rainer Hedrich, Georg Nagel, Shiqiang Gao or Kai R. Konrad.

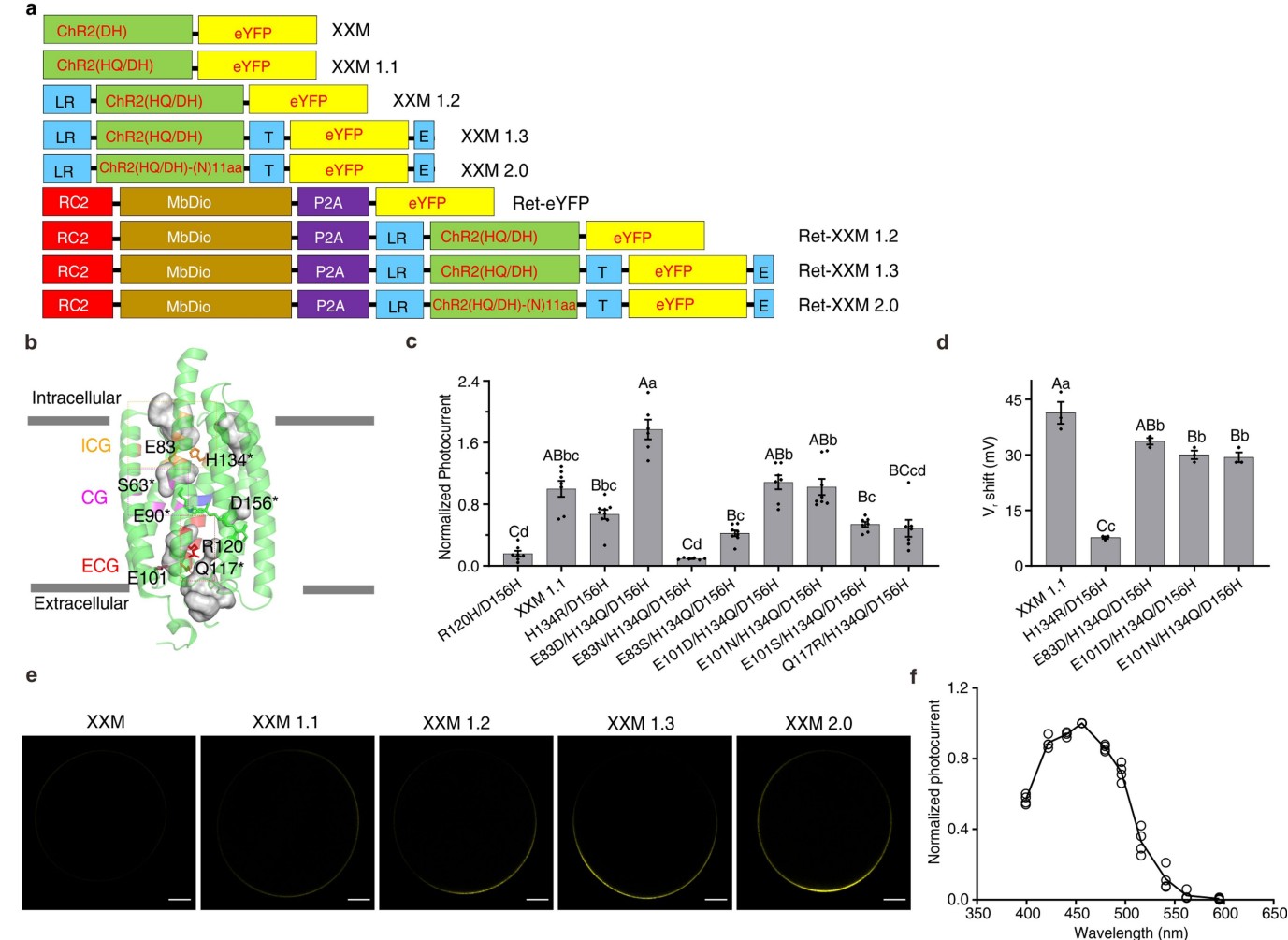

**Extended Data Fig. 1 | Improved channelrhodopsin variant XXM 2.0 with enhanced expression and plasma membrane targeting. a**, Scheme of DNA constructs used in this study. ChR2(DH), ChR2 D156H single mutant; eYFP, enhanced yellow fluorescent protein; ChR2(HQ/DH), ChR2 H134Q and D156H double mutant (XXM1.1 for short in this study); LR, the cleavable N-terminal signal peptide Lucy-Rho; T, the plasma membrane trafficking signal from Kir2.1; E, the endoplasmic reticulum (ER) export signal from Kir2.1; RC2, a synthetic chloroplast transit peptide; MbDio, a β-carotene 15,15′-dioxygenase from a marine bacterium; P2A, the self-cleaving peptide from porcine teschovirus; ChR2(HQ/DH)-(N)11aa, a truncated ChR2(HQ/DH) mutant lacking 11 amino acids at the N terminus named XXM 2.0 when the LR, T and E sequences were added. Sequences of XXM1.1 and XXM 2.0 are provided in the Supplementary File 1. **b**, Structure of ChR2 (PDB ID: 6EID) with highlighted retinal chromophore and molecular gates (red for extracellular molecular gates (ECG), magenta for center molecular gates (CG), orange for intracellular molecular gates (ICG), blue for 'DC gate'). Note that D156 was covered by the calculated internal cavity (in grey) of ChR2. Residues in these gates with strong influence on ion selectivity of ChR2 were marked by asterisk*. Side-chain highlighted residues were selected for mutagenesis analysis. **c**, Photocurrent comparison of ChR2 variants triggered by blue light (473 nm, 3 mW mm⁻²) in *Xenopus laevis* oocytes. Photocurrent was normalized to the mean photocurrent of XXM1.1. Error bars = s.e.m., n = 7, 7, 9, 6, 6, 8, 7, 8, 7, 7 cells from three batches of oocytes. Significance analysis was analyzed by One-way ANOVA followed by a Games-howell post hoc test. **d**, Reversal potential shift of ChR2 variants-expressing *Xenopus laevis* oocytes during blue light (473 nm, 3 mW mm⁻²) illumination. Error bars = s.e.m., n = 3 cells from two batches of oocytes. Significance analysis was performed by One-way ANOVA followed by a Tukey post hoc test. **e**, Representative confocal images of *Xenopus laevis* oocytes expressing different eYFP fused channelrhodopsin XXM variants as indicated above the images. Scale bar = 100 μm. n = 6 cells from two batches of oocytes. **f**, Photocurrents of XXM 2.0-expressing *Xenopus laevis* oocytes following light illumination of different wavelengths (399.3, 422, 440.7, 456, 479.5, 496, 516, 541, 562, 595 nm). Equal photon flux was set for each wavelength. The action spectrum of XXM 2.0 was normalized to photo-stimulation at 456 nm. Error bars = s.e.m., n = 4 from two batches of oocytes.

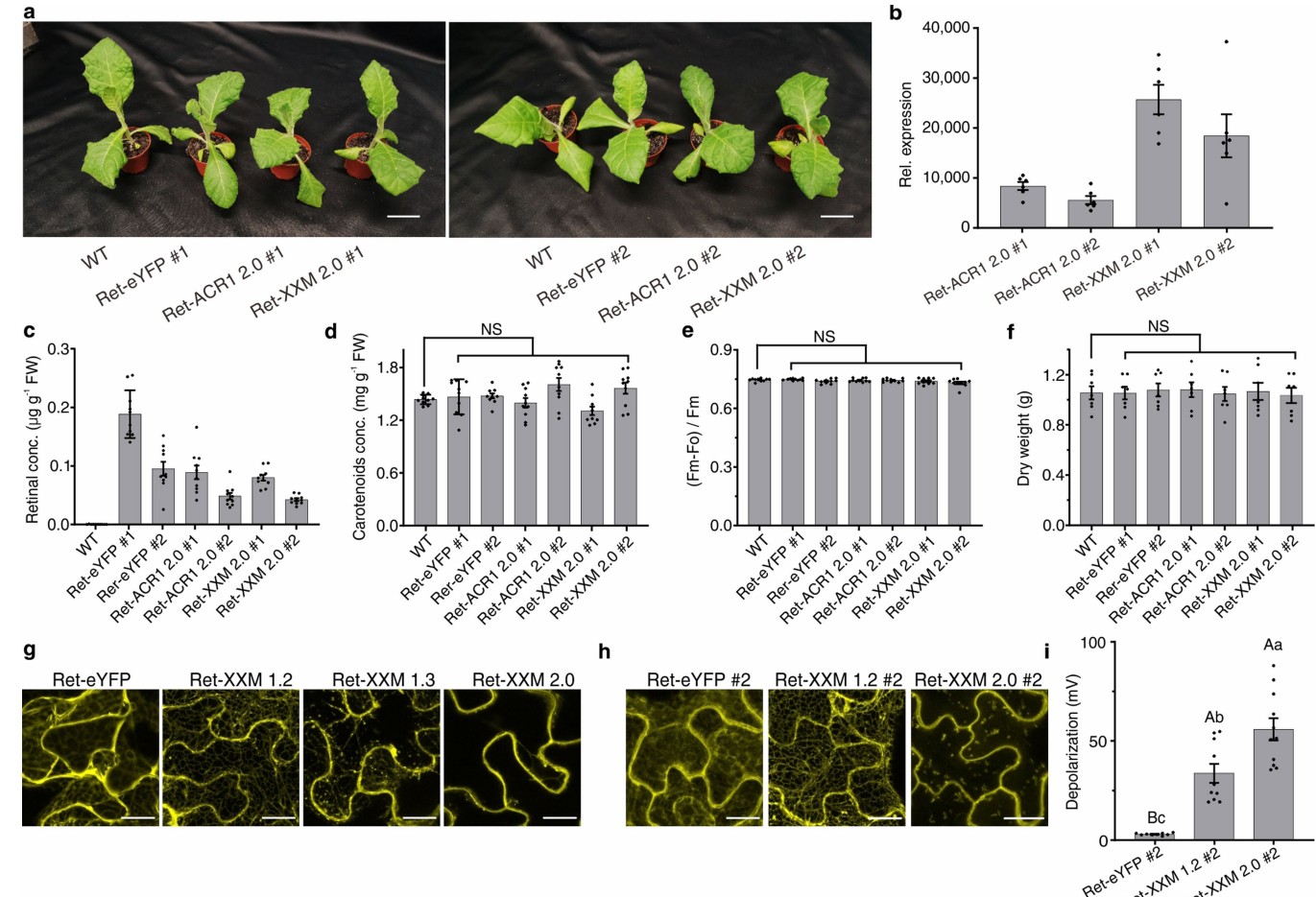

**Extended Data Fig. 2 | Transgenic *N. tabacum* plants grow like wild-type plants in red light condition. a**, WT, Ret-eYFP (#1 and #2), Ret-ACR1 2.0 (#1 and #2) and Ret-XXM 2.0 (#1 and #2) *N. tabacum* plants grown in constant red light (30 µW mm⁻²) for 45 days. Scale bar = 5 cm. n = 6 plants from two batches of *N. tabacum* plants. **b**, Transcript level of Ret-ACR1 2.0 (line #1, line #2) and Ret-XXM 2.0 (line #1, line #2) in transgenic plants. The transcript numbers of genes were normalized to 10,000 actin molecules. Error bars = s.e.m., n = 6 leaves from two batches of *N. tabacum* plants. **c**, Retinal concentrations measured in extracts from WT plants, Ret-eYFP, Ret-ACR1 2.0 and Ret-XXM 2.0-expressing *N. tabacum* lines; leaves were collected after 45 days' grown in red light. Error bars = s.e.m., n = 10 leaves from two batches of *N. tabacum* plants. **d**, Carotenoid concentrations measured from the same batches of plants as in **c**. Error bars = s.e.m., n = 10 leaves from two batches of *N. tabacum* plants. One-way ANOVA followed by Dunnett T3 post hoc test was performed for significance analysis. **e**, Maximum quantum yield of energy conversion in leaves of WT, Ret-eYFP (line #1, line #2), Ret-ACR1 2.0 (line #1, line #2) and Ret-XXM 2.0 (line #1, line #2) *N. tabacum* plants grown under red light condition. Error bars = s.e.m., n = 10 leaves from three batches of *N. tabacum* plants.

Significance analysis was analyzed by one-way ANOVA following a Tukey post hoc test. **f**, Dry weight of *N. tabacum* plants' aboveground part when grown in red light for 45 days. Error bars = s.e.m., n = 7 plants from two *N. tabacum* plants batches. Significance analysis was analyzed by one-way ANOVA following a Tukey post hoc test. **g**, Representative confocal images of *N. benthamiana* leaf epidermal cells transiently expressing Ret-eYFP, Ret-XXM 1.2, Ret-XXM 1.3 and Ret-XXM 2.0. Images were taken 3 days post *Agrobacterium* infiltration, scale bar = 20 µm. n = 7 leaves from two batches of *N. benthamiana* plants. **h**, Representative confocal images of transgenic *N. tabacum* leaf epidermal cells expressing Ret-eYFP (line #2), Ret-XXM 1.2 (line #2), Ret-XXM 2.0 (line #2). Scale bar = 20 µm. n = 6 leaves from two batches of *N. tabacum* plants. **i**, Mesophyll cell plasma membrane depolarizations induced by 5 s green light (532 nm, 180 µW mm⁻²) were compared for WT, Ret-eYFP, Ret-XXM 1.2 and Ret-XXM 2.0-expressing transgenic *N. tabacum* plants (line #2 for all). Error bars = s.e.m., n = 9, 10, 11 leaves from two batches of *N. tabacum*. One-way ANOVA followed by a Games-Howell post hoc test was performed for the significance analysis.

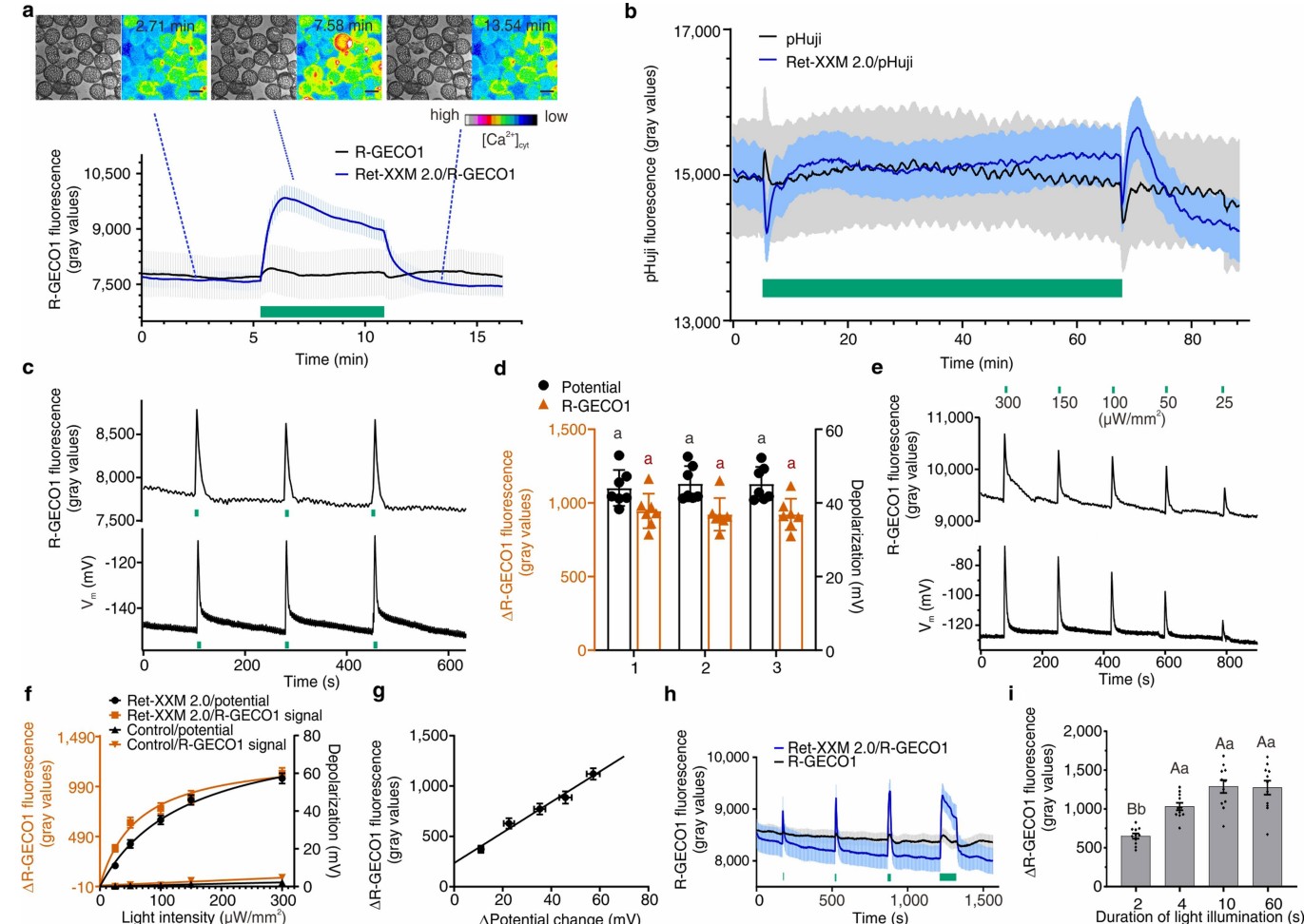

**Extended Data Fig. 3 | Variations in cytosolic pH and Ca²⁺ by XXM 2.0 stimulation. a**, R-GECO1 fluorescence in *N. benthamiana* mesophyll protoplasts following green light (532 nm, 180 µW mm⁻²) illumination. Error bars = s.e.m., n = 12 samples of protoplast from 6 *N. benthamiana* leaves transiently expressing R-GECO1. n = 14 samples of protoplast from 7 *N. benthamiana* leaves transiently co-expressing Ret-XXM 2.0 and R-GECO1. Two batches of *N. benthamiana* plants were used. Scale bar = 50 µm. **b**, Mean pHuji fluorescence in pHuji and Ret-XXM 2.0 with pHuji transgenic *N. tabacum* mesophyll cells following global green light (520 nm, 9 µW mm⁻²) illumination as indicated by the green bar. A decrease of pHuji fluorescence indicates an increase in cytosolic H⁺ concentration. Error bars = s.e.m., n = 10 for Ret-XXM 2.0 with pHuji transgenic *N. tabacum* plants and n = 6 for pHuji transgenic *N. tabacum* plants. Three batches of *N. tabacum* plants were used. Note the identical steady-state pHuji fluorescence values in pHuji and Ret-XXM 2.0 with pHuji transgenic *N. tabacum* mesophyll cells during global green light (520 nm, 9 µW mm⁻²) illumination. **c**, Simultaneous recording of cytosolic Ca²⁺ levels ([Ca²⁺]$_{cyt}$) (R-GECO1 fluorescence, top) and plasma membrane potential (V$_m$) by intracellular microelectrodes (bottom) in Ret-XXM 2.0 with R-GECO1 transgenic *N. tabacum* mesophyll cell. Three technical replicates of local green light (2 s, 532 nm, 100 µW mm⁻²) illumination were applied to stimulate XXM 2.0 as indicated by the green bars below the traces. **d**, Mean changes in [Ca²⁺]$_{cyt}$ (R-GECO1 fluorescence change, red) and V$_m$ changes (black) following 3 technical replicates of local green light (2 s, 532 nm, 100 µW mm⁻²)

irradiation. Error bars = s.e.m., n = 7 leaves from two batches of *N. tabacum* plants. One-way ANOVA followed by a Tukey post hoc test was performed for significance analysis. **e**, Representative traces of a simultaneous recording of [Ca²⁺]$_{cyt}$ levels (R-GECO1 fluorescence, top) and V$_m$ (bottom) in Ret-XXM 2.0 with R-GECO1 transgenic *N. tabacum* mesophyll cells following XXM 2.0 stimulation. Green light (2 s, 532 nm) with different light intensities (300, 150, 100, 50, 25 µW mm⁻²) was applied to stimulate XXM 2.0 as indicated by the green bars. **f**, Mean V$_m$ change and mean [Ca²⁺]$_{cyt}$ change (R-GECO1 fluorescence change) plotted against the green light intensities used in **e**. Error bars = s.e.m., n = 11 for R-GECO1 (Control in figure) and n = 15 for Ret-XXM 2.0 with R-GECO1 (Ret-XXM 2.0 in figure) transgenic plants. Three batches of *N. tabacum* plants were used. **g**, Relationship between V$_m$ change and R-GECO1 fluorescence change, fitted with a linear function (data obtained from **f**). Error bars = s.e.m., n = 15 plants from three batches of *N. tabacum* plants. **h**, R-GECO1 fluorescence recordings in R-GECO1 and Ret-XXM 2.0 with R-GECO1 transgenic *N. tabacum* mesophyll cells following green light (532 nm, 50 µW mm⁻²) treatment with different durations (2 s, 4 s, 10 s, 60 s). Error bars = s.e.m., n = 11 for R-GECO1 and Ret-XXM 2.0 with R-GECO1 transgenic plants. Two batches of *N. tabacum* plants were used. **i**, R-GECO1 fluorescence change of Ret-XXM 2.0 with R-GECO1 transgenic *N. tabacum* mesophyll cells shown in **h**. Error bars = s.e.m., n = 11 plants. Two batches of *N. tabacum* plants were used. One-way ANOVA followed by a Dunnett T3 post hoc test was performed for significance analysis.

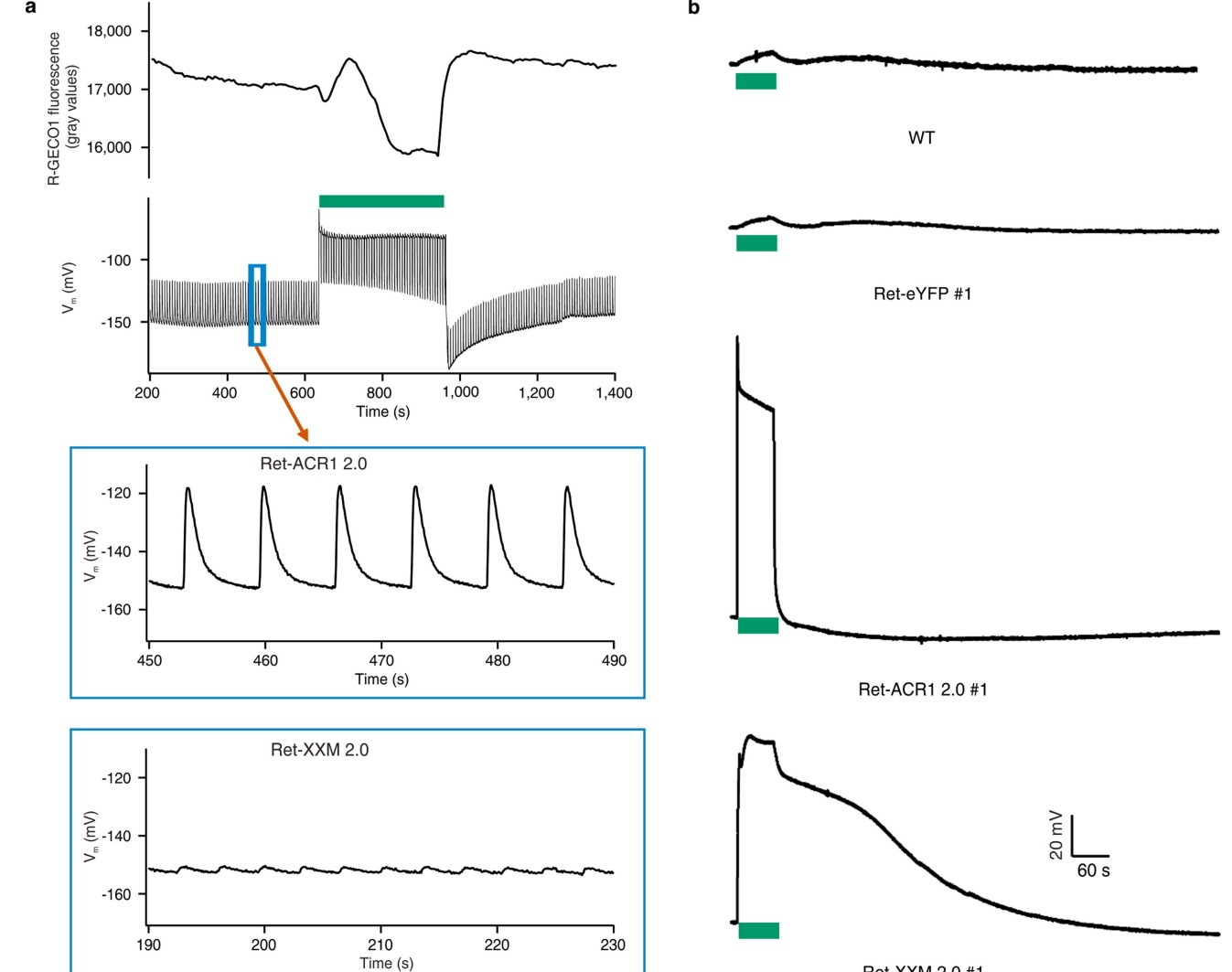

**Extended Data Fig. 4 | Plasma membrane potential and R-GECO1 fluorescence recording in *N. tabacum* plants. a**, Simultaneous recording of $[Ca^{2+}]_{cyt}$ (R-GECO1 fluorescence) and $V_m$ in Ret-ACR1 2.0 with R-GECO1 transgenic *N. tabacum* mesophyll cells, global green light (520 nm, 9 μW mm$^{-2}$) illumination was indicated by the green bar. Enlargement of the marked segment is shown in the blue box in the middle. A marked $V_m$ change (-36 mV) was induced in Ret-ACR1 2.0 with R-GECO1 expressing mesophyll cell by a 50 ms 570 nm excitation light pulse which was used for R-GECO1 Ca$^{2+}$-imaging. In contract, $V_m$ recordings in Ret-XXM 2.0 with R-GECO1 transgenic *N. tabacum*

mesophyll cells, shown in the lower blue box, does not induce considerable $V_m$ changes (-2 mV) by 50 ms 570 nm excitation light during the R-GECO1 fluorescence Ca$^{2+}$ imaging routine. Thus, it should be noted that combining Ca$^{2+}$-imaging and optogenetic activation of Ret-ACR1 2.0 suffers from spectral overlap due to the red-shifted absorption spectrum of ACR1, which is not the case for Ret-XXM 2.0. **b**, Representative mesophyll membrane voltage traces in leaves of WT and transgenic Ret-eYFP (line #1), Ret-ACR1 2.0 (line #1) and Ret-XXM 2.0 (line #1) *N. tabacum* plants, 1 min global green light illumination was indicated by the green bar.

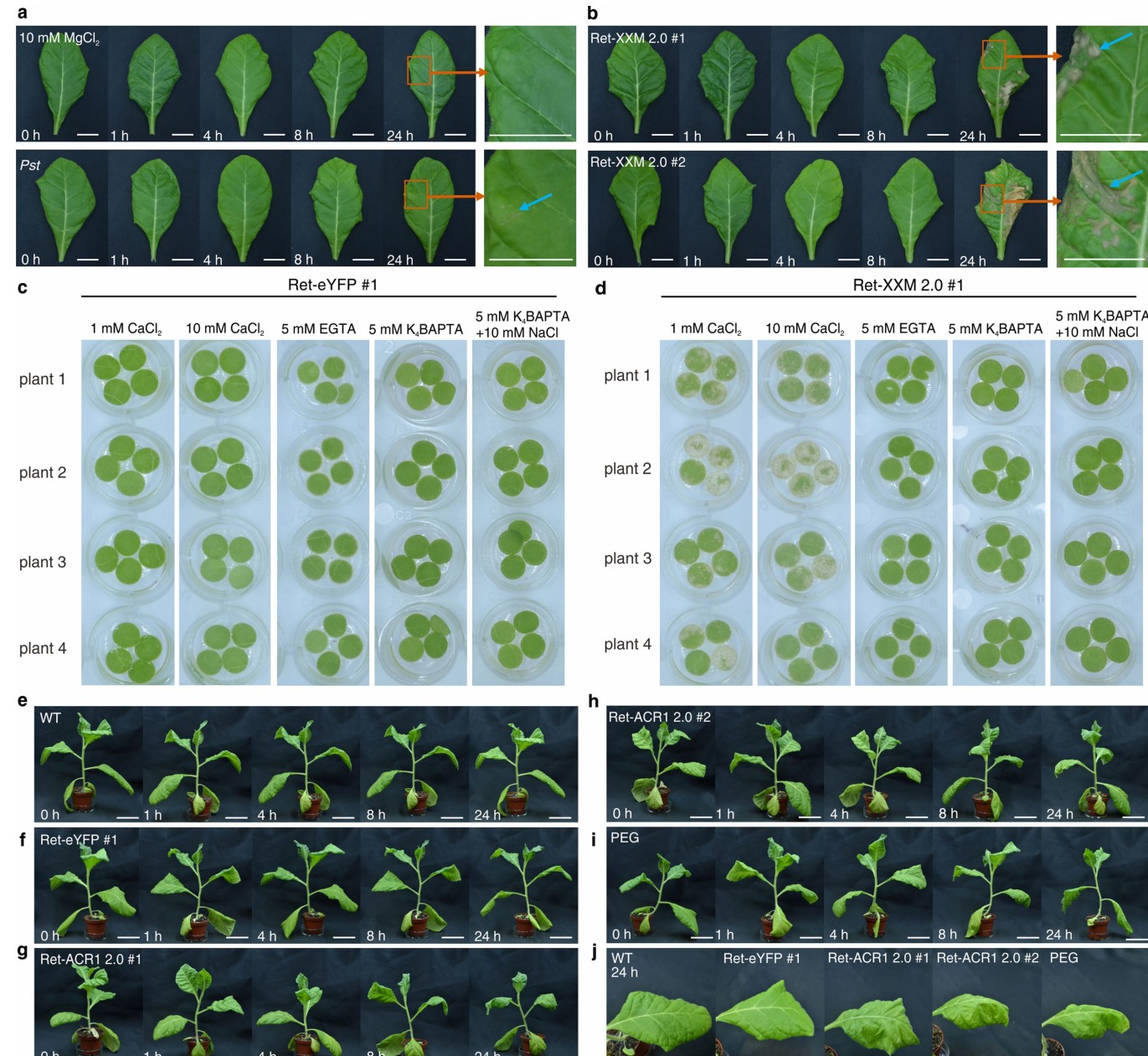

**Extended Data Fig. 5 | Phenotypes caused by ACR1 2.0 and XXM 2.0 activation or PEG and *Pst* treatment in transgenic *N. tabacum* plants.**
**a**, Phenotype of Ret-eYFP #1 transgenic *N. tabacum* plants at different time points after the treatment with 10 mM MgCl$_2$ or *Pst*. Scale bar = 3 cm. Enlarged images of leaf areas in the red boxes are presented in the right-most figures. The blue arrows point to necrotic leaf parts. Scale bar = 3 cm. n = 6 plants from two batches of *N. tabacum* plants. **b**, Phenotype of Ret-XXM 2.0 #1 or Ret-XXM 2.0 #2 transgenic *N. tabacum* lines at the indicated time points at the bottom when exposed to global continuous green light (520 nm, 9 μW mm$^{-2}$). Scale bar = 3 cm. Enlarged images of leaf areas in the red boxes are presented in the right-most figures. The blue arrows point to necrotic leaf parts. Scale bar = 3 cm. n = 6 plants from two batches of *N. tabacum* plants. **c, d**, Phenotypes of leaf discs in response to global continuous green light illumination. Leaf discs of (**c**) Ret-eYFP #1 and (**d**) Ret-XXM 2.0 #1 transgenic lines were floated either in 1 mM CaCl$_2$, 10 mM CaCl$_2$, 5 mM EGTA (pH 7.0, KOH), 5 mM K$_4$BAPTA or

5 mM K$_4$BAPTA + 10 mM NaCl solution in the dark for 1 h prior to the movement to global continuous green light condition. Pictures were captured after 24 h green light treatment. n = 7 leaves from two batches of *N. tabacum* plants. **e–h** Phenotype of (**e**) WT, (**f**) Ret-eYFP #1, (**g**) Ret-ACR1 2.0 #1, and (**h**) Ret-ACR1 2.0 #2 *N. tabacum* plants after different durations of constant global green light illumination, as indicated in the figures. Plants were grown in red light (650 nm, 30 μW mm$^{-2}$) condition for 45 days before additional green light (520 nm, 9 μW mm$^{-2}$) was applied at t = 0 h for the experimental group. Scale bar = 5 cm. n = 6 plants from two batches of *N. tabacum* plants. **i**, Phenotype of *N. tabacum* Ret-eYFP #1 control plants at different time points after watering with 35% PEG. Scale bar = 5 cm. n = 6 plants from two batches of *N. tabacum* plants. **j**, The fifth leaves of *N. tabacum* plants after global continuous green light or 35% PEG treatment. Photos were taken after 24 h treatment with green light or PEG. Scale bar = 5 cm. n = 6 plants from two batches of *N. tabacum* plants.

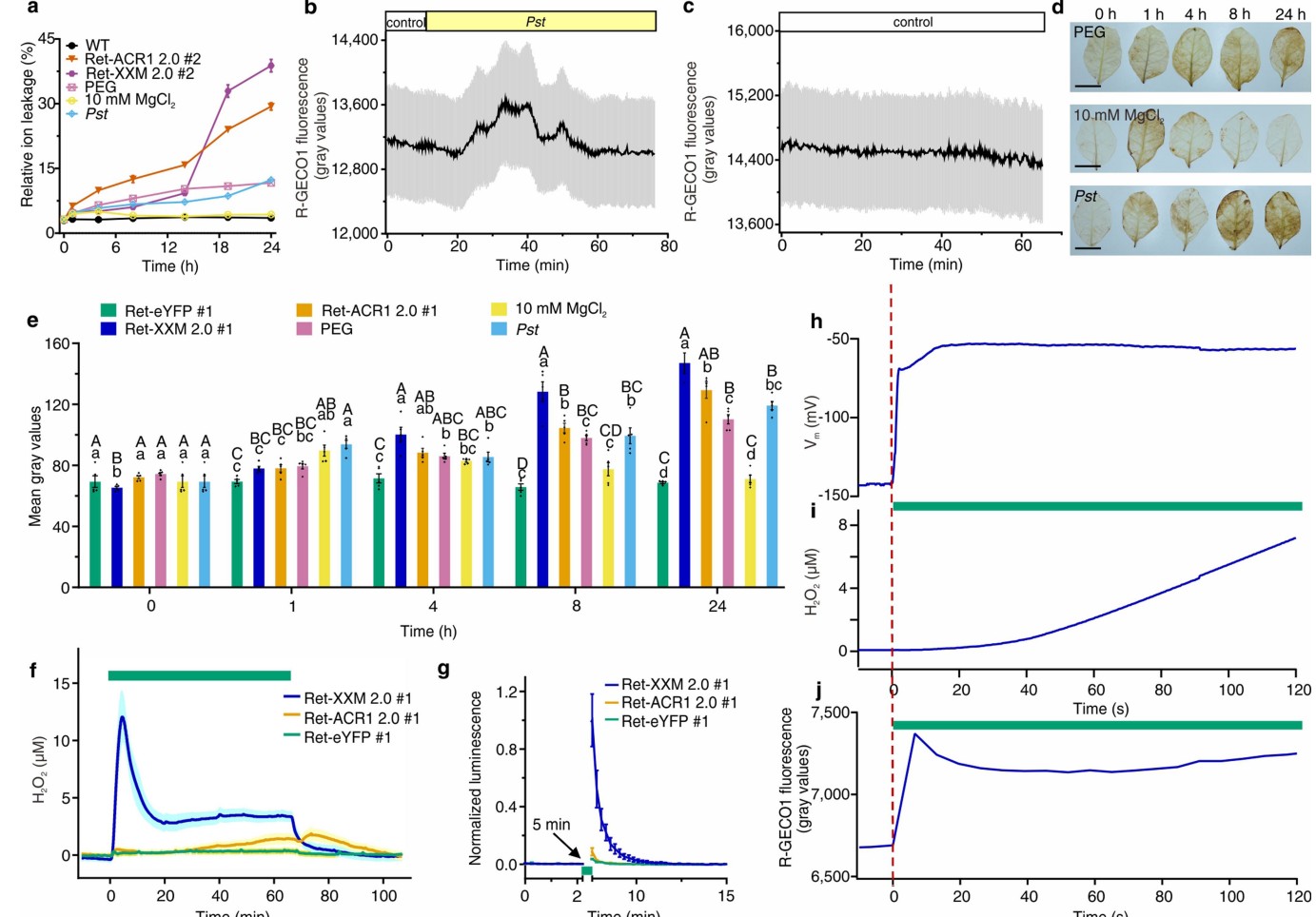

**Extended Data Fig. 6 | Ion leakage, voltage, $[Ca^{2+}]_{cyt}$ and ROS dynamics in *N. tabacum* leaves. a**, Relative ion leakage from *N. tabacum* leaf tissues at different time points after global green light (520 nm, 9 µW mm⁻²) illumination, spraying of 10 mM $MgCl_2$, *Pst*, or watering with 35% PEG. Error bars = s.e.m., n = 6 leaves from two batches of *N. tabacum* plants. **b**, **c**, R-GECO1 fluorescence change in R-GECO1 transgenic *N. tabacum* leaves in the (**b**) presence or (**c**) absence of *Pst* treatment, as indicated by the bar above the trace. Bath solution without *Pst* was used as the control solution (control). Error bars = s.e.m., n = 6 leaves from two batches of *N. tabacum* plants. **d**, ROS detection in *N. tabacum* leaves by diaminobenzidine staining. *N. tabacum* leaves were collected at different time points after 10 mM $MgCl_2$, *Pst*, or 35% PEG treatment. Scale bar = 5 cm. n = 5 leaves from two batches of *N. tabacum* plants. **e**, Mean staining intensities of whole leaves after the diaminobenzidine staining for ROS detection in **d** and Fig. 2e at the indicated time points. Error bars = s.e.m., n = 5 leaves from two batches of *N. tabacum* plants. Significance analysis was analyzed by One-way

ANOVA following a Dunnett T3 post hoc or Tukey post hoc test. **f**, Amperometric quantification of hydrogen peroxide ($H_2O_2$) dynamics in transgenic *N. tabacum* leaves, global green light illumination was indicated by the green bar. Error bars = s.e.m., n = 6 for Ret-XXM 2.0 (line #1), n = 14 for Ret-ACR1 2.0 (line #1) and n = 4 for Ret-eYFP (line #1) plants. Two batches of *N. tabacum* plants were used. **g**, ROS production following 5 min global green light illumination detected by luminol chemiluminescence assay. The luminescence was normalized to the mean value of Ret-XXM 2.0 samples. Error bars = s.e.m., n = 5 leaves from two batches of *N. tabacum* plants. **h**, **i**, Example of simultaneous (**h**) $V_m$ recording and (**i**) $H_2O_2$ recoding in Ret-XXM 2.0 transgenic *N. tabacum* leaves, global green light illumination was indicated by the green bar. **j**, R-GECO1 fluorescence in Ret-XXM 2.0 with R-GECO1-expressing *N. tabacum* mesophyll cells indicating cytosolic $Ca^{2+}$ levels in response to global green light illumination as indicated by the green bar.

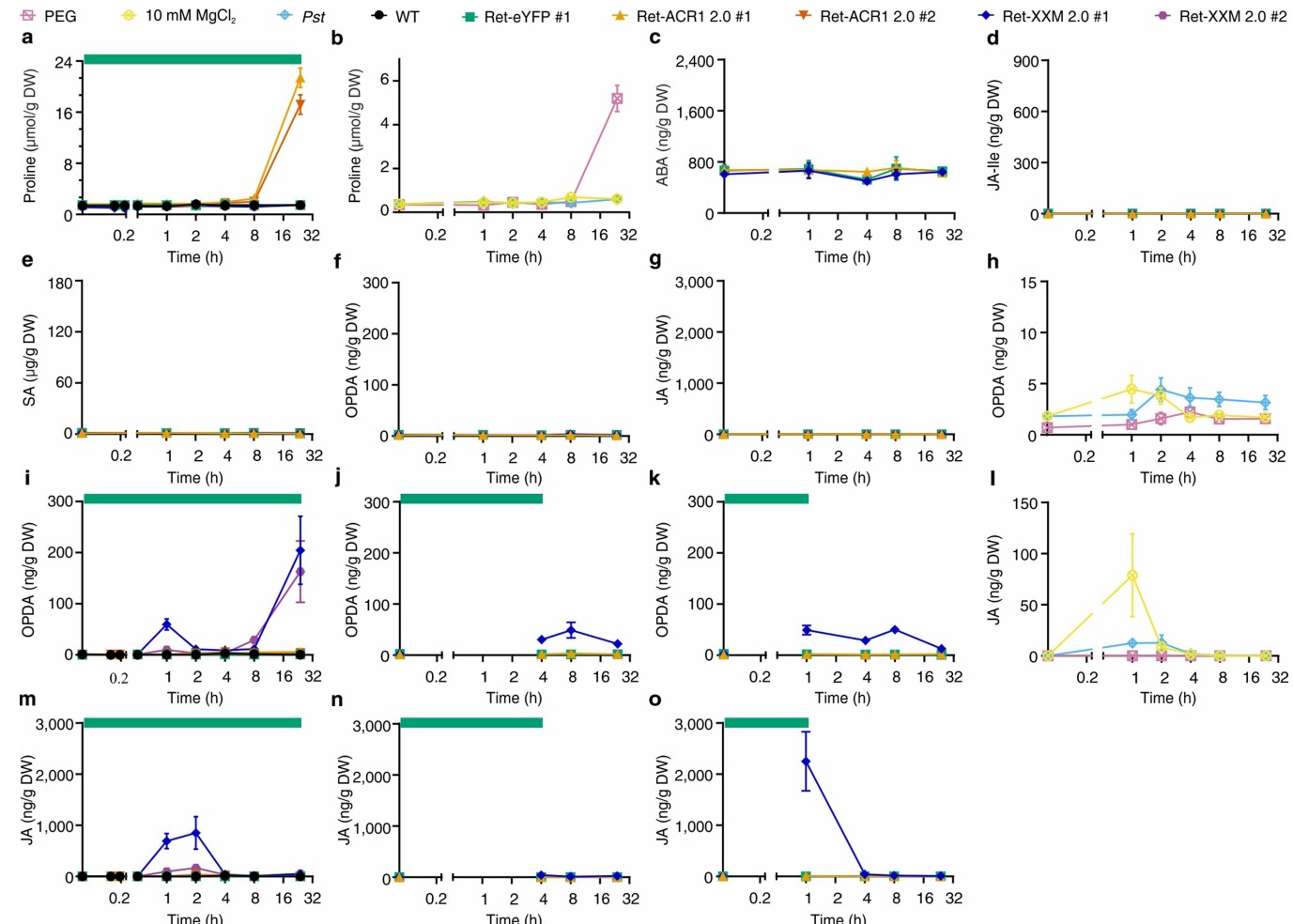

**Extended Data Fig. 7 | Distinct patterns of metabolites in *N. tabacum* plants following different treatments. a**, Proline content in WT and transgenic *N. tabacum* leaves during constant global green light (520 nm, 9 μW mm⁻²) illumination. Error bars = s.e.m., n = 4, 5 leaves from two batches of *N. tabacum* plants. **b**, Proline content in Ret-eYFP #1 transgenic *N. tabacum* leaves following watering 35% PEG as well as spraying 10 mM MgCl₂ or *Pst* at t = 0 h on the whole plants. Error bars = s.e.m., n = 5, 6 leaves from two batches of *N. tabacum* plants. **c**–**g**, Quantification of (**c**) ABA, (**d**) JA-Ile, (**e**) SA, (**f**) OPDA, and (**g**) JA in transgenic *N. tabacum* leaves in the absence (0 h) of green light treatment. Error bars = s.e.m., n = 5, 6 leaves from two batches of *N. tabacum* plants. **h**, Quantification of OPDA in Ret-eYFP #1 transgenic *N. tabacum* leaves at different time points after the treatment with 35% PEG, spray-inoculation with *Pst* or 10 mM MgCl₂ as control. Error bars = s.e.m., n = 5, 6 leaves from two

batches of *N. tabacum* plants. **i**–**k**, OPDA content in WT or transgenic *N. tabacum* leaves at different time points with (**i**) constant, (**j**) 4 h or (**k**) 1 h global green light illumination. Error bars = s.e.m., n = 4, 5, 6, 7 leaves from two batches of *N. tabacum* plants. **l**, JA in leaves of Ret-eYFP #1 upon watering with 35% PEG as well as spraying 10 mM MgCl₂ or *Pst*. Error bars = s.e.m., n = 5, 6 leaves from two batches of *N. tabacum* plants. **m**–**o**, JA content in transgenic *N. tabacum* leaves with (**m**) constant, (**n**) 4 h, (**o**) 1 h global green light illumination. Error bars = s.e.m., n = 4, 5, 6, 7 leaves from two batches of *N. tabacum* plants. *N. tabacum* plants were grown in red light (650 nm, 30 μW mm⁻²) for 45 days and additional constant global green light (520 nm, 9 μW mm⁻²) was applied from t = 0 h. The exact sample size (n) for each experimental group was listed in Supplementary Table 1.

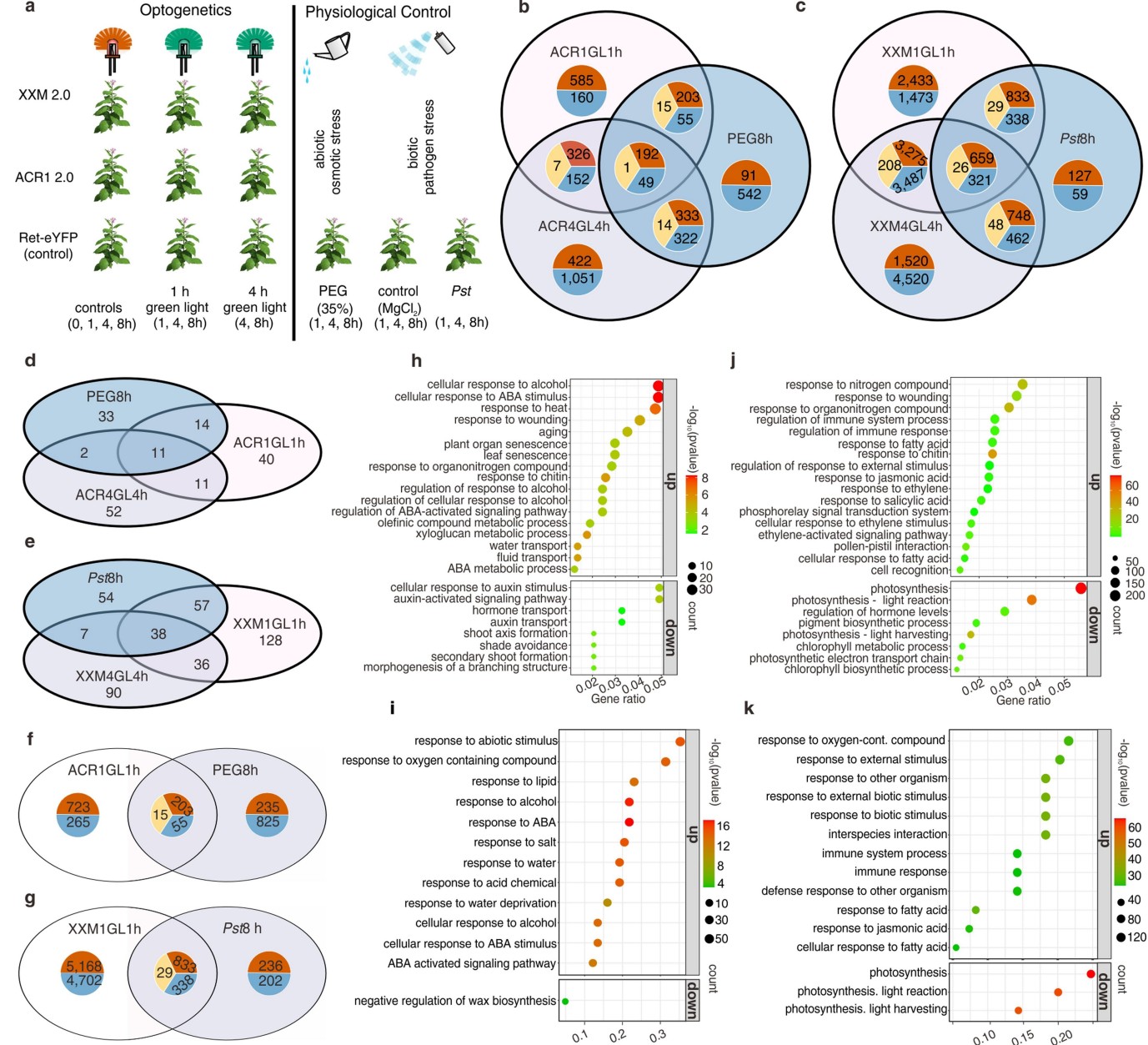

**Extended Data Fig. 8 | Venn diagram and gene ontology analysis showing the overlap of DEGs following different stimulation. a**, Illustration of the experimental design for the transcriptomic analysis in Ret-XXM 2.0 #1, Ret-ACR1 2.0 #1 and Ret-eYFP #1 control plants subjected to red light only, or illumination with an additional 1 or 4 h of green light (520 nm, 9 μW mm⁻²) followed by light shut-off. As physiological control to the ACR1 2.0 or XXM 2.0 light stimulation, Ret-eYFP #1 control plants were watered with 35% PEG or sprayed with *Pst*, respectively. As a negative control experiment to the *Pst* spraying treatment, Ret-eYFP #1 control plants were sprayed with buffer solution (10 mM MgCl₂) at t = 0 h. n = 3 plants from two batches of *N. tabacum* plants were collected at 0, 1, 4 and 8 h. **b, c**, Venn diagrams showing the overlap of (**b**) DEGs (differentially expressed genes) from Ret-eYFP #1 plants 8 h after PEG-treatment (PEG8h) with those from 1 or 4 h global green light treated Ret-ACR1 2.0 #1 plants (ACR1GL1h, ACR4GL4h) or (**c**) DEGs from Ret-eYFP #1 plants 8 h after *Pst* treatment (*Pst*8h) with DEGs from 1 or 4 h global green light treated Ret-XXM 2.0 #1 plants (XXM1GL1h, XXM4GL4h). Numbers highlighted in red represent up-regulated DEGs, numbers highlighted in blue denote down-regulated DEGs, and those highlighted in yellow exhibit inverse regulation. **d, e**, Venn diagrams showing the overlap of (**d**) gene ontology (GO) terms from

PEG8h, ACR1GL1h and ACR4GL4h or (**e**) GO terms from *Pst*8h with XXM1GL1h and XXM4GL4h. **f, g**, Venn diagrams showing the overlap of (**f**) DEGs from ACR1GL1h with those from PEG8h or (**g**) DEGs from XXM1GL1h with those from *Pst*8h. Numbers highlighted in red represent up-regulated DEGs, numbers highlighted in blue denote down-regulated DEGs, and those highlighted in yellow exhibit inverse regulation. **h**, Bubble plot depicting prevalent significantly enriched GO terms of ACR1GL1h. *P*-values were indicated by color. The size of the bubble plot represents the numbers of DEGs mapped in the represented GO terms. **i**, Bubble plots of prevalent enriched GO terms shared between ACR1GL1h and PEG8h. *P*-values were indicated by color. The size of the bubble plot represents the numbers of DEGs mapped in the represented GO terms. **j**, Bubble plot depicting prevalent significantly enriched GO terms of XXM1GL1h plants. *P*-values were indicated by color. The size of the bubble plot represents the numbers of DEGs mapped in the represented GO terms. **k**, Bubble plots of prevalent enriched GO terms shared between XXM1GL1h and *Pst*8h. *P*-values were indicated by color. The size of the bubble plot represents the numbers of DEGs mapped in the represented GO terms. The image of the tobacco plant in **a** is adapted from TurboSquid.

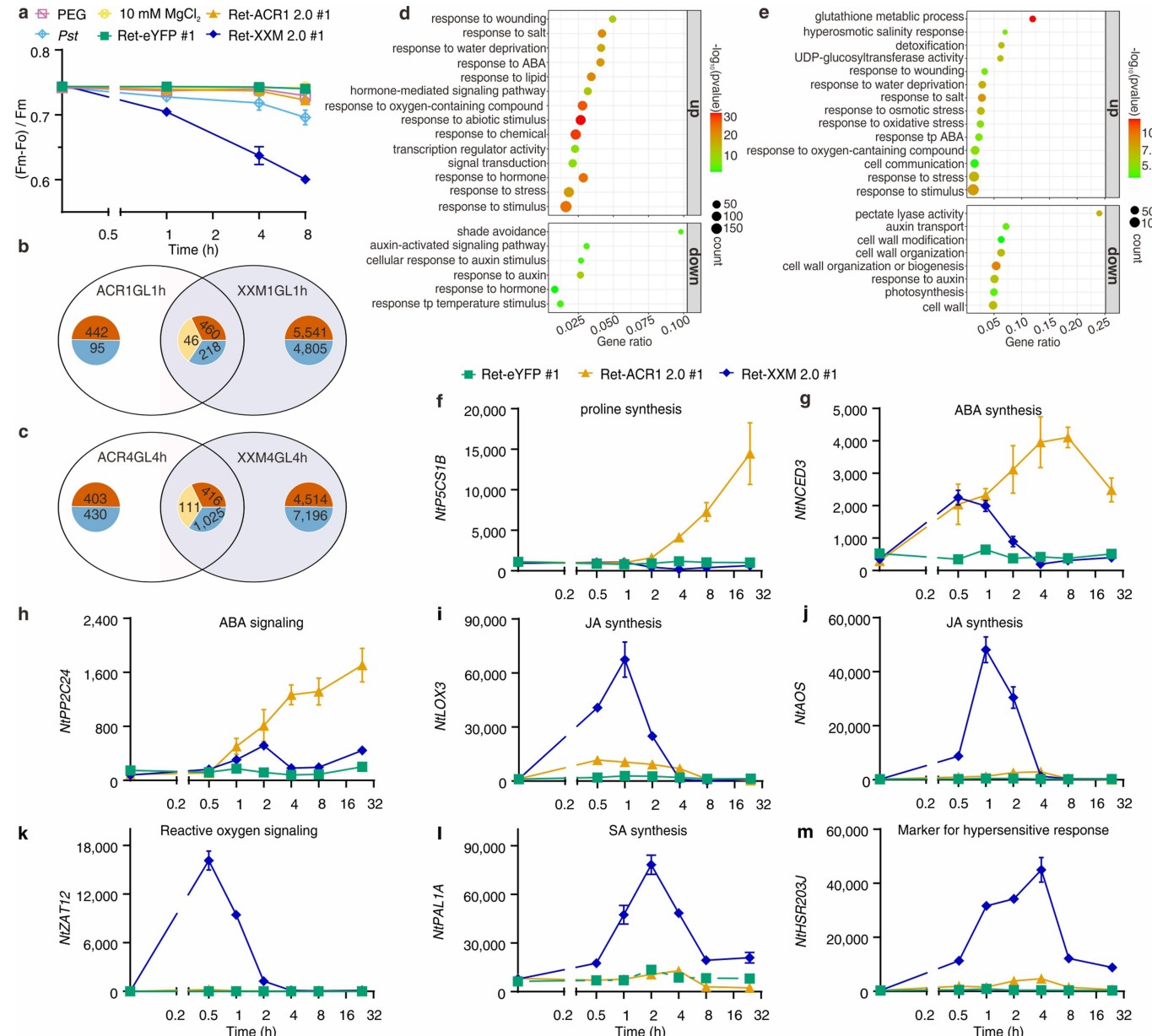

**Extended Data Fig. 9 | Distinct transcription profiles and impaired photosynthesis induced by different treatments. a**, Maximum quantum yield of photosystem II energy conversion in Ret-eYFP #1, Ret-ACR1 2.0 #1 and Ret-XXM 2.0 #1 *N. tabacum* leaves following global green light (520 nm, 9 µW mm$^{-2}$) treatment or in Ret-eYFP #1 leaves when plants were watered with 35% PEG or leaves were sprayed with 10 mM MgCl$_2$ or *Pst*. Error bars = s.e.m., n = 5, 6, 7, 9 leaves from two batches of *N. tabacum* plants. *N. tabacum* plants were grown in red light (650 nm, 30 µW mm$^{-2}$) for 45 days and additional global green light, 35% PEG, 10 mM MgCl$_2$ or *Pst* treatment was applied from t = 0 h. **b, c**, Venn diagrams showing the (**b**) overlap of DEGs from ACR1GL1h and XXM1GL1h or (**c**) overlap between DEGs from ACR4GL4h and XXM4GL4h. Numbers highlighted in red represent up-regulated DEGs, numbers highlighted in blue denote down-regulated DEGs, and those highlighted in yellow exhibit inverse regulation. **d, e**, Bubble plots of prevalent enriched GO terms (**d**) shared

between ACR1GL1h and XXM1GL1h or (**e**) shared between ACR4GL4h and XXM4GL4h. *P*-values were indicated by color. The size of the bubble plot represents the numbers of DEGs mapped in the represented GO terms. **f-m**, Relative expression levels of (**f**) stress related genes *NtP5CS1B*, (**g**) ABA synthesis gene *NtNCED3*, (**h**) ABA signaling pathway gene *NtPP2C24*, (**i**) JA synthesis genes *NtLOX3*, (**j**) *NtAOS*, (**k**) the specific ROS-controlled transcription factor *NtZAT12*, (**l**) SA synthesis gene *NtPAL1A* and (**m**) the hypersensitive maker *HSR203J* for Ret-eYFP (line #1), Ret-ACR1 2.0 (line #1) and Ret-XXM 2.0 (line #1) transgenic *N. tabacum* leaves after the indicated durations of global green light illumination. The transcript numbers of genes were normalized to 10,000 molecules of actin. Error bars = s.e.m., n = 4, 5 plants from two batches of *N. tabacum* plants. The exact sample size (n) for each experimental group was listed in Supplementary Table 1.

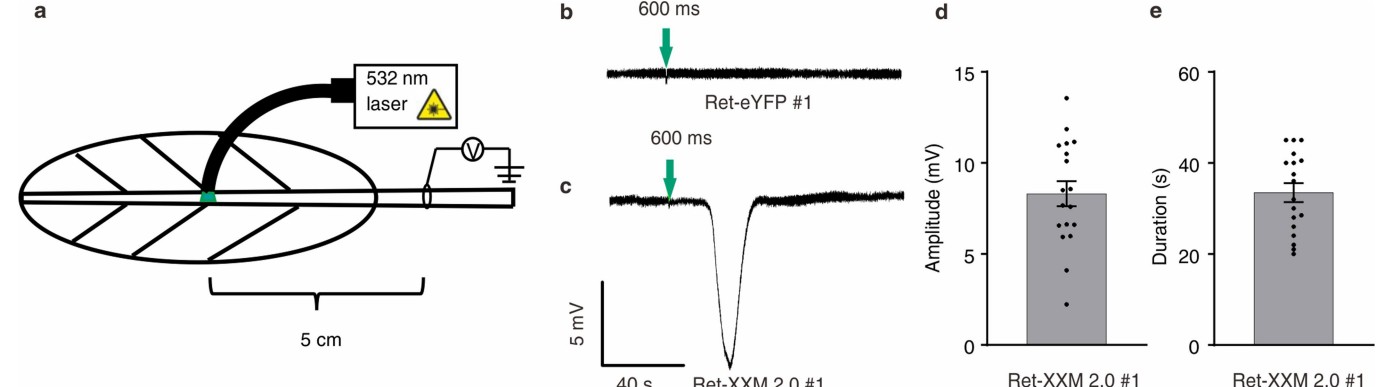

**Extended Data Fig. 10 | Local green light stimulation triggers a long distance electrical signal traveling in XXM plants. a**, Diagram of the experimental design for detecting long-distance traveling electrical signal induced by local green light (532 nm, 5.3 mW mm⁻²) illumination on Ret-eYFP #1 or Ret-XXM 2.0 #1 transgenic *N. tabacum* leaves. Green light illumination, indicated by the green spot, was applied through a black covered optical fiber directly on the main vein of the leaves. Surface potential was recorded at a distance of 5 cm from the local illumination site. **b**, **c**, Surface potential recordings of electrical waves in (**b**) Ret-eYFP #1 or (**c**) Ret-XXM 2.0 #1 transgenic *N. tabacum* leaves by applying 600 ms local green light illumination. Green arrows indicate the time when the 600 ms green light illumination applied. During green light treatment, no obvious electrical signal was detectable at the unilluminated region (see the electrical trace where the green arrow pointed). A depolarization event at a distance of 5 cm from the illuminated region was only detected in Ret-XXM 2.0 #1 transgenic *N. tabacum* about 30 s after green light treatment. **d**, **e**, The mean values of (**d**) amplitude and (**e**) duration of electrical waves triggered in Ret-XXM 2.0 #1 transgenic *N. tabacum* plants. Error bars = s.e.m., n = 18 leaves from 4 batches of *N. tabacum* plants.

# Reporting Summary

## Statistics

For all statistical analyses, confirm that the following items are present in the figure legend, table legend, main text, or Methods section.

| n/a | Confirmed | |
|---|---|---|
| ☐ | ☒ | The exact sample size (*n*) for each experimental group/condition, given as a discrete number and unit of measurement |
| ☐ | ☒ | A statement on whether measurements were taken from distinct samples or whether the same sample was measured repeatedly |
| ☐ | ☒ | The statistical test(s) used AND whether they are one- or two-sided *Only common tests should be described solely by name; describe more complex techniques in the Methods section.* |
| ☐ | ☒ | A description of all covariates tested |
| ☐ | ☒ | A description of any assumptions or corrections, such as tests of normality and adjustment for multiple comparisons |
| ☐ | ☒ | A full description of the statistical parameters including central tendency (e.g. means) or other basic estimates (e.g. regression coefficient) AND variation (e.g. standard deviation) or associated estimates of uncertainty (e.g. confidence intervals) |
| ☐ | ☒ | For null hypothesis testing, the test statistic (e.g. *F*, *t*, *r*) with confidence intervals, effect sizes, degrees of freedom and *P* value noted *Give P values as exact values whenever suitable.* |
| ☒ | ☐ | For Bayesian analysis, information on the choice of priors and Markov chain Monte Carlo settings |
| ☒ | ☐ | For hierarchical and complex designs, identification of the appropriate level for tests and full reporting of outcomes |
| ☒ | ☐ | Estimates of effect sizes (e.g. Cohen's *d*, Pearson's *r*), indicating how they were calculated |

*Our web collection on statistics for biologists contains articles on many of the points above.*

## Software and code

Policy information about availability of computer code

Data collection: All data are collected with commercially available equipment and software. e.g. WinWCP V5.3.4 (University of Strathclyde, UK) for voltage or current recording; Leica LAS AF Version 2.7.3.9723 (Leica Microsystems, Wetzlar, Germany) for confocal images; VisiView Version 2.1.1 Software (Visitron Systems GmbH) for all-optical physiology experiments and Ca2+ imaging; The in vivo amperometric measurement of the production of H2O2 was performed by Patch-Master software V2x90 (HEKA) for ROS detection; Luminescence meaurements for ROS quantification were detected by Skanlt software Version 6.1; Analyst version 1.6.3 and MultiQuant Version 3.0.2 software from Sciex were used for mass spec detection of hormones and metabolites; Aequorin luminescence was detected by a photomultiplier (Photo Counting Module MP 1983 RS CPM, Perkin Elmer) controlled by IGI-MPRS232 (IGIsystems) and Labview V14.0.0 (National Instruments) were used in this measurement; Proline content was quantified by a spectrophotometer (lab/ Hitachi U-1500); Eppendorf Mastercycler ep realplex Version 2.2 software for qRT-PCR measurements; Quantification of photosynthetic performance was done with a Maxi PAM fluorometer controlled by the IMAGING WIN v2.41a FW MULTI RGB software (Walz, Effeltrich, Germany); The ChR2 structure model was generated with the standard molecular viewer PyMOL and the internal cavities were calculated using Hollow suit, which is a published Python script; In the framework of the transcriptomic analysis, transcript mapping to the tobacco genome was peformed with the Amalgkit software pipeline (https://github.com/kfuku52/amalgkit), for gene annotation the Dicots PLAZA 5.0 repository was used; Differential gene expression analysis and subsequent gene ontology analysis and heat map presentations was performed with the DIANE package, Venn diagrams were generated with the GOVenn script of the GOPlot package, GO analysis on Venn subsets was carried out with the help of gprofiler2, all R-packages used are described in detail within the Materials and Methods section.

| Data analysis | Data analysis are all processed by commercially available software. e.g. ImageJ-win 64/FIJI for image analysis; Videos were generated by Adobe Premiere Pro CC 2015 with H.264 encoding. Origin Pro 2019 and GraphPad Prism v8.4.2 for graphs ; Excel (Office Professional) for source data handling; Student's t-test or One-way analysis of Variance (ANOVA) were used to analyze significance using IBM SPSS statistics (version 26.0). |

For manuscripts utilizing custom algorithms or software that are central to the research but not yet described in published literature, software must be made available to editors and reviewers. We strongly encourage code deposition in a community repository (e.g. GitHub). See the Nature Portfolio guidelines for submitting code & software for further information.

## Data

Policy information about availability of data

All manuscripts must include a data availability statement. This statement should provide the following information, where applicable:
- Accession codes, unique identifiers, or web links for publicly available datasets
- A description of any restrictions on data availability
- For clinical datasets or third party data, please ensure that the statement adheres to our policy

All the data generated in this study are available in the paper and the Supplementary Information. Source data behind graphs are provided with this paper. The RNA-seq data have been deposited in the National Center for Biotechnology Information (NCBI) database (Bioproject ID: PRJNA1108451).

## Research involving human participants, their data, or biological material

Policy information about studies with human participants or human data. See also policy information about sex, gender (identity/presentation), and sexual orientation and race, ethnicity and racism.

| Reporting on sex and gender | NA |
| Reporting on race, ethnicity, or other socially relevant groupings | NA |
| Population characteristics | NA |
| Recruitment | NA |
| Ethics oversight | NA |

Note that full information on the approval of the study protocol must also be provided in the manuscript.

## Field-specific reporting

Please select the one below that is the best fit for your research. If you are not sure, read the appropriate sections before making your selection.

☒ Life sciences    ☐ Behavioural & social sciences    ☐ Ecological, evolutionary & environmental sciences

For a reference copy of the document with all sections, see nature.com/documents/nr-reporting-summary-flat.pdf

## Life sciences study design

All studies must disclose on these points even when the disclosure is negative.

| Sample size | No sample sizes were predetermined by an statistical method. The sizes of samples were determined depended on experimental trials to allow for confident statistical tests. For each experiment, in Xenopus laevis oocytes we have at least 3 biological samples. For plants experiments, we included 2 transgene lines and more than 4-5 biological repeats for each. For transcriptomics analysis, 3 biological replicates were used. At least two times of biological repetition were done for each experiment. |
| Data exclusions | We excluded some electrical measurement due to the losing of electrical signals (the electrodes were not in the cell anymore or the cells were dead) during measurement. We also excluded some Ca2+ or pH measurement due to the losing of focus of cells during measurement. For the images of leaves, plants and oocytes, we have made many biological repeats with similar results, but did not show them all due to space limitations and the sake of clarity. |
| Replication | All the experimental replications were very successful and clear. The numbers of biological replicates were performed with more than 2 independent batches of plants or Xenopus laevis oocytes. The repeat numbers are listed in the legend part. |
| Randomization | For global green light treatment, all tobacco plants of different genotypes were marked on pots and randomly placed in the light controlled growth chamber. In order to verify the reliability and reproducibility of results, we always choose the materials in experiments randomly. If allocation was needed, the grouped samples were randomly allocated for each experiment. For instance, to detect all trans retinal production in the transgene tobacco lines, we took 10 leaves of different plants randomly from each line for the detection. In the metabolites measurement or transcriptome analysis, we also collected the same age leaves randomly from independent plants of different lines. For |

Xenopus laevis oocytes experiments, we picked out the healthy oocytes first and separated them randomly for each expression vectors. We always use oocytes from the same supplier in one comparison experiment.

| Blinding | In this study, there was no blinding test, because the differences among different experimental groups were extremely obvious. In addition to that all the experiment showed very good replication with enough biological repeats. |

# Reporting for specific materials, systems and methods

We require information from authors about some types of materials, experimental systems and methods used in many studies. Here, indicate whether each material, system or method listed is relevant to your study. If you are not sure if a list item applies to your research, read the appropriate section before selecting a response.

## Materials & experimental systems

| n/a | Involved in the study |
|-----|-----------------------|
| ☒ | Antibodies |
| ☒ | Eukaryotic cell lines |
| ☒ | Palaeontology and archaeology |
| ☐ | ☒ Animals and other organisms |
| ☒ | Clinical data |
| ☒ | Dual use research of concern |
| ☒ | Plants |

## Methods

| n/a | Involved in the study |
|-----|-----------------------|
| ☒ | ChIP-seq |
| ☒ | Flow cytometry |
| ☒ | MRI-based neuroimaging |

## Animals and other research organisms

Policy information about studies involving animals; ARRIVE guidelines recommended for reporting animal research, and Sex and Gender in Research

| Laboratory animals | Oocytes of Xenopus laevis frogs bought from EcoCyte Bioscience (Dortmund Germany) or obtained from frogs in the Julius-von-Sachs-Institute. |
|---|---|
| Wild animals | The study did not involve wild animals. |
| Reporting on sex | This information has not been collected. We use oocyte cells and sex was not considered in study design. |
| Field-collected samples | The study did not involve samples from the field. |
| Ethics oversight | Xenopus laevis oocytes were either bought from EcoCyte Bioscience (Dortmund Germany) or obtained from frogs in the Julius-von-Sachs-Institute, Würzburg University. The laparotomy to obtain oocytes from Xenopus laevis was carried out in accordance with the principles of the Basel Declaration and recommendations of Landratsamt Wuerzburg Veterinaeramt. The protocol under License #70/14 from Landratsamt Wuerzburg, Veterinaeramt, was approved by the responsible veterinarian. |

Note that full information on the approval of the study protocol must also be provided in the manuscript.

## Plants

| Seed stocks | N. tabacum transgeic lines were generated in Molecular Plant Physiology and Biophysics, Julius-von-Sachs-Institute, University of Wuerzburg, 97082 Wuerzburg, Germany. UBQ10_Ret_ACR1 2.0 #1 and #2 , UBQ10_Ret_XXM 1.2 #1 and #2, UBQ10_Ret_XXM 2.0 #1 and #2, UBQ10_Ret_eYFP #1 and #2, UBQ10_ Ret-XXM 2.0/35s_NES_2*R-GECO1 #1, UBQ10_ Ret-XXM 2.0/35s_NES_2*pHuji #1. |
|---|---|
| Novel plant genotypes | N. tabacum transgeic lines were generated by bath the leaf discs in MS medium containing A. tumefaciens strain GV3101 harboring the pCAMBIA3300 vector with BASTA resistance or pCAMBIA1300 vector with hygromycin resistance. Explants and calli were generated on the Medium and the explants were moves to rooting medium. Finally got the trangene plants and seeds. At least 2 independent lines were selected to verify the genotypes. |
| Authentication | The transgenic plants were verified by eYFP, pHuji or R-GECO1 fluorescence in leaves. Seeds of individual plant were collected and selected for BASTA or hygromycin resistance using selection medium. PCR using gDNA as the template was applied to confirm the transgenic lines. The growth phenotype, biomass, photosynthesis parameters and the leaf carotenoid were compared between the transgene lines and wild type plants to check the influence. |

