## [Peer Review File · Nature]

Manuscript Title: Probing plant signal processing optogenetically by two channelrhodopsins

Editorial Notes:

Redactions – unpublished data

Reviewer Comments & Author Rebuttals

Reviewer Reports on the Initial Version:

Referees' comments:

Referee #1 (Remarks to the Author):

The manuscript of Ding et al describes the introduction of optogenetic tools in plants to manipulate membrane polarization in general, and the flow of Ca⁺⁺ ions in particular, towards unravelling the role played by electrical signals in the triggering of different physiological responses/programmes upon environmental cues. For this, they employed two different light-gated ion channels: a) ACR1, an anion channelrhodopsin previously introduced into plants by some of the authors of this article, and b) XXM, a further engineered channelrhodopsin with high Ca⁺⁺ permeability (previously used in animal cells, published by some of the authors of this article). They also employed the plants engineered previously for production of the necessary cofactor, retinal. They performed a functional characterization of the tools in transiently transformed *Nicotiana benthamiana* and transgenic *Nicotiana tabacum* leaves. This includes determination of membrane potential changes, cellular Ca⁺⁺ level changes using the R-GECO1 sensor, ROS production. They analysed certain particular metabolic markers of plant stress responses (e.g. hormones) and the transcription of a few genes related to some of the metabolic and signalling pathways mediated by those particular metabolites (plant-pathogen interaction, drought stress). The article deals with the introduction of technologies, that despite being already available in other fields/biological systems and having helped advancing our understanding of complex cellular processes, are unfortunately not yet being used broadly in plant research. Therefore, this kind of approaches are very much necessary and welcome! The experimental strategy used by the authors is very clever and potentially useful, namely to perform a targeted manipulation at a particular point/level within the series of events leading to the activation of a signalling cascade thereby trying to determine the relative contribution of the components involved in the resulting physiological response observed. The introduction of the (functional!) tools and the proof of principle example of the potential relevance for the field that it represents being able to trigger of cellular responses upon targeted manipulation of membrane potential and fluxes of specific signalling ions (Ca⁺⁺), both known to be involved in the plant responses to environmental cues, are the main strengths of the article and represent the main point and message of the paper. However, this reviewer considers that the article falls certainly short in two main aspects, and the central claims are not backed up with sufficient data which precludes assessing its generalised/straightforward usability:

Firstly, in order to make the approach really available for use by the community a much more complete molecular and functional characterization of the optogenetic systems is needed. A non-limiting list includes as is state-of-the-art and essential, i) for the tools tunability and quantitative control by using pulsing schemes, different intensity and time dose responses, reversibility/frequency of ion waves,

dynamic ranges obtained, ii) for the plants: cell-cell and plant-plant variability/reproducibility, full phenotypic analysis and effects on membranes, major metabolic and physiological parameters (with and without activation) – the current work just shows some representative pictures of the plants that do not allow a proper evaluation, and just dry weight data. This is still ok if just showing that the tools work, but it is critical to have it available if the plants are to be used to study biological processes.

Secondly, several/most of the key advantages of using light as an inducer and having a tool that permits targeted manipulation at a high spatiotemporal resolution and with reversibility & quantitative control are not explored at all in the work. These are all aspects that optogenetics enabled in the several biological systems in which it is revolutionising the way of studying complex cellular networks. For instance, one could generate waves of activation, compare different time-lengths of activation vs stronger activation, etc. Besides this, the experiments are bulk. Without the need to generate any biological material single cell activation can be easily performed and the set up/activity analysed.

Finally, a main concern is about the claims drawn in terms of applicability of the system, raised in the abstract, and text/discussion. Namely that with the experiments performed using the tools it is possible to determine/discriminate the differential contribution of the different electrochemical and ionic signals elicited in terms of a specific response either biotic or abiotic. This is not justified nor backed up with the experimental design used and data obtained. Plant-pathogen interactions, wounding and abiotic environmental stresses such as drought elicit a broad range of metabolic and physiological responses, are perceived by the plants via different dedicated receptors or simply produce cell mechanical damage, and activate a complex mesh of signalling pathways integrating the exogenous cues with the genetic program of the cells/tissues. Dozens, hundreds of regulatory networks are involved, with partially overlapping, redundant, complementary or opposing functions among different stresses, are tissue and developmental stage specific, and mediate the signal relay via hundreds of metabolites including hormones and other small molecules. In the present study a very limited set of metabolites were chosen and were analysed, and similarly only a couple of targeted and limiting selection of target genes studied. The elements are per se interesting and involved in some of the pathways, but so are hundreds of others, the choice is at some point arbitrary, as other (very) important components are not included. Just particular features are picked up that are not sufficient and necessarily representative a of the processes involved and of the global changes produced. Moreover, the experiments do not include the counterparts including the natural environmental hardships to whom they are compared. To be able to demonstrate the applicability of the tool to study those processes and be able to yield useful data, and given the complexity of the systems under study, the obvious experimental strategy has to comprise performing a broad metabolomics and transcriptomics profiling of the plant tissue with and without optogenetic activation of both channels, together with the use of standard plant-pathogen interactions and drought stresses. The methods for this are currently state of the art, trivial (and readily available, standard). This would provide a picture of the specificity of action of the pathways involved, crosstalk between the two type of signals and the relevance of those signals in the framework of a wide range of other components simultaneously activated during such environmental challenges. Moreover, there is a quantitative and time-resolved aspect involved in the activation of different pathways during the stress conditions which can only be understood if a whole network analysis is performed. Optogenetic approaches have demonstrated to be optimal to perform such targeted manipulations at highest quantitative control capabilities (waves of activation, compare different time-lengths of activation vs

stronger activation, etc all factors involved in plant-environment interactions), and such experimental means were not used here. This would inform on the applicability of the tools and if successful demonstrate the full potential of the strategy to the plant community, most willing to achieve such kind of pressing analysis.

Minor points:

why was it in the experiments in *Xenopus* oocytes blue light used? The activation/absorption/excitation spectra of the tool(s) should be included as well.

It is not clear in several experiments what the “n” refers to: samples from one leaf, different leaves of one plant, different plants, independent lines, etc? I haven't found the info in the Summary report neither.

Legend to figure 2 is not clear

A side by side comparison is not available for various experiments and under the same conditions, for instance Ca^{++} measurement with the ACR1 line. This would be very useful to analyse the selectivity/specificity.

In the figure 3d) what is the meaning of the peak of OPDA after one hour?

figure 3a, magnifications and contrast of the different panels are not consistent

Referee #2 (Remarks to the Author):

Authors genetically engineered a channelrhodopsin variant with high Ca²⁺ conductance, named XXM 2.0, to trigger cytosolic Ca²⁺ elevations in plant cells. Light-induced Ca²⁺ influx was studied in comparison with the light-gated anion channelrhodopsin ACR1 that causes anion efflux-dependent depolarization. While a XXM-induced Ca²⁺ signal caused plants to initiate ROS production and defense mechanism, anion efflux caused drought-stress responses. These findings imply that Ca²⁺ signals and anion efflux serve as elicitors for distinct responses.

Optogenetic approach for channel manipulation has been well documented in animal systems but much less in plant cells. This study, together with previously published works from the same authors, indicates that channelrhodopsins provide promising tools for altering membrane potential and ionic transport activities in plant cells. As far as I am aware, the use of anion efflux ACR1 in plants was previously published, whereas Ca-permeable XXM was used here in plants for the first time. The XXM variants can be particularly useful in eliciting calcium elevations in plant cells, a common second messenger that transmits numerous extracellular signals. However, the study as it stands now is too preliminary for publication in *Nature* for the reasons described below.

What is conceptually new? Previous studies demonstrated importance of calcium signals under stress conditions and developmental processes. Such studies include genetic manipulations of calcium transporters and application of channel blockers or calcium chelators. Most of these approaches produce “loss of function” effects on specific signaling processes in which calcium is essential, including defense and drought responses, the processes discussed in the current manuscript. The optogenetics approach used here provides “gain of function” effect (inducible calcium influx) without actually applying stress signals. This approach reminded me earlier microinjection of caged calcium or calcium inducers to close stomata (Gilroy S, Read ND, Trewavas AJ. Elevation of cytoplasmic calcium by caged calcium or caged inositol triphosphate initiates stomatal closure. *Nature*. 1990 Aug 23;346(6286):769-71). While the methods are different, conceptually they achieve the same effect. That said, the current methodology does have its advantage of less invasiveness and more versatility. Authors are encouraged to take this advantage and extend the current work to yield new biological insights.

What is a surprising finding in the current work? The primary conclusion is that XXM-induced calcium influx causes a response distinct from the effect of anion efflux. This is not surprising. Although both anion efflux and cation influx causes a membrane depolarization, it becomes more complex if the cation is calcium, a second messenger in addition to a charge carrier. The key issue here is that not all calcium influxes lead to defense responses as seemed to be the case in this study. The word “decrypting” in the title entails what is more than the current results can support.

What is problematic? Using the transient expression (*N. Benthamiana*) model, authors correlated short green light illumination with XXM-induced calcium elevation and ACR1-responsive depolarization. They then shifted to the transgenic tobacco plants to measure metabolites, defense gene marker expression, and whole plant phenotype following sustained green light treatments (ranging from 0.5-24 hours). A

critical omission is the corresponding calcium changes in XXM plants stemming from such long-term green light exposures. Likewise, the consequence of long-term treatments was also unknown for ACR1 plants in terms of membrane potential or anion efflux. This is clearly important in the context of current knowledge about calcium signals: without exception they are either single transient elevation, multiple peaks, or oscillations. What does sustained activation of XXM do to the cellular calcium levels, a single peak as in the case of transient activation, or sustained calcium overloading? I spent considerable amount of time looking for such information but failed to find any in the manuscript. Related to this is the fact that calcium overloading kills cells in all systems tested (both animal and plant cells), consistent with what the author found after 24 hour treatment. Is this a demonstration of calcium as a signal to induce SPECIFIC cellular responses or a non-specific response to calcium overloading? Answer to this question may benefit from suggested experiments below.

Do different calcium changes cause distinct responses? It is generally accepted in the field that different temporal and spatial patterns of calcium elevation elicit distinct cellular responses. The work here seems to suggest that calcium influx means only one thing: defense response leading to cell death. As stated earlier, this could be a non-specific toxic effect due to calcium overloading. In principle, XXM can be used to produce quantitative calcium influx episodes through variable green light schemes, reminiscent to a variety of calcium signals generated in response to different stress conditions. If authors produce such a variety of calcium changes and compare the subsequent responses, they can use XXM as a tool to “decrypt” the specificity of calcium signatures. This holds the key to the improvement of this manuscript. In order to achieve this goal, authors can utilize the transgenic lines co-expressing XXM and a genetically-encoded calcium indicator, produce multiple distinct calcium elevation patterns mimicking those triggered by stress factors (e.g., drought, cold, salt, pathogen elicitors), and then compare the responses. To ensure biological relevance, they can compare each response to that actually triggered by a specific stress factor. These studies are highly feasible and will be necessary to achieve the proof-of-concept of “decrypting” different signals and give biological significance to this work.

Some other comments for improvements include:

- To claim the XXM-induced responses are calcium-dependent, authors need to show such responses can be blocked by the XXM-specific blockers. This is important because XXM is permeable to other cations.
- Authors measured XXM subcellular localization in epidermal cells, but do the work using mesophyll cells instead.
- Leakage of anions from cells can generate drought effect. Does this mean naturally occurring drought stress results from anion efflux? If authors want to use ACR1 to mimic drought signal, they need to show that drought conditions imposed to plant cells start with or at least involve anion efflux. Otherwise, it will be difficult to render biological relevance to this approach.

Referee #3 (Remarks to the Author):

This is a very elegant study that used engineered microbial rhodopsins to induce calcium or anion flux upon shining of green light (that would not cause photosynthetic effects) in plants that produce retinal. Depending on the engineered rhodopsin, a pulse of green light could therefore induce calcium flux or a change in electric (membrane potential) signal.

Using this system, the authors showed that a calcium signal can trigger ROS production, while an electric signal cannot. The calcium and ROS signals triggered necrotic spot formation, JA accumulation and pathogen responses. In contrast, the electric signal produced an osmotic like response with increases in ABA and proline. Transcriptomic analysis of the two responses supported these two different pathways.

Overall, this is an interesting and elegant study. However, I have a few questions regarding the conclusions that can be drawn from it:

1. How natural are the responses the authors are studying? The systems they developed are certainly very sophisticated, but they are far from resembling what happens in a natural situation. When stress occurs, or pathogen attacks, is it only one signal that is generated (as the authors attempt to create in their system)? The answer is most likely not. I think the authors would be better served if they studied what happened in mutants of key regulators of these responses such as *glr3.3glr3.6*, *rbohDrobhF*, *msl10*, *aha1* and others. In a previous study it was found for example that in the *glr3.3glr3.6* mutant the calcium and electric waves were absent, but a ROS wave still occurred, in response to a local high light stress treatment (Fichman Y, Mittler R. Integration of electric, calcium, reactive oxygen species and hydraulic signals during rapid systemic signaling in plants. *Plant J.* 2021 Jul;107(1):7-20. doi: 10.1111/tpj.15360. Epub 2021 Jun 25. PMID: 34058040.).

2. What about pH? Rhodopsins are also proton pumps/channels, and they could alter the pH of the cell affecting different processes. In a previous study (long ago) it was found that expressing a microbial rhodopsin in tobacco plants (that do not accumulate retinal) resulted in lesion formation and activation of pathogen responses and this phenotype was proposed to result from pH changes (Mittler R, Shulaev V, Lam E. Coordinated Activation of Programmed Cell Death and Defense Mechanisms in Transgenic Tobacco Plants Expressing a Bacterial Proton Pump. *Plant Cell.* 1995 Jan;7(1):29-42. doi: 10.1105/tpc.7.1.29. PMID: 12242350; PMCID: PMC160762.). I think that at a minimum the authors need to measure the pH of cells to rule this out.

3. The methods for measuring ROS should be revised. At the minimum the authors should grind the plant tissues in liquid nitrogen resuspend in TCA, spin, neutralize the supernatant, and quantify with Amplex-Red against a standard curve of H₂O₂.

4. Growing plants under constant red light for 45 days before green light treatment is not normal for plants. Not only that the phytochrome system that is very important for ROS signaling (Phytochrome B regulates reactive oxygen signaling during abiotic and biotic stress in plants. Yosef Fichman, Haiyan

Xiong, Soham Sengupta, Rajeev K. Azad, Julian M. Hibberd, Emmanuel Liscum, Ron Mittler. bioRxiv 2021.11.29.470478; doi: <https://doi.org/10.1101/2021.11.29.470478>) will be altered, but also the constant light is a stress.

In summary, pending text revisions (disclaimers) and additional references (see above), as well as measuring pH changes, this manuscript could be accepted.

Very elegant indeed!

Author Rebuttals to Initial Comments:

Point-by-point answer list to Referees' comments:

Referee #1 (Remarks to the Author):

General comment:

The manuscript of Ding et al describes the introduction of optogenetic tools in plants to manipulate membrane polarization in general, and the flow of Ca⁺⁺ ions in particular, towards unravelling the role played by electrical signals in the triggering of different physiological responses/programmes upon environmental cues. For this, they employed two different light-gated ion channels: a) ACR1, an anion channelrhodopsin previously introduced into plants by some of the authors of this article, and b) XXM, a further engineered channelrhodopsin with high Ca⁺⁺ permeability (previously used in animal cells, published by some of the authors of this article). They also employed the plants engineered previously for production of the necessary cofactor, retinal. They performed a functional characterization of the tools in transiently transformed *Nicotiana benthamiana* and transgenic *Nicotiana tabacum* leaves. This includes determination of membrane potential changes, cellular Ca⁺⁺ level changes using the R-GECO1 sensor, ROS production. They analysed certain particular metabolic markers of plant stress responses (e.g. hormones) and the transcription of a few genes related to some of the metabolic and signalling pathways mediated by those particular metabolites (plant-pathogen interaction, drought stress). The article deals with the introduction of technologies, that despite being already available in other fields/biological systems and having helped advancing our understanding of complex cellular processes, are unfortunately not yet being used broadly in plant research. Therefore, this kind of approaches are very much necessary and welcome! The experimental strategy used by the authors is very clever and potentially useful, namely to perform a targeted manipulation at a particular point/level within the series of events leading to the activation of a signalling cascade thereby trying to determine the relative contribution of the components involved in the resulting physiological response observed. The introduction of the (functional!) tools and the proof of principle example of the potential relevance for the field that it represents being able to trigger of cellular responses upon targeted manipulation of membrane potential and fluxes of specific signalling ions (Ca⁺⁺), both known to be involved in the plant responses to environmental cues, are the main strengths of the article and represent the main point and message of the paper. However, this reviewer considers that the article falls certainly short in two main aspects, and the central claims are not backed up with sufficient data which precludes assessing its generalised/straightforward usability:

Question 1A

Firstly, in order to make the approach really available for use by the community a much more complete molecular and functional characterization of the optogenetic systems is needed. A non-limiting list includes as is state-of-the-art and essential, i) for the tools tunability and quantitative control by using pulsing schemes, different intensity and time dose responses, reversibility/frequency of ion waves, dynamic ranges obtained.

Answer: We have extended the functional characterization of the light-gated Ca²⁺ channel presented here rigorously. We now added the demonstration of the reproducibility of XXM 2.0-induced Ca²⁺ signals and the ability to modify Ca²⁺ signatures by light intensity and light duration.

Question 1B

ii) for the plants: cell-cell and plant-plant variability/reproducibility, full phenotypic analysis and effects on membranes, major metabolic and physiological parameters (with and without activation) – the current work just shows some representative pictures of the plants that do not allow a proper evaluation, and just dry weight data. This is still ok if just showing that the tools work, but it is critical to have it available if the plants are to be used to study biological processes.

Answer: We fully agree with the reviewer that it is important to display the variability and reproducibility of the experiments. We are sorry if this was not made clear. We now display all our results with appropriate standard error bars and displaying the individual data points. The biological and technical replicates are indicated in the legends.

The reviewer asked for a full phenotypic characterization of the transgene optotool plants in the presence or absence of activation light. In addition to the previous gross phenotypical analysis, we now included pH-imaging experiments in XXM 2.0 plants, Ca²⁺-imaging in ACR1 plants, maximal quantum yield of photosynthesis and ion-leakage experiments.

Furthermore, we added data on the comparison of the transcriptomes and targeted metabolic approaches upon different degree of optotool activation side-by-side with negative and positive controls inducing biotic and abiotic stress. We believe that this quite extended phenotypical analysis cover the range of plant parameters to judge on the major biological pathways triggered by the two optotools we compared in this work. Since the wide ranging and counteractive phenotypical effects are backed up by the transcriptomics and targeted metabolic data we generated, we believe our conclusions on the biological processes controlled by Ca²⁺, voltage, hydraulic signals are justified and straightforward.

Question 2A

Secondly, several/most of the key advantages of using light as an inducer and having a tool that permits targeted manipulation at a high spatiotemporal resolution and with reversibility & quantitative control are not explored at all in the work. These are all aspects that optogenetics enabled in the several biological systems in which it is revolutionising the way of studying complex cellular networks. For instance, one could generate waves of activation, compare different time-lengths of activation vs stronger activation, etc. Besides this, the experiments are bulk. Without the need to generate any biological material single cell activation can be easily performed and the set up/activity analysed.

Answer: In line with the reviewer's suggestion, we have conducted a new set of experiments. These involved varying the durations of optogenetic stimulation and comparing the XXM 2.0 and ACR1-induced metabolic and transcript response patterns with those of a representative biotic or abiotic stress. As requested, we chose representative controls with water stress (PEG) for ACR1 and pathogen attack (Pst) for XXM 2.0. We could

show that short (1 h), medium (4 h) or continuous (24 h) low green light exposures induced a dose-response effect at both the transcript and metabolite levels, with significant:

- i) Immune response initiated by XXM 2.0, and
- ii) Water stress response triggered by ACR1.

From these experiments we concluded that both metabolites/hormones and transcript responses can be switched On/Off in a rapid and light dose-dependent manner. We discuss the fact that the described responses depend on the strength of XXM 2.0/ACR1 stimulation and duration and are reversible which is made clear in a new chapter of the revised manuscript and now displayed in Fig. 3&4, Extended Data Fig. 16&19.

Furthermore, the reviewer proposed to test several different light-application protocols like waves of activation, including different light intensities, to study the physiological effects of those signatures at the cellular level caused by ACR1 or XXM 2.0 activation. In fact, we share the enthusiasm of the reviewer and agree that these are very interesting approaches that should be pursued. However, understanding the role of different Ca^{2+} or electric signatures, a topic of increasing interest nowadays, requires a separate, in-depth project. We are currently working on this very complex project and can say that solving the question about signature specificity exceeds the scope of the present manuscript by far. With the experimental design presented here, we already analyzed 108 leaf samples for the transcriptomics approach and 1018 samples to quantify (the few) metabolites/hormones levels over time. To extend this methodology to study Ca^{2+} and voltage signatures with different frequency, light duration, and intensity, along with a comprehensive metabolomics and transcriptomics approach, is beyond the scope of our current work. We respectfully propose that such a comprehensive investigation will require a dedicated manuscript of its own.

Question 3

Finally, a main concern is about the claims drawn in terms of applicability of the system, raised in the abstract, and text/discussion. Namely that with the experiments performed using the tools it is possible to determine/discriminate the differential contribution of the different electrochemical and ionic signals elicited in terms of a specific response either biotic or abiotic. This is not justified nor backed up with the experimental design used and data obtained. Plant-pathogen interactions, wounding and abiotic environmental stresses such as drought elicit a broad range of metabolic and physiological responses, are perceived by the plants via different dedicated receptors or simply produce cell mechanical damage, and activate a complex mesh of signalling pathways integrating the exogenous cues with the genetic program of the cells/tissues. Dozens, hundreds of regulatory networks are involved, with partially overlapping, redundant, complementary or opposing functions among different stresses, are tissue and developmental stage specific, and mediate the signal relay via hundreds of metabolites including hormones and other small molecules. In the present study a very limited set of metabolites were chosen and were analysed, and similarly only a couple of targeted and limiting selection of target genes studied. The elements are per se interesting and involved in some of the pathways, but so are hundreds of others, the choice is at some point arbitrary, as other (very) important components are not included. Just

particular features are picked up that are not sufficient and necessarily representative of the processes involved and of the global changes produced. Moreover, the experiments do not include the counterparts including the natural environmental hardships to whom they are compared. To be able to demonstrate the applicability of the tool to study those processes and be able to yield useful data, and given the complexity of the systems under study, the obvious experimental strategy has to comprise performing a broad metabolomics and transcriptomics profiling of the plant tissue with and without optogenetic activation of both channels, together with the use of standard plant-pathogen interactions and drought stresses. The methods for this are currently state of the art, trivial (and readily available, standard). This would provide a picture of the specificity of action of the pathways involved, crosstalk between the two type of signals and the relevance of those signals in the framework of a wide range of other components simultaneously activated during such environmental challenges. Moreover, there is a quantitative and time-resolved aspect involved in the activation of different pathways during the stress conditions which can only be understood if a whole network analysis is performed. Optogenetic approaches have demonstrated to be optimal to perform such targeted manipulations at highest quantitative control capabilities (waves of activation, compare different time-lengths of activation vs stronger activation, etc all factors involved in plant-environment interactions), and such experimental means were not used here. This would inform on the applicability of the tools and if successful demonstrate the full potential of the strategy to the plant community, most willing to achieve such kind of pressing analysis.

Answer: At the transcript level, we have now exploited the full availability of the transcriptome and compared its changes by different degrees of optotool activation side by side with regulation by water stress (abiotic) and pathogen attack (biotic) by means of watering with PEG and spraying *Pseudomonas syringae* pat. *tomato* (*Pst*) onto the leaves, respectively, as requested by the reviewer. We could lay out those DEGs and GO-terms which were different between XXM 2.0 and ACR1 and shared by XXM 2.0 and Pst treatment and those shared by water stress (PEG) and ACR1 activation, which strengthened our claim that both optotools address distinct and very different plant responses. Namely, ACR1 induced water stress responses and XXM 2.0 induced pathogen stress and immune responses.

With regard to the metabolomics approach, the reviewer's criticism isn't fully clear to us, because we have investigated exactly these metabolites important in immune/ pathogen responses (JA and its precursors, SA) or in drought stress (ABA and proline). In the context of our time-resolved approach with different optotool activation strength, an untargeted metabolic approach is beyond feasibility, given the amount of different metabolites. It is impossible to accomplish this because one would need to adjust the sample extractions and preparation (for each and every time-point, optotool and control treatment) for the different metabolite polarities which require distinct modifications or ionizations before they can be analyzed by mass spec. We believe that our targeted metabolomics approach mentioned above made visible that the two optotools responses matched the biotic/abiotic control treatments used (Pst/PEG), and demonstrated nicely that ACR1 and XXM 2.0 trigger distinct and convergent pathways, which is one of the main messages of the paper. Our transcriptomics analyses highlighting relevant biotic and abiotic metabolic and

phytohormone pathways (Figure 4, Extended Data Fig. 19 and Extended Data Table1) substantiated the picture of a specific triggering of separate signaling pathways by ACR1 2.0 and XXM 2.0

Minor points:

#1 why was it in the experiments in *Xenopus* oocytes blue light used? The activation/absorption/excitation spectra of the tool(s) should be included as well.

Answer: Blue light was used in *Xenopus* oocytes since the peak of the action spectrum of XXM 2.0 is ~460 nm. However, for plant experiments, we were particularly careful to avoid activating the native blue light photoreceptors. In fact for XXM 2.0 activation in plants we made use of green light (520-532 nm), which is least absorbed by plants and does not address the known blue or red-light receptors. Our transcriptomics analyses also showed green light illumination (520 nm, 9 $\mu\text{W}/\text{mm}^2$) to barely induced different expressed genes (DEGs) (see Extended Data Table1). We now explained this in more detail in the text and also provide the action spectrum of XXM 2.0 (Extended Data Fig. 5). The action spectrum of ACR1 was already published in our previous *Nature Plants* paper (Zhou, Y. *et al.* Optogenetic control of plant growth by a microbial rhodopsin. *Nature Plants* **7**, 144-151, doi:10.1038/s41477-021-00853-w (2021)).

#2 It is not clear in several experiments what the “n” refers to: samples from one leaf, different leaves of one plant, different plants, independent lines, etc? I haven’t found the info in the Summary report neither.

Answer: We changed the description about “n” in the text and the legends accordingly to make clear how many biological or technical replicates were used for a particular dataset.

#3 Legend to figure 2 is not clear

Answer: We revised the figure legend and made clear what the panels in the Fig show.

#4 A side by side comparison is not available for various experiments and under the same conditions, for instance Ca^{++} measurement with the ACR1 line. This would be very useful to analyse the selectivity/specificity.

Answer: A side by side comparison of the Ca^{2+} -signal upon ACR1 and XXM 2.0 activation was already given by the aequorin measurements (Fig. 1i), but we now included another Ca^{2+} -recording with R-GECO1 including ACR1 as well (Extended Data Fig. 9). We had previously omitted such measurements because of an overlap of the excitation spectra of R-GECO1 (Zhao, Y. *et al.* An Expanded Palette of Genetically Encoded Ca^{2+} Indicators. *Science (New York, N.Y.)* **333**, 1888-1891, doi:10.1126/science.1208592 (2011)) and ACR1 2.0 (Zhou, Y. *et al.* Optogenetic control of plant growth by a microbial rhodopsin. *Nature Plants* **7**, 144-151, doi:10.1038/s41477-021-00853-w (2021) and Wietek, J., Broser, M., Krause, B. S. & Hegemann, P. Identification of a Natural Green Light Absorbing Chloride Conducting Channelrhodopsin from *Proteomonas sulcata*. *J Biol Chem* **291**, 4121-4127,

doi:10.1074/jbc.M115.699637 (2016)) which complicates straight-forward interpretation of the imaging analysis. In the Ret-XXM 2.0 line instead an excitation spectrum cross-talk free Ca²⁺-imaging with R-GECO1 can be performed, because the action spectrum of XXM 2.0 is blue-shifted.

Upon the reviewer's suggestion, we found it valuable to demonstrate the R-GECO1 Ca²⁺ imaging routine with ACR1 2.0, despite the fact that it does activate ACR1 2.0. The ACR1 2.0 action/absorbance spectrum is in the green range peaking at ~520 nm and exciting R-GECO1 at 570 nm activates ACR1 2.0 considerably (Extended Data Fig. 9). R-GECO1-based Ca²⁺-imaging every 5-6 seconds resulted in fast ACR1 2.0 activation and deactivation events demonstrated by transient depolarizations when simultaneous voltage recordings are performed (Extended Data Fig. 9). While the R-GECO1 fluorescence already remains stable at those imaging control conditions it does not increase in the presence of additional green light (in the imaging interval times), which confirmed the Aequorin-based results presented in Fig. 1h, that ACR1 2.0 does not induce Ca²⁺-signals as presented in the new Extended Data Fig. 9.

#5 In the figure 3d) what is the meaning of the peak of OPDA after one hour?

Answer: Since OPDA is discussed to not only to represent a metabolic precursor of JA but also a signaling molecule that regulates diverse biological processes in a JA-independent manner (Jimenez Aleman, G. H., Thirumalaikumar, V. P., Jander, G., Fernie, A. R. & Skirycz, A. OPDA, more than just a jasmonate precursor. *Phytochemistry* **204**, 113432, doi:10.1016/j.phytochem.2022.113432 (2022)) we quantified and chose to present OPDA, JA and JA-Ile without any further interpretation of the OPDA peak at 1h. OPDA is required for the production of the active form of JA and subsequently JA-Ile. Consequently, OPDA must be present at a similar time scale as JA/JA-Ile, but whether the speed of OPDA conversion to JA is regulated under certain conditions is unclear. We have no evidence for the active regulation of this process which is why these considerations have not been discussed in the text and we did not speculate about the role that OPDA (as a signaling molecule) has at time point 1 h after XXM 2.0 activation.

#6 figure 3a, magnifications and contrast of the different panels are not consistent

Answer: We appreciate the particular attention to details by the reviewer in evaluating our manuscript. We have made a concerted effort to ensure consistency across different panels throughout the document. However, if there appears to be any overlooked inconsistency, we are more than happy to correct for that. Any specific guidance or suggestion on this front is very welcome.

Referee #2 (Remarks to the Author):

General comment:

Authors genetically engineered a channelrhodopsin variant with high Ca²⁺ conductance,

named XXM 2.0, to trigger cytosolic Ca²⁺ elevations in plant cells. Light-induced Ca²⁺ influx was studied in comparison with the light-gated anion channelrhodopsin ACR1 that causes anion efflux-dependent depolarization. While a XXM-induced Ca²⁺ signal caused plants to initiate ROS production and defense mechanism, anion efflux caused drought-stress responses. These findings imply that Ca²⁺ signals and anion efflux serve as elicitors for distinct responses.

Optogenetic approach for channel manipulation has been well documented in animal systems but much less in plant cells. This study, together with previously published works from the same authors, indicates that channelrhodopsins provide promising tools for altering membrane potential and ionic transport activities in plant cells. As far as I am aware, the use of anion efflux ACR1 in plants was previously published, whereas Ca-permeable XXM was used here in plants for the first time. The XXM variants can be particularly useful in eliciting calcium elevations in plant cells, a common second messenger that transmits numerous extracellular signals. However, the study as it stands now is too preliminary for publication in Nature for the reasons described below.

What is conceptually new? Previous studies demonstrated importance of calcium signals under stress conditions and developmental processes. Such studies include genetic manipulations of calcium transporters and application of channel blockers or calcium chelators. Most of these approaches produce “loss of function” effects on specific signaling processes in which calcium is essential, including defense and drought responses, the processes discussed in the current manuscript. The optogenetics approach used here provides “gain of function” effect (inducible calcium influx) without actually applying stress signals. This approach reminded me earlier microinjection of caged calcium or calcium inducers to close stomata (Gilroy S, Read ND, Trewavas AJ. Elevation of cytoplasmic calcium by caged calcium or caged inositol triphosphate initiates stomatal closure. *Nature*. 1990 Aug 23;346(6286):769-71). While the methods are different, conceptually they achieve the same effect. That said, the current methodology does have its advantage of less invasiveness and more versatility. Authors are encouraged to take this advantage and extend the current work to yield new biological insights.

Question 1

What is a surprising finding in the current work? The primary conclusion is that XXM-induced calcium influx causes a response distinct from the effect of anion efflux. This is not surprising. Although both anion efflux and cation influx causes a membrane depolarization, it becomes more complex if the cation is calcium, a second messenger in addition to a charge carrier. The key issue here is that not all calcium influxes lead to defense responses as seemed to be the case in this study. The word “decrypting” in the title entails what is more than the current results can support.

Answer: We have changed the title of the manuscript. The new title is: “Probing plant signal processing non-invasively by two channelrhodopsins”

Question 2

Using the transient expression (N. Benthamiana) model, authors correlated short green light illumination with XXM-induced calcium elevation and ACR1-responsive depolarization. They then shifted to the transgenic tobacco plants to measure metabolites, defense gene marker expression, and whole plant phenotype following sustained green light treatments (ranging from 0.5-24 hours). A critical omission is the corresponding calcium changes in XXM plants stemming from such long-term green light exposures. Likewise, the consequence of long-term treatments was also unknown for ACR1 plants in terms of membrane potential or anion efflux. This is clearly important in the context of current knowledge about calcium signals: without exception they are either single transient elevation, multiple peaks, or oscillations. What does sustained activation of XXM do to the cellular calcium levels, a single peak as in the case of transient activation, or sustained calcium overloading? I spent considerable amount of time looking for such information but failed to find any in the manuscript. Related to this is the fact that calcium overloading kills cells in all systems tested (both animal and plant cells), consistent with what the author found after 24 hour treatment. Is this a demonstration of calcium as a signal to induce SPECIFIC cellular responses or a non-specific response to calcium overloading? Answer to this question may benefit from suggested experiments below.

Answer: The reviewer's concern about the potential impact of sustained Ca^{2+} elevation over an extended period, and the worry that the effects induced by XXM 2.0 activation might result from overloading the cells with Ca^{2+} (which could consequently lead to cell death) are duly noted.

Based on a new set of experiments we in the revision show that strong transcriptional changes of genes associated with a hypersensitive response and JA synthesis genes by XXM 2.0 activation together with metabolite levels for JA-Ile and its precursors are already observed 30-60 min after XXM 2.0 activation (Fig. 3 and 4, Extended Data Fig. 21). Furthermore, and important in the context of the reviewers' hypothesis of Ca^{2+} -overloading, we do provide data now that after shutting-off XXM 2.0 after 1 or 4 hours illumination, a rapid decline in transcripts associated with immune responses and hormone levels within the following 2-3 hours occurs. This strongly indicates that the tissue is intact, having all the capabilities to respond to changing signaling cues. This assumption is also in line with the transcriptomics and metabolomics data, showing a rapid decline and in many cases a return to almost background levels after shutting off the green light.

Question 3

Do different calcium changes cause distinct responses? It is generally accepted in the field that different temporal and spatial patterns of calcium elevation elicit distinct cellular responses. The work here seems to suggest that calcium influx means only one thing: defense response leading to cell death. As stated earlier, this could be a non-specific toxic effect due to calcium overloading. In principle, XXM can be used to produce quantitative calcium influx episodes through variable green light schemes, reminiscent to a variety of calcium signals generated in response to different stress conditions. If authors produce such a variety of calcium changes and compare the subsequent responses, they can use XXM as a tool to "decrypt" the specificity of calcium signatures. This holds the key to the improvement of this manuscript. In order to achieve this goal, authors can utilize the transgenic lines co-

expressing XXM and a genetically-encoded calcium indicator, produce multiple distinct calcium elevation patterns mimicking those triggered by stress factors (e.g., drought, cold, salt, pathogen elicitors), and then compare the responses. To ensure biological relevance, they can compare each response to that actually triggered by a specific stress factor. These studies are highly feasible and will be necessary to achieve the proof-of-concept of “decrypting” different signals and give biological significance to this work.

Answer: The reviewer mentions that it is established in the plant Ca^{2+} field that different Ca^{2+} signatures cause specific responses in plants. However, this hypothesis, coming from the animal field has, with very few exceptions (Allen, G. J. et al. A defined range of guard cell calcium oscillation parameters encodes stomatal movements. *Nature* 411, 1053-1057 (2001) and Whalley, H. J. & Knight, M. R. Calcium signatures are decoded by plants to give specific gene responses. *New Phytologist* 197, 690-693, doi:10.1111/nph.12087 (2013), Whalley, H. J. et al. Transcriptomic Analysis Reveals Calcium Regulation of Specific Promoter Motifs in Arabidopsis. *The Plant Cell* 23, 4079-4095, doi:10.1105/tpc.111.090480 (2011)) over the last 30 years of intense plant Ca^{2+} research, not been confirmed with strong evidence-based data and it could never be confirmed with phenotypical outputs on the level of whole plants. In fact, there is still no consensus that defined Ca^{2+} signatures are produced in response to distinct stimuli, and a wide range of Ca^{2+} responses with a huge variability (from cytosolic Ca^{2+} decreases to increases in the seconds to minute and hour range) to the same stimuli have been demonstrated in the literature i.e. with osmotic (NaCl) stress (Feng, W. et al. The FERONIA Receptor Kinase Maintains Cell-Wall Integrity during Salt Stress through Ca^{2+} Signaling. *Curr Biol* 28, 666-675.e665, doi:10.1016/j.cub.2018.01.023 (2018); Schmöckel, S. M. et al. Different NaCl-Induced Calcium Signatures in the Arabidopsis thaliana Ecotypes Col-0 and C24. *PLOS ONE* 10, e0117564, doi:10.1371/journal.pone.0117564 (2015); Knight, H., Trewavas, A. J. & Knight, M. R. Calcium signalling in Arabidopsis thaliana responding to drought and salinity. *Plant J* 12, 1067-1078, doi:10.1046/j.1365-313x.1997.12051067.x (1997); Cramer, G. R. & Jones, R. L. Osmotic stress and abscisic acid reduce cytosolic calcium activities in roots of Arabidopsis thaliana. *Plant, Cell and Environment* 19, 1291-1298, doi:10.1111/j.1365-3040.1996.tb00007.x (1996); Halperin, S. J., Gilroy, S. & Lynch, J. P. Sodium chloride reduces growth and cytosolic calcium, but does not affect cytosolic pH, in root hairs of Arabidopsis thaliana. *J Exp Bot* 54, 1269-1280, doi:10.1093/jxb/erg134 (2003); Gao, D., Knight, M. R., Trewavas, A. J., Sattelmacher, B. & Plieth, C. Self-Reporting Arabidopsis Expressing pH and Ca^{2+} Indicators Unveil Ion Dynamics in the Cytoplasm and in the Apoplast under Abiotic Stress. *Plant Physiology* 134, 898-908, doi:10.1104/pp.103.032508 (2004); Liu, L. et al. Both NaCl and H_2O_2 long-term stresses affect basal cytosolic Ca^{2+} levels but only NaCl alters cytosolic Ca^{2+} signatures in Arabidopsis. *Frontiers in Plant Science* 9, doi:10.3389/fpls.2018.01390 (2018)). The reviewer is correct in his assessment that the XXM 2.0 is excellently suited for asking whether defined Ca^{2+} signatures encode specific responses. Additional experiments (Extended Data Fig.7, 8, 9) added for the functional characterization of the XXM 2.0 tool showed that this optogenetic tool has great potential to address the reviewer’s urgent question about Ca^{2+} -signature decoding.

However, the focus of this work is different and a prerequisite to understanding how plants read Ca^{2+} signatures, decode them, and initiate appropriate responses. The establishment of

XXM 2.0 is the kick-start of a new era in plant Ca^{2+} -signaling research where the assessment of the signature hypothesis is just one of the many fundamental questions to be answered. The goal of the work presented here was to establish rhodopsin-based optogenetic tools to impose in a non-invasive “clean” way Ca^{2+} and electric signals to address leaf stress pathway analyses. This is why we compared XXM 2.0 (Ca^{2+} plus depolarization) and ACR1 (depolarization) responses. Triggering either electric or Ca^{2+} -signals was not possible in the past, but most stimuli applied up until now were based on invasive methods that caused both, electric and Ca^{2+} -signals.

We do provide solid evidence here that the combination of a Ca^{2+} and electric signal (XXM 2.0), but not a depolarization alone, induce an immune response, which is currently under debate in the community (Farmer, E. E., Gao, Y.-Q., Lenzoni, G., Wolfender, J.-L. & Wu, Q. Wound- and mechanostimulated electrical signals control hormone responses. *New Phytologist* 227, 1037-1050, doi:10.1111/nph.16646 (2020); Sukhova, E. & Sukhov, V. Electrical Signals, Plant Tolerance to Actions of Stressors, and Programmed Cell Death: Is Interaction Possible? *Plants* 10, doi:10.3390/plants10081704 (2021)). In addition we show that ACR1-induced anion efflux mediated depolarization address osmostress pathways with accurate on/off features. By the removing of light and inactivation of ACR1 in particular, it will be possible to study the “recovery from wilting response” that occurs within 30 min, which is not possible by any other approach. Having said that, we want to point out that ACR1 can induce the redistribution of chloride ions and water associated with corresponding genetic and metabolic reprogramming in a non-invasive way that represents an experimentally “clean” system to study in detail the molecular mechanisms to cope with water stress. To elucidate the exact molecular and regulatory components involved in a specific Ca^{2+} signature and/or distinct electrical fluctuation, further studies are needed, some of which require very complex and extensive analyses that are beyond the scope of this work.

Some other comments for improvements include:

1) To claim the XXM-induced responses are calcium-dependent, authors need to show such responses can be blocked by the XXM-specific blockers. This is important because XXM is permeable to other cations.

Answer: The reviewer’s expressed concern that the plant response to XXM 2.0 activation may not be attributed to the Ca^{2+} conductance and suggested using a ChR2 specific blocker. This is a common strategy in plant and animal science, but is pointless in optogenetic approaches. Despite the fact that there is no ChR2 specific blocker, we can turn channel activity on and off by the presence or absence of green light, which is much more accurate than applying blockers.

The concern that XXM 2.0, the light-gated cation channel with high Ca^{2+} conductance, allows other ions to pass is valid, and we thank the reviewer for pointing this out, because it prompted us to include all-optical physiology experiments with cellular pH imaging in Ret-XXM 2.0 plants and Ca^{2+} -imaging in Ret-ACR1 2.0 plants in the manuscript. From these new

experiments (Extended Data Fig. 6&9), we can conclude that ACR1 does not induce Ca^{2+} -signals and that XXM 2.0 activation only causes transient, but no sustained pH change as seen for cellular Ca^{2+} concentration. This may suggest that XXM 2.0 has a small H^+ conductance and is unable to cause a significant change in cellular pH, but it may also mean that the sudden increase in cytosolic Ca^{2+} concentration has a short-term effect on H^+ transport via Ca^{2+} -signaling (Behera, S. et al. Cellular Ca^{2+} Signals Generate Defined pH Signatures in Plants. *The Plant Cell* 30, 2704-2719, doi:10.1105/tpc.18.00655 (2018)). Anyway, from the experiments we are able to deduce that XXM 2.0 can cause large and sustained Ca^{2+} elevations but it can not cause sustained pH changes.

2) Authors measured XXM subcellular localization in epidermal cells, but do the work using mesophyll cells instead.

Answer: The localization studies shown in the manuscript were intended to show the optimization steps by molecular engineering to improve subcellular localization in order to target the Chr2 variant from the ER to the plasma membrane. This could be best shown with the epidermis, which is often in one focal plane. The electrophysiological measurements with mesophyll cells showed the same trend as the localization improvement in the epidermal cells, which supports the analogous improvement in mesophyll cells and epidermal cells.

3) Leakage of anions from cells can generate drought effect. Does this mean naturally occurring drought stress results from anion efflux? If authors want to use ACR1 to mimic drought signal, they need to show that drought conditions imposed to plant cells start with or at least involve anion efflux. Otherwise, it will be difficult to render biological relevance to this approach.

Answer: Osmotically driven water fluxes are crucial in drought stress or stomatal movement. Passive water export followed by turgor loss has already been demonstrated by ACR1 2.0 activation in guard cells (Huang, S. et al. Optogenetic control of the guard cell membrane potential and stomatal movement by the light-gated anion channel GtACR1. *Science Advances* 7, eabg4619, doi:10.1126/sciadv.abg4619 (2021)). We make use of the same process in the present work. ACR1 2.0 activation leads to anion efflux, the resulting depolarization leads to efflux of potassium, resulting in a net efflux of KCl from the cell. This will lower the water potential in the apoplast and thus water passively diffuses out of the cell, the turgor pressure drops. A low water potential in the apoplast is also critical in water stress. Important to mention is that water stress or salt stress also leads to membrane depolarization and efflux of potassium (Graus, D. et al. Tobacco leaf tissue rapidly detoxifies direct salt loads without activation of calcium and SOS signaling. *New Phytol* 237, 217-231, doi:10.1111/nph.18501 (2023)), thus the ACR1 2.0 induced change and water stress effects are very similar, cells lose water due to an osmotically driven process involving cellular KCl release.

Another important feature of our ACR1 2.0-induced osmostress is that via onset and termination of green light stimulation plant scientists have the possibility to study early steps associated with the loss and recovery of the plant water status. After termination of the

green light stimulus the plant regains turgor within only 20-30 min (see extended Data Movie2). This fast “recovery from wilting response”, which is hardly possible by any other technology, might be helpful to identify plant ion transporters that have potential to serve as markers for modern plant breeding strategies to improve drought stress tolerance in crops.

Referee #3 (Remarks to the Author):

General comment:

This is a very elegant study that used engineered microbial rhodopsins to induce calcium or anion flux upon shining of green light (that would not cause photosynthetic effects) in plants that produce retinal. Depending on the engineered rhodopsin, a pulse of green light could therefore induce calcium flux or a change in electric (membrane potential) signal.

Using this system, the authors showed that a calcium signal can trigger ROS production, while an electric signal cannot. The calcium and ROS signals triggered necrotic spot formation, JA accumulation and pathogen responses. In contrast, the electric signal produced an osmotic like response with increases in ABA and proline. Transcriptomic analysis of the two responses supported these two different pathways.

In summary, pending text revisions (disclaimers) and additional references (see below), as well as measuring pH changes, this manuscript could be accepted. Very elegant indeed!

Overall, this is an interesting and elegant study. However, I have a few questions regarding the conclusions that can be drawn from it:

Questions

1) How natural are the responses the authors are studying? The systems they developed are certainly very sophisticated, but they are far from resembling what happens in a natural situation. When stress occurs, or pathogen attacks, is it only one signal that is generated (as the authors attempt to create in their system)? The answer is most likely not. I think the authors would be better served if they studied what happened in mutants of key regulators of these responses such as *glr3.3glr3.6*, *rbohDrobhF*, *msl10*, *aha1* and others. In a previous study it was found for example that in the *glr3.3glr3.6* mutant the calcium and electric waves were absent, but a ROS wave still occurred, in response to a local high light stress treatment (Fichman Y, Mittler R. Integration of electric, calcium, reactive oxygen species and hydraulic signals during rapid systemic signaling in plants. *Plant J.* 2021 Jul;107(1):7-20. doi: 10.1111/tpj.15360. Epub 2021 Jun 25. PMID: 34058040.).

Answer: The reviewer is addressing potentially very important molecular components in controlling of long-range Ca^{2+} and electrical signals. Indeed, Ted Farmer has shown that the GLR double mutant is only involved in mediating long range Ca^{2+} signals from leaf to leaf and does not affect local responses (Sukhova, E. & Sukhov, V. Electrical Signals, Plant Tolerance to Actions of Stressors, and Programmed Cell Death: Is Interaction Possible? *Plants* 10, doi:10.3390/plants10081704 (2021); Mousavi, S. A., Chauvin, A., Pascaud, F., Kellenberger, S.

& Farmer, E. E. GLUTAMATE RECEPTOR-LIKE genes mediate leaf-to-leaf wound signalling. *Nature* 500, 422-426, doi:10.1038/nature12478 (2013)). While recently it was shown that GLR3.3 is required for touch/wound induced local Ca^{2+} waves (Bellandi, A. et al. Diffusion and bulk flow of amino acids mediate calcium waves in plants. *Science Advances* 8, eabo6693, doi:doi:10.1126/sciadv.abo6693 (2022)).

However, in the framework of the current manuscript we were interested in distinguishing between Ca^{2+} /electrical versus electrical/hydraulic signaling in a local response. We really understand the reviewer's interest in long-range signaling because this could be nicely addressed by the two optotools, but this topic is beyond the scope of the present work.

2) What about pH? Rhodopsins are also proton pumps/channels, and they could alter the pH of the cell affecting different processes. In a previous study (long ago) it was found that expressing a microbial rhodopsin in tobacco plants (that do not accumulate retinal) resulted in lesion formation and activation of pathogen responses and this phenotype was proposed to result from pH changes (Mittler R, Shulaev V, Lam E. Coordinated Activation of Programmed Cell Death and Defense Mechanisms in Transgenic Tobacco Plants Expressing a Bacterial Proton Pump. *Plant Cell*. 1995 Jan;7(1):29-42. doi: 10.1105/tpc.7.1.29. PMID: 12242350; PMCID: PMC160762.). I think that at a minimum the authors need to measure the pH of cells to rule this out.

Answer: The reviewer is right about the relative conductivity of ChR2-versions and therefore his comment has been very helpful to improve the manuscript in that respect. In order to attribute the effect of XXM 2.0 to an ion species (Ca^{2+} or H^+), we have now included pH-imaging experiments with XXM 2.0, as well as Ca^{2+} -imaging experiments with ACR1 plants in the manuscript. These could show that ACR1 2.0 does not trigger Ca^{2+} increase and that the XXM 2.0 causes only a transient H^+ increase that quickly drops to the control level even with continued optogenetic stimulation. Thus XXM 2.0 does not induce a sustained pH change, but a sustained increase in Ca^{2+} , from which we conclude the XXM 2.0 induced plant responses are caused by the Ca^{2+} -increase.

The reviewer refers to one of the three papers by the group of Ron Mittler in which an opsin-based H^+ pump led to similar necrotic spots in leaves as the ones reported by XXM 2.0 in our current manuscript. The reviewer states that the phenotype associated with the H^+ -pump opsin was proposed to result from pH changes, but in the 3 related publications we could not find any data of pH measurements nor light-induced pump characteristics or photocurrents (Mittler, R., Shulaev, V. & Lam, E. Coordinated Activation of Programmed Cell Death and Defense Mechanisms in Transgenic Tobacco Plants Expressing a Bacterial Proton Pump. *Plant Cell* 7, 29-42, doi:10.1105/tpc.7.1.29 (1995); Pontier, D., Mittler, R. & Lam, E. Mechanism of cell death and disease resistance induction by transgenic expression of bacterio-opsin. *Plant J* 30, 499-509, doi:10.1046/j.1365-313x.2002.01307.x (2002); Rizhsky, L. & Mittler, R. Inducible expression of bacterio-opsin in transgenic tobacco and tomato plants. *Plant Mol Biol* 46, 313-323, doi:10.1023/a:1010617220067 (2001)). Furthermore, as the reviewer has already pointed out, the work on the H^+ -pump opsin was performed in the absence of the essential cofactor retinal. In our experience, however, the presence of retinal plays a crucial role for rhodopsin-based plant optogenetics, because without the cofactor

the opsins are neither stable nor are they correctly targeted (Zhou, Y. et al. Optogenetic control of plant growth by a microbial rhodopsin. *Nature Plants* 7, 144-151, doi:10.1038/s41477-021-00853-w (2021); Ullrich, S., Gueta, R. & Nagel, G. Degradation of channelopsin-2 in the absence of retinal and degradation resistance in certain mutants. *Biological chemistry* 394, 271-280, doi:10.1515/hsz-2012-0256 (2013)), which is why we step back from speculations about any similarity of the phenotypic results between XXM 2.0 and the H⁺-pump opsin mentioned by the reviewer.

3) The methods for measuring ROS should be revised. At the minimum the authors should grind the plant tissues in liquid nitrogen resuspend in TCA, spin, neutralize the supernatant, and quantify with Amplex-Red against a standard curve of H₂O₂.

Answer: Unfortunately, the reviewer does not give any reasons why we should revise the two methods to detect ROS which are established in the animal and plant field. With the help of the DAB staining, a highly appreciated method used for decades in phytopathology and beyond (Daudi, A. & O'Brien, J. A. Detection of Hydrogen Peroxide by DAB Staining in Arabidopsis Leaves. *Bio-protocol* 2, e263, doi:10.21769/BioProtoc.263 (2012); Venkidasamy, B., Karthikeyan, M. & Ramalingam, S. in *Reactive Oxygen, Nitrogen and Sulfur Species in Plants* 421-435 (2019); Busch, A. et al. MpTCP1 controls cell proliferation and redox processes in *Marchantia polymorpha*. *New Phytologist* 224, 1627-1641, doi:https://doi.org/10.1111/nph.16132 (2019); Rivero, R. M. et al. Delayed leaf senescence induces extreme drought tolerance in a flowering plant. *Proc Natl Acad Sci U S A* 104, 19631-19636, doi:10.1073/pnas.0709453104 (2007)), we could qualitatively show ROS production over time at the level of whole leaves upon optotool activation and upon control treatments (Fig. 2e, Extended Data Fig. 13b,c). With the aid of the platinum/iridium electrode, we were able to amperometrically monitor H₂O₂ production in real-time with quantitative readout upon XXM 2.0 activation (Fig. 2f, Extended Data Fig. 13d), and using the *Atrboh-D* mutant in a previous study, demonstrated that we can measure the production of H₂O₂ by the NADPH oxidase *in planta* (Güzel Deger, A. et al. Guard cell SLAC1-type anion channels mediate flagellin-induced stomatal closure. *New Phytologist* 208, 162-173, doi:10.1111/nph.13435 (2015)). The platinum electrodes have emerged as a valuable tool for amperometric ROS detection due to their unique properties, since platinum offers a wide electrochemical potential window, a high catalytic activity towards ROS reduction, and excellent stability in various electrolyte solutions. These features make platinum electrodes highly selective and sensitive to ROS, enabling their precise detection (Amatore, C. et al. Monitoring in Real Time with a Microelectrode the Release of Reactive Oxygen and Nitrogen Species by a Single Macrophage Stimulated by its Membrane Mechanical Depolarization. *ChemBioChem* 7, 653-661, doi:https://doi.org/10.1002/cbic.200500359 (2006); van Erk, M. R. et al. Reactive oxygen species affect the potential for mineralization processes in permeable intertidal flats. *Nature Communications* 14, 938, doi:10.1038/s41467-023-35818-4 (2023); Wang, Y. et al. Nanoelectrodes for determination of reactive oxygen and nitrogen species inside murine macrophages. *Proceedings of the National Academy of Sciences* 109, 11534-11539, doi:doi:10.1073/pnas.1201552109 (2012); Li, Y. et al. Highly Sensitive Platinum-Black Coated Platinum Electrodes for Electrochemical Detection of Hydrogen Peroxide and Nitrite in Microchannel. *Electroanalysis* 25, 895-902, doi:https://doi.org/10.1002/elan.201200456

(2013)). Ca²⁺-dependent ROS production by plant NADPH oxidases could only be shown in the heterologous HEK cell system so far (Ogasawara, Y. et al. Synergistic Activation of the Arabidopsis NADPH Oxidase AtrbohD by Ca²⁺ and Phosphorylation. *Journal of Biological Chemistry* 283, 8885-8892, doi:10.1074/jbc.M708106200 (2008)). We, by using this optogenetic strategy did (to the best of our knowledge) provide the first direct evidence for Ca²⁺ dependent ROS production in planta. Additionally, in the XXM 2.0 induced RNA transcripts we find ROS induced genes differentially expressed (DEGs) within a time-scale of ≤ 30 min (Fig. 4n, Extended Data Fig. 21f).

4) Growing plants under constant red light for 45 days before green light treatment is not normal for plants. Not only that the phytochrome system that is very important for ROS signaling (Phytochrome B regulates reactive oxygen signaling during abiotic and biotic stress in plants. Yosef Fichman, Haiyan Xiong, Soham Sengupta, Rajeev K. Azad, Julian M. Hibberd, Emmanuel Liscum, Ron Mittler. *bioRxiv* 2021.11.29.470478; doi: <https://doi.org/10.1101/2021.11.29.470478>) will be altered, but also the constant light is a stress.

Answer: The growth of plants under red light may not be a naturally occurring scenario, but they obviously are very healthy and we cannot detect any signs of stress, neither visually, nor at the metabolite level or with the transcriptomics approach. We have also been confronted with similar concerns associated with red light cultivation in our last two optogenetics research papers (Zhou, Y. et al. Optogenetic control of plant growth by a microbial rhodopsin. *Nature Plants* 7, 144-151, doi:10.1038/s41477-021-00853-w (2021); Huang, S. et al. Optogenetic control of the guard cell membrane potential and stomatal movement by the light-gated anion channel GtACR1. *Science Advances* 7, eabg4619, doi:10.1126/sciadv.abg4619 (2021)), but together with data in this manuscript we have quantified a large number of plant parameters from red light grown plants (biomass, reproduction, photosynthesis, stress metabolites, transcriptomics) which all indicate that the plants are doing well and there are no significant physiological limitations in plant behavior. Nevertheless, the paper the reviewer mentioned is relevant to our explanations about the importance of ROS signals in biotic and abiotic stress signaling and we cited it in the introduction part.

Reviewer Reports on the First Revision:

Referees' comments:

Referee #1 (Remarks to the Author):

The revised version of the article by Ding et al. represents an improvement and tackles most of the issues raised. However following points still need clarification:

Main point:

1) A key breakthrough of optogenetic approaches as shown for several biological systems already is the high spatial and temporal resolution it provides. The revised version improved considerably and exploited the temporal control, however still fully ignores the spatial control capabilities of the optogenetic tools. In question 2A, I raised the issue that this fundamental aspect is not dealt with at all in the article and proposed "...the experiments are bulk (meaning global illumination, whole plant/leaves, remote control was used but not spatial resolution). Without the need to generate any biological material single cell activation can be easily performed and the set up/activity analysed". This was not considered in the revised version. I think the article falls much short on this and this is my main concern, a main advantage/applicability is not demonstrated, shown. A simple experimental approach would cover this and considerably increase the impact and usefulness of the report for generalized implementation by future users.

Other points:

- 2) Last sentence of the abstract might profit from rewriting. The concept is not clear/mixed and doesn't seem to reflect the outcome.
- 3) Optogenetics is not non-invasive, it is minimally invasive but still... (Line 43/48/80; title). I suggest down-tuning.
- 4) Fig 3: why is it in 3e), Ja-Ile not induced upon Pst? (also JA and OPDA, extended data fig 17). The authors claim (lines 204-205) that this is normal for a biotrophic pathogen, but then if this is the case, then it's not that useful to compare the induction by XXM to that of such a pathogen, but maybe to one that does lead to changes in JA-related metabolism...
- 5) Line 214 (and elsewhere): "metabolomics" is rather overdoing it, it is rather a small-panel metabolite profiling, a handful of those.
- 6) Chapter "XXM and ACR1 control transcript profiles": This reviewer wonders whether the timeline of actual natural stress is much longer than the optogenetically activated signalling, it takes several hours until a drought stress kicks in. How was the protocol of PEG stress selected? Do plants watered with PEG show stress symptoms already after 1-4 h? It seems to be more a kind of gradual effect, in contrast to the activation by the optogenetic tools and this might have led to the difference in profiles. The authors compare in figure 4 actually 8 h PEG vs 1 h green light what seems to go into this direction (lines 269-271; fig 4e; idem for Pst in 4f). To help the readers, the authors should explain better in the text this issue and why/how they have come up with these comparisons/times.

7)Extended data:

Fig 2: relative/normalized to 10,000 actin? What does the label in the y-axis refer to? E.g. for Ret-ACR1 2.0#1 does it mean it is 10,000-fold over actin or 1:1?

Referee #2 (Remarks to the Author):

The manuscript has been improved and some issues raised in the first-round review have been addressed. This reviewer just read a recent study about green light response in Arabidopsis, which may be relevant to the general applicability of green-light-activated optogenetic systems in plants. Authors should cite this paper in the Discussion to make readers aware of potential limitations of optogenetic approaches should plants respond to a broad range of light spectra including the green light.

Yuhan Hao, Zexian Zeng, Xiaolin Zhang, Dixiang Xie, Xu Li, Libang Ma, Muqing Liu, Hongtao Liu, Green means go: Green light promotes hypocotyl elongation via brassinosteroid signaling, *The Plant Cell*, Volume 35, Issue 5, May 2023, Pages 1304–1317, <https://doi.org/10.1093/plcell/koad022>

I have three major questions on the data, one of which was raised before but was not addressed during the revision. The other two are on the new results. All three are interconnected regarding the validity of XXM-type optogenetic system and its connection with immunity.

1. The Ret-XXM2.0-mediated cation permeability has not been sufficiently clarified and can be misleading to readers. One of the subtitles of Main Text is “A light-gated Ca²⁺ channel for plants”. In a previous paper published by authors [Xiaodong Duan, Georg Nagel and Shiqiang Gao, Mutated Channelrhodopsins with Increased Sodium and Calcium Permeability, *Applied Science*, Feb 2019], XXM (ChR2 D156H) significantly increased both Na⁺ (to some extent, K⁺) and Ca²⁺ permeability while reducing H⁺ permeability. The Ret-XXM2.0 described in this paper has introduced a new mutation (H134Q) and other modifications to enhance protein expression and localization to the plasma membrane. It is fair to assume that the Ret-XXM2.0 not only enhanced Ca²⁺ influx but also Na⁺ influx, which can be toxic for plant cells. In this regard, neither a specificity test (Ca²⁺ vs Na⁺) nor proper clarification and justification has been provided, which might lead readers to believe that Ca²⁺ influx is the only causal factor of the XXM2.0-induced responses. Other Reviewers also asked related questions. Authors did provide evidence supporting that H⁺ homeostasis may not be a big concern, but if the D156H mutation in XXM, and very likely D156H/H134Q mutations in XXM1.1/1.2/1.3/2.0, enhanced both Na⁺ and Ca²⁺ permeability, authors should explicitly state so and explain why Na⁺ influx is not a problem for the conclusion drawn in the current manuscript. If authors choose to do a due diligence test, they can specifically block the Ca²⁺ influx through the XXM (using Ca-conductivity blockers or removing external Ca with EGTA etc) and see if the effect of the green light is blocked as well. (Another suggestion under question 1 is inclusion of description of mutant screen for H134Q. In a paper

reporting technical advances, readers will expect and benefit from a detailed procedure for experimental design and screening. If Main text has space limitation for such description, it should be included in the supplemental data/methods. If the procedure was previously described, the publications should be cited and acknowledged.)

2. The authors made a great effort in correlating ACR1-anion efflux to PEG treatment. Unfortunately, the work establishing a correlation between XXM-Ca²⁺ influx and Pst-infection was less robust. For example, there was no data showing that Pst-infection induces Ca²⁺ influx as XXM did. This is not to be taken for granted because I am not aware of any literature demonstrating whether Pst infection in *N. Benthamiana* induces Ca²⁺ influx. If authors know of such studies, please cite and discuss these in the context of Pst-XXM comparison. Otherwise, it is critical for authors to have this data added to the manuscript and bridge the gap. Authors have both AEQ and R-GECO1 lines to perform such a Ca²⁺ assay upon Pst-infection. Please also be reminded that *Arabidopsis* and *N. Benthamiana* may behave differently in pathogen responses.

3. The transcriptional profiling data from Pst-infection samples showed ~1000 DEGs, whereas light-activated XXM samples have 10-fold more (~10000) DEGs. If Ca²⁺ influx is a common signal of both treatments, it is difficult to comprehend why such a big difference was observed. It is generally believed that bacterial pathogens may cause more complex responses than just Ca-influx. Would we expect more diverse DEGs identified in response to bacteria instead of the other way around? Is the effect of XXM caused by Ca influx alone or are other factors also introduced into this system? If only 10% of the response to XXM activation mimics plant immune responses, what do other 90% represent? In this regard, question 1 and 2 become more important to address.

Comments on data presentation and analysis:

Fig. 1f,g,i: According to the figure legend, if “WT” in Fig. 1f,g denotes non-transgenic *N. tabacum*, what does the “WT” in Fig. 1i represent? It is not correct to use “WT” non-transgenic *N. tabacum* as a control for Ret-XXM2.0 in R-GECO1 detection? Please clarify what “WT” exactly means in all the figures (e.g., Fig. 2a, b, d and some Extended Data Figures). A general term “WT” without clear denotation is confusing.

Fig. 3i, j, k, l: High-concentration (OD 0.5) of Pst DC3000 induced a peak level of SA production at 8 hpi and the value is at 2-3 ug/ g DW. However, the light-activated XXM2.0 induced a peak level of 100 ug/ g DW or even higher at 8h after light exposure in both (24h) and (4h) light exposure. Such large differences (50-fold) have not been clearly explained. The “shortcut” speculation wasn’t supported by any experiments. Given the fact that both treatments had peak levels of SA at the same time, this cannot be simply a reason that XXM activation “bypass” pathogen perception.

In all the Figure legends: n=? biological replicates should be exact how many samples / leaves / oocyte cells and how many times the experiments have been done. The “biological replicates” do not convey explicit information regarding the robustness of the experiments. For single-cell assays, 6 cells in one repeat should not be described as “n=6 biological repeats”. It should be described as “n=6 cells from one batch of oocytes”. For all the figures, “n” should be clearly described to support the data robustness.

Line 124 and Extended Data Fig. 6: In fact, the pH homeostasis was changed as soon as the green light excitation was on. Although pH deflection is transient as the author pointed out, the pH changes seemed sustained to a similar level as the original version of ChR2 after the large, short-lived deflection. Therefore, it is not appropriate to conclude “Thus, activation of XXM most likely does not affect pH homeostasis.” This is important especially considering the procedures used in pH assay and Ca²⁺ influx. The green light excitation was very short for pH assay (i.e 5 min) as compared to other treatments that elicited Ca²⁺ responses (1h to 24 h). Therefore, authors may not want exclude the possibility that pH did change during longer excitations.

Extended Data Fig. 13a should include ACR1 and XXM samples for comparisons with PEG and Pst-inoculated samples, like what the authors did in Extended Data Fig.13c.

Lines 166-167: “ROS production was significantly less in leaves of ACR1 and PEG-treated plants (Extended Data Fig. 13c)”. I didn’t see they are significantly less compared to other samples, such as the Pst-treated samples.

Fig. 3b,c,d and Fig. 3j, k, l: In all panels for the same hormone, if ever possible, data should be presented in the same y-axis scale for readers to better compare the values.

Fig. 3e, f, g, h: The data indicated that Pst-inoculation does not induce JA but light-gated XXM 2.0 induces significantly higher JA production. Authors stated in Line 202: “Levels of JA-Ile and its precursors JA and OPDA were moderately elevated”? Please check Fig. 3e and Extended Data Fig. 17a, b. I didn’t see any elevation because the negative control sample (MgCl₂) also elevated to a similar level.

Extended Data Fig. 16 h, i: Please check the y-axis to make sure they are correctly labelled.

Extended Data Fig. 16 and 17 should be reorganized and merged. Line up the panels for the same hormone start with the panel that contains PEG and Pst samples. It will be similar to the Fig.3 for better comparison. Again, the same issue is the y-axis scale for these panels of the same hormone. The issues also occur in Fig. 4a, b. Another way to present subtle changes caused by the large y-axis scale might be to use “inset” within the panel or simply label the number on the data points.

Fig. 4 or Lines 211-293: This part (80 lines) is lengthy considering the whole results section only has 210 lines (Lines 83-293). Authors may consider shortening the detailed description and adding on the reasons why XXM activation caused 10X more complex transcriptional changes than the Pst.

Referee #4 (Remarks to the Author):

This is a revised manuscript following comments from reviewers. I was not one of the previous reviewers, but was guided to address the authors' response letter. This is a very interesting and nicely conducted research that uses synthetic light induced construct to study the transition from a macro world signal to a cell biology molecular level signaling.

The authors have reported adequately their methods and statistics in the text and figures. They have used t-Test and ANOVA (and post-hoc analysis). Per journal instructions, they mentioned p-value for the ANOVA and post hoc tests in the figures, while, to the opinion of this reviewer, it might be more informative to denote different letters to show significant of a group between all the groups and not just the statistical closest group.

As for the methods, I have noticed the reply to reviewer 3 regarding their query about the correctness of the ROS measurements. Reviewer 3 asked the authors to improve the way they measure ROS, but they declined. Using DAB is outdated and might give inaccurate results (As can be seen in the weird pattern of sample Ret-XXM2 in figure 2e, 1hr sample - that can occur due to de-staining conditions). The other method the authors have chosen is interesting, but not highly used by ROS researchers in living systems (the Nature Comm paper from 2023 is for sand samples). The ROS researchers have recently published a consensus statement (Murphy, M.P., Bayir, H., Belousov, V. et al. Guidelines for measuring reactive oxygen species and oxidative damage in cells and in vivo. *Nat Metab* 4, 651–662 (2022).

<https://doi.org/10.1038/s42255-022-00591>), where they guide the proper ways to measure ROS.

Amplex Red is a valid method to measure ROS, with limitations. The authors also mentioned in this reply the novelty of the connection between direct calcium activation of ROS production by RBOHs. I think that the ROS wave paper have shown it already in 2009 when they used calcium ionophores and recorded ROS responses.

Nevertheless, I think the hormones and transcriptomic analysis support the stress signal induction, and with a few additional corrections, this manuscript is a persuasive and clear research.

Author Rebuttals to First Revision:

Referees' comments:

Referee #1 (Remarks to the Author):

The revised version of the article by Ding et al. represents an improvement and tackles most of the issues raised. However following points still need clarification:

Main point:

1) A key breakthrough of optogenetic approaches as shown for several biological systems already is the high spatial and temporal resolution it provides. The revised version improved considerably and exploited the temporal control, **however still fully ignores the spatial control capabilities of the optogenetic tools**. In question 2A, I raised the issue that this fundamental aspect is not dealt with at all in the article and proposed "...the experiments are bulk (meaning global illumination, whole plant/leaves, remote control was used but not spatial resolution). **Without the need to generate any biological material single cell activation can be easily performed and the set up/activity analysed**". This was not considered in the revised version. I think the article falls much short on this and **this is my main concern**, a main advantage/applicability is not demonstrated, shown. A simple experimental approach would cover this and considerably increase the impact and usefulness of the report for generalized implementation by future users.

Answer: We appreciate this very good point about the advantage of spatial control capabilities of the optogenetic tools. Indeed, spatially confined optogenetic stimulation using ACR1 has already been demonstrated by us to guide single pollen tubes upon subcellular activation¹, and for studying stomatal movement at the cellular level².

We initially submitted this manuscript back-to-back with another one where we addressed the XXM based remote control of Ca²⁺ activation of guard cell anion channels and stomatal closure. For the convenience of the referee, we compiled 3 figures (appendix) to document that we have spatial control capabilities of the optogenetic tools.

At the level of whole plants, we agree with the reviewer about the potential that spatial-precise optogenetics control has to study topics such as long-distance signaling. To convince the referee that a spatial stimulation is possible in intact plants with XXM, we would like to share some more unpublished data (see Ref Fig 1) demonstrating that a local XXM activation at the major vein of a tobacco leaf (~2mm² area of illumination) can set off a traveling long-range electrical signal that can be measured at a distal site. These data, in our opinion, are not in any way related to the present manuscript focusing on the distinct electrical signals of ACR1 and XXM. Nevertheless, we do agree that such optogenetic studies of long-distance signaling are of great interest in future projects. In case the reviewers and editor think that these results can somehow be helpful to the current manuscript, we can add them as an extended data figure to strengthen our discussion on spatial resolution (main text: page 10, lines 333-337).

Despite of this, we added new experimental data to the manuscript documenting XXM-induced Ca²⁺ increases at the cell level in mesophyll protoplasts (Extended Data Fig 6). Thus, Ca²⁺ signals triggered by XXM stimulation are presented at the level of intact leaves (Fig. 1h), tissue (Fig. 1i) and cell level (Extended Data Fig. 6). The newly added protoplast experimental approach presented is valuable to the community since transient transformation and quantification of response parameters in protoplasts is a widespread method used, and was recently shown to be helpful in characterizing new optotools for plants³.

Ref Fig 1: Local green light stimulation at the main vein of leaves (indicated by the green spot on the left figure) triggers a long-distance traveling electric signal in XXM plants. Green arrow of the middle figure indicates the time when the 600 ms green light illumination applied. Error bars = s.e.m., n = 18 plants.

Other points:

2) Last sentence of the abstract might profit from rewriting. The concept is not clear/mixed and doesn't seem to reflect the outcome.

Answer: We followed the reviewer's advice and rephrased the last sentence of the abstract. The sentence "Our non-invasive optogenetics approach revealed that in plant leaves, distinct physiological responses cannot be triggered by electrical signals per se, unless they are combined with specific ion fluxes" was changed to "Our minimal-invasive optogenetics approach revealed that in plant leaves, specific physiological responses are elicited by distinct ion fluxes that are accompanied by comparable electric signals". To summarize our rationale briefly: In our study, strong depolarizations are triggered by both ACR1 and XXM. However, the observed physiological responses evoked are contrasting, raising the question about the depolarization encoded information. Given that ACR1 and XXM responses have little in common, we hypothesize that information specificity within an optogenetic stimulation is carried by the ion fluxes accompanying voltage changes.

3) Optogenetics is not non-invasive, it is minimally invasive but still... (Line 43/48/80; title). I suggest down-tuning.

Answer: We replaced non-invasive with minimally invasive in the main text and changed the title of the manuscript accordingly. Instead of "Probing plant signal processing non-invasively by two channelrhodopsins", the title now reads "Probing plant signal processing optogenetically by two channelrhodopsins"

4) Fig 3: why is it in 3e), Ja-Ile not induced upon Pst? (also JA and OPDA, extended data fig 17). The authors claim (lines 204-205) that this is normal for a biotrophic pathogen, but then if this is the case, then it's not that useful to compare the induction by XXM to that of such a pathogen, but maybe to one that does lead to changes in JA-related metabolism...

Answer: We want to point out that we never expected the *Pst* or any other physiological control to repeat in quality and quantity the exact response of XXM-induced Ca^{2+} signaling, especially in the case of DEGs. Our major aim was to confirm the most crucial biotic stress

indicators and signaling pathways by the physiological *Pst* control, which we basically did. To this purpose, we consider *Pst* to be an appropriate pathogen control due to the resemblance of its phenotype upon XXM activation, exhibiting necrotic spots, ROS production, and impaired photosynthesis, mirroring those observed during *Pst* infection in tobacco (*N. tabacum*)^{4,5}. Application of *Pseudomonas* bacteria, as well as the flagella epitope flg22 are known to induce a robust Ca²⁺ response in tobacco leaves^{6,7}, underscoring the potential for physiological control of a *Pst* infection. For the reviewers accessibility we confirmed these responses with those tobacco plants used in the present study and share the data about the induction of Ca²⁺ signals by flg22 and *Pst* (see Ref Fig 2 at page 7) in the framework of the revision with you. Since such responses are already reported, we refrained from adding them to the manuscript and cited the literature instead. However, if the reviewers and editor think that this data should be added as an Extended Data Figure, we can also include these results.

Our experimental design was conceived to circumvent genetic and metabolic reprogramming induced by wounding, so that we are able to compare it with the optogenetic response. We thus opted for a gentle spray inoculation method instead of *Pst* infiltration, ensuring accuracy in our transcriptomic experiments. *Pst* infiltration via a syringe would trigger a robust wounding response which is why we think it is reasonable to argue the difference in metabolite levels to originate from the above mentioned. *Pst* infection replicated all other biotic stress indicators (SA, ROS, necrosis, transcriptional reprogramming, negative impact on photosynthesis) as XXM stimulation. JA levels reported in the literature post *Pst* infection (via infiltration) exhibit inconsistencies in peak time after infection and the amount that is produced⁵, making our experimental outcome rather unpredictable. Nonetheless, our transcriptional analysis highlighted 'response to jasmonic acid' as a significantly enriched GO term shared well by XXM and *Pst* samples (Extended Data Fig. 22d). Our sampling protocol and its temporal resolution aimed to provide a more accurate transcriptomic core data, although it has compromised some resolution in measuring JA-Ile and its precursors, known to often appear in a transient fashion^{8,9}.

5) Line 214 (and elsewhere): “metabolomics” is rather overdoing it, it is rather a small-panel metabolite profiling, a handful of those.

Answer: Agreed – we performed requested changes

6) Chapter “XXM and ACR1 control transcript profiles”: This reviewer wonders whether the timeline of actual natural stress is much longer than the optogenetically activated signalling, it takes several hours until a drought stress kicks in. How was the protocol of PEG stress selected? Do plants watered with PEG show stress symptoms already after 1-4 h? It seems to be more a kind of gradual effect, in contrast to the activation by the optogenetic tools and this might have led to the difference in profiles. The authors compare in figure 4 actually 8 h PEG vs 1 h green light what seems to go into this direction (main text: lines 269-271; fig 4e; idem for *Pst* in 4f). To help the readers, the authors should explain better in the text this issue and why/how they have come up with these comparisons/times.

Answer: The reviewer is right. Both, PEG as well *Pst* evoked stress symptoms develop on a slower timescale compared to optogenetically activated signalling (see Fig. 4).

Based on the observed wilting phenotype with ACR1 plants, we aimed on designing a physiological control experiment with PEG that would mimic the wilting phenotype in a comparable time frame – within about 4-5 hours. We tested this in advance in an experimental series using different PEG concentrations and identified a concentration of 35% PEG to meet this criteria. After watering with PEG, the leaf wilting started from the lower leaves and moved upwards, reaching the 5th leaf (we used to quantify hormones, metabolites and transcripts) after 4 h (Extended Data Fig. 14).

In a similar approach, we also aimed to find a suitable control for the phenotypes observed with XXM in order to be able to discuss the results obtained in a physiological context. Since XXM plants develop necrotic lesions and exhibit elevated ROS, JA and SA levels, we hypothesized that XXM addresses at least wounding and/or immunity signaling processes. Therefore, we decided to employ *Pst* infection as appropriate physiological control and used spray inoculation to minimize wounding effects (as possibly evoked by infiltration). With spray inoculation, *Pst* bacteria have to overcome natural barriers before reaching the inner leaf tissues. Thus, one can assume that with spray inoculation disease symptoms develop even slower when compared to infiltration. We would also like to point out that comparable experimental approaches have not yet been carried out with tobacco plants. Based on this knowledge gap and in view of the fact that gene expression precedes the activation of physiological processes, we decided to investigate different time points: 1h, 4h, and 8h after activation of the rhodopsins or after application of the abiotic/biotic stress factors.

Analyzing the transcriptional profiles (and in line with the reviewer's comment) we learned that optogenetically activated transcriptional reprogramming was fast and reversible, while PEG as well as the *Pst* evoked stress symptoms developed continuously on a slower timescale (Fig. 4a-b).

In the main text (page 8, lines 238-242) we explain better and added new analysis in Fig. 4c, d for the rationale to compare the 8h PEG with the 1h and 4h ACR1 stimulation and the 8h *Pst* with the 1h and 4h XXM stimulation for the transcriptomics analysis. Analysis on the basis of gene function (GO terms) was used to erase the bias from the large differences in DEG numbers (Fig. 4a-d, Extended Data Fig. 21a-b). The triple venn diagram now displayed in Fig. 4c, d clearly demonstrates that the GO terms are shared best when 1h of the optogenetic stimulations (ACR1 and XXM) and 8h of the physiological treatments (PEG and *Pst*) are compared. As requested, we added more detailed information into the Materials and Methods section why we used 35% PEG as a physiological control treatment for the ACR1 treatment. The main text (page 5, line150-156) now contains explanations about this question as well.

7)Extended data:

Fig 2: relative/normalized to 10,000 actin? **What does the label in the y-axis refer to?** E.g. for Ret-ACR1 2.0#1 does it mean it is 10,000-fold over actin or 1:1?

Answer: In the Materials and Methods section we stated that transcripts were normalized to 10,000 molecules of actin. This means that in Ret-ACR1 line#1 the number of ACR1 transcripts is close to the level of actin.

Referee #2 (Remarks to the Author):

The manuscript has been improved and some issues raised in the first-round review have been addressed. This reviewer just read a recent study about green light response in Arabidopsis, which may be relevant to the general applicability of green-light-activated optogenetic systems in plants. Authors should **cite this paper** in the Discussion to make readers aware of potential limitations of optogenetic approaches **should plants respond to a broad range of light spectra including the green light.**

Yuhan Hao, Zexian Zeng, Xiaolin Zhang, Dixiang Xie, Xu Li, Libang Ma, Muqing Liu, Hongtao Liu, Green means go: Green light promotes hypocotyl elongation via brassinosteroid signaling, The Plant Cell, Volume 35, Issue 5, May 2023, Pages 1304–1317, <https://doi.org/10.1093/plcell/koad022>

Answer: From Extended Data Fig. 20 (the scheme of the transcriptomics study design), this study included the controls for green light treatment in Ret-YFP plants. Detailed analyses on green light-induced transcript regulation in control plants (Ret-YFP) are provided now as an Extended Data Table 5 to make visible the relative small number of DEGs (GO-terms and individual gene name presentation) induced by 1 h green light (38 DEGs up, 51 DEGs down) (main text: page 9, lines 295-297). We already had these results summed up in the Extended Data Table1 S1 before for transparency, but decided against discussing them intensively to keep the manuscript focused on the optogenetic outcome and not to get the reader sidetracked. As requested, in main text (page 9, lines 291-298) we now describe the small impact of green light on the transcriptome in the main text and discuss this result in the context of optogenetic experiments in plants with green light activation.

I have three major questions on the data, one of which was raised before but was not addressed during the revision. The other two are on the new results. All three are interconnected regarding the validity of XXM-type optogenetic system and its connection with immunity.

1. The Ret-XXM2.0-mediated cation permeability has not been sufficiently clarified and can be misleading to readers. One of the subtitles of Main Text is “A light-gated Ca²⁺ channel for plants”. In a previous paper published by authors [Xiaodong Duan, Georg Nagel and Shiqiang Gao, Mutated Channelrhodopsins with Increased Sodium and Calcium Permeability, Applied Science, Feb 2019], XXM (Chr2 D156H) significantly increased both Na⁺ (to some extent, K⁺) and Ca²⁺ permeability while reducing H⁺ permeability. The Ret-XXM2.0 described in this paper has introduced a new mutation (H134Q) and other modifications to enhance protein expression and localization to the plasma membrane. **It is fair to assume that the Ret-XXM2.0 not only enhanced Ca²⁺ influx but also Na⁺ influx, which can be toxic for plant cells.** In this regard, **neither a specificity test (Ca²⁺ vs Na⁺) nor proper clarification and justification has been provided**, which might lead readers to believe that Ca²⁺ influx is the only causal factor of the XXM2.0-induced responses. Other Reviewers also asked related questions. Authors did provide evidence supporting that H⁺ homeostasis may not be a big concern, but if the D156H mutation in XXM, and very likely D156H/H134Q mutations in XXM1.1/1.2/1.3/2.0, enhanced both Na⁺ and Ca²⁺ permeability, **authors should explicitly state so and explain why Na⁺ influx is not a problem for the conclusion drawn in the current manuscript.** If authors choose to do a due diligence test, they can specifically block the Ca²⁺

influx through the XXM (using Ca-conductivity blockers or removing external Ca with EGTA etc) and see if the effect of the green light is blocked as well.

Answer: In experiments with *Xenopus laevis* oocytes Na^+ current can be measured by XXM in addition to the Ca^{2+} current because the solutions to measure photocurrents in oocytes routinely contain 96 mM NaCl. In plants, however, the Na^+ content in the leaf apoplast is as low as ~ 1 mM^{10,11}. As requested, in the revised version of the manuscript we now refer to the Na^+ permeability of XXM and explain why in plants the Na^+ permeability of XXM is unproblematic under the prevailing experimental conditions (main text: page 9-10, lines 302-313). Only a very small inward directed Na^+ gradient can be assumed in plants¹², while that for Ca^{2+} is $\sim 10,000$ -fold and to be potentiated by the electrical gradient. An Extended Data Table 3 S2 is now presented showing that among the $> 11,000$ XXM DEGs, we do not find evidence for enriched salt-related marker transcripts. Furthermore, the osmoprotectant proline was reported to increase when Na^+ accumulates in cells¹³, which is also not the case in our study upon XXM stimulation (Extended Data Fig. 17a).

Besides the aforementioned explanations, we performed new experiments as requested. We setup an experimental approach proposed by the reviewer to discriminate between a possible Na^+ or Ca^{2+} -response. Following the reviewer's advice we monitored optogenetically-induced leaf phenotypes by facing 1 and 10 mM CaCl_2 (leaf apoplast Ca^{2+} is ~ 1 mM¹⁴), 10 mM NaCl and two different Ca^{2+} -chelators (BAPTA and EGTA). Our new results which are now discussed in the main text (page 10, lines 309-911) confirmed that the phenotype caused by XXM is due to Ca^{2+} , but not Na^+ (Extended Data Fig. 13).

(Another suggestion under question 1 is inclusion of description of mutant screen for H134Q. In a paper reporting technical advances, readers will expect and benefit from a detailed procedure for experimental design and screening. If Main text has space limitation for such description, it should be included in the supplemental data/methods. If the procedure was previously described, the publications should be cited and acknowledged.)

Answer: As requested, a detailed description on the screening procedure is placed in the Material and Methods section and more results from the screening process was added into Extended Data Fig. 1b-d.

2. The authors made a great effort in correlating ACR1-anion efflux to PEG treatment. Unfortunately, the work establishing a correlation between XXM- Ca^{2+} influx and Pst-infection was less robust. For example, there was no data showing that Pst-infection induces Ca^{2+} influx as XXM did. This is not to be taken for granted because I am not aware of any literature demonstrating whether Pst infection in *N. Benthamiana* induces Ca^{2+} influx. If authors know of such studies, please cite and discuss these in the context of Pst-XXM comparison. Otherwise, it is critical for authors to have this data added to the manuscript and bridge the gap. Authors have both AEQ and R-GECO1 lines to perform such a Ca^{2+} assay upon Pst-infection. Please also be reminded that *Arabidopsis* and *N. Benthamiana* may behave differently in pathogen responses.

Answer: A Ca^{2+} response to infection with *Pseudomonas* bacteria in tobacco has been reported in the literature^{7,15}. After infiltration with *Pseudomonas* bacteria, a prolonged Ca^{2+} uptake across the plasma membrane was detected between 1-7 h in *Nicotiana benthamiana* plants¹⁵. Furthermore, we recently studied bacterial-induced ion signals with flg22, a proxy for *Pst* infection⁶. In this work, we demonstrated a strong Ca^{2+} signal evoked

by flg22 in tobacco. We discuss this issue in the context of *Pst*-XXM comparison in the revised version of the manuscript (main text: page 5, lines 157-158).

For the reviewer's convenience, we performed *Pst* and flg22-dependent Ca^{2+} -imaging experiments using a perfusion system for the treatment as described^{6,16-18}, which eliminates motion and touch-induced imaging artifacts. The leaf discs were perfused with a control solution (bath solution for Ca^{2+} imaging in the method part) or the control solution containing *Pst* (final OD600 = 0.5) or 100 nM flg22, which resulted in cytosolic Ca^{2+} increases in *N. tabacum* (see Ref Fig 2). In case the reviewers and editor want to get these results included into the manuscript, we can add them as an Extended Data Figure.

Ref Fig 2: a,b R-GECO1 fluorescence change in R-GECO1 transgene *N. tabacum* in the (a) presence or (b) absence of *Pst* treatment, as indicated by the bar above the trace. Error bars = s.e.m., n = 6 plants from two batches of *N. tabacum* in (a) and (b). c, R-GECO1 fluorescence change induced by flg22 treatment. Error bars = s.e.m., n = 9 leaves from R-GECO1 transiently expressed *N. benthamiana* leaves. d, R-GECO1 fluorescence change in R-GECO1 transgene *N. tabacum* leaves upon flg22 treatment. Error bars = s.e.m., n = 6 plants from one batch of *N. tabacum*.

3. The transcriptional profiling data from *Pst*-infection samples showed ~1000 DEGs, whereas light-activated XXM samples have 10-fold more (~10000) DEGs. If Ca^{2+} influx is a common signal of both treatments, it is difficult to comprehend why such a big difference was observed. It is generally believed that bacterial pathogens may cause more complex responses than just Ca^{2+} influx. Would we expect more diverse DEGs identified in response to bacteria instead of the other way around? Is the effect of XXM caused by Ca^{2+} influx alone or are other factors also introduced into this system? **If only 10% of the response to XXM activation mimics plant immune responses, what do other 90% represent? In this regard, question 1 and 2 become more important to address.**

Answer: So far, the effect of optogenetic tools – including XXM – on transcriptional

reprogramming has not been investigated. Compared to plants treated with *Pst*, the enormous number of DEGs addressed by XXM seems surprising at first glance. For the recognition of pathogens, plants possess a considerable number of both membrane-bound and cytosolic receptors, which control specific cellular responses in the context of pathogen and effector triggered immunity (PTI and ETI). The specificity of recognition and the downstream processes is determined not only by the structure of the receptors themselves, but also by their specific interdependence with interacting mediator proteins as well as their temporal and spatial proximity to each other (signalosomes). Depending on the pathogens' nature, Ca²⁺ signaling may be transient or long lasting and temporarily and spatially confined^{7,15,19,20}. For the second messenger Ca²⁺, we believe that the reviewer would agree that the number of Ca²⁺-addressed DEGs will correspond to the duration/strength of the signal. Thus, we may not expect that the *Pst* physiological control covers exactly the response of XXM stimulation.

According to our results, upon activation, XXM seems to activate a rather 'global' Ca²⁺ response, addressing several Ca²⁺ dependent pathways (Extended Data Table 3 S1) at once. It is tempting to speculate that depending on the activation protocol or XXM spatial localization downstream responses can be fine-tuned and dissected. We discussed this in the last section of the manuscript (main text: page 8, lines 265-269).

The Venn diagram, to which the reviewer refers, shows the shared and unshared DEGs from a comparison of 8 h *Pst* treatment and 1 h XXM stimulation with a total of 1,638 and 11,070 DEGs, respectively. Of the *Pst* treatment DEGs, more than 70% are covered by XXM stimulation. Due to the difference in DEG quantity, a comparison of all the *Pst* and XXM DEGs results in coverage of ~11%. However, if we compare the DEGs of *Pst* and XXM leaves by gene function using gene ontology analysis, of all the XXM triggered GO terms, ~37% are covered by *Pst* treatment GO terms (see Figure 4d).

When gene function of the XXM specific DEGs that were previously not shared with *Pst* were analyzed using GOSlim methodology, merging GO terms of identical or similar biological function, (Extended Data Table 2 S1-S3), we found many of them to actually participate in plant immune responses. The DoubleVenn diagram shows now more than 54% from the GOSlim terms of XXM specific DEGs were covered by *Pst* treatment GOSlim terms (Extended Data Table 2 S4). In the up-regulated XXM specific DEGs, ~77% GOSlim terms were shared by *Pst* induced immune responses while cell death appears more specific for XXM (Extended Data Table 2 S4). In the main text (page 8, lines 260-265), we added explanations to the XXM specific DEGs compared to *Pst*.

Comments on data presentation and analysis:

Fig. 1f,g,i: According to the figure legend, if "WT" in Fig. 1f,g denotes non-transgenic *N. tabacum*, what does the "WT" in Fig. 1i represent? It is not correct to use "WT" non-transgenic *N. tabacum* as a control for Ret-XXM2.0 in R-GECO1 detection? Please clarify what "WT" exactly means in all the figures (e.g., Fig. 2a, b, d and some Extended Data Figures). A general term "WT" without clear denotation is confusing.

Answer: "WT" in Fig. 1f,g represent non-transgenic *N. tabacum*. "WT" in Fig. 1i represent the wild type *N. bentamiana* transiently expressing R-GECO1. "Ret-XXM2.0" in Fig. 1i denote wild type *N. bentamiana* transiently expressing R-GECO1 and Ret-XXM2.0. We now changed all the legends to state clearly what plant lines were used as controls.

Fig. 3i, j, k, l: High-concentration (OD 0.5) of *Pst* DC3000 induced a peak level of SA production at 8 hpi and the value is at 2-3 ug/ g DW. However, the light-activated XXM2.0 induced a peak level of 100 ug/ g DW or even higher at 8h after light exposure in both (24h) and (4h) light exposure. Such large differences (50-fold) have not been clearly explained. The “shortcut” speculation wasn’t supported by any experiments. Given the fact that both treatments had peak levels of SA at the same time, this cannot be simply a reason that XXM activation “bypass” pathogen perception.

Answer: It is well known that spray inoculation of *Pst* is accompanied by a steady increase in pathogen concentrations inside leaves, resulting in a successive increase in leaf infection over time still increasing within days²¹. In main text (page 6-7, lines 200-210), we explained in more detail that spraying the leaf surface with *Pst*, subsequent entry of the bacteria through the stomata, followed by *Pst* multiplication inside, will not lead to infection of the entire leaf tissue, whereas the XXM stimulation reaches all areas of the leaf instantaneously, which fits to data we presented (Fig. 3i-j, Fig. 4b) .

Spray inoculation was used in our study as a control, because infiltration of *Pst* into the leaf via a syringe would have caused a strong wounding response, a scenario we succeeded to overcome by optogenetics at last. We think it is thus reasonable to argue the difference in metabolite levels to originate from the above mentioned conditions and discuss this in more detail in the context of the results (main text: page 6-7, lines 200-210). With respect to the comment of the difference in time-line of metabolites, we did only have one time point between 4h and 24h, namely 8h. Thus, some uncertainty remains about the metabolite dynamics and exact peak time of SA. However, when we compared the SA levels at 1 h and 2 h, no difference between *Pst* 1h and *Pst* 2h was detected, but with XXM at 2h a ~10 fold change of SA levels compared with XXM 1h (Fig. 3j) was observed. This supports our notion that the *Pst* delay-effect in metabolite synthesis is due to strength of the infection of leaf tissue by *Pst* and it is likely to develop gradually. We explained this better in the main text now (page 6-7, lines 200-210).

In all the Figure legends: n=? biological replicates should be exact how many samples / leaves / oocyte cells and how many times the experiments have been done. The “biological replicates” do not convey explicit information regarding the robustness of the experiments. For single-cell assays, 6 cells in one repeat should not be described as “n=6 biological repeats”. It should be described as “n=6 cells from one batch of oocytes”. For all the figures, “n” should be clearly described to support the data robustness.

Answer: The requested information is now given for each panel in the legend of the figures. In these experiment, n refers to n of *Xenopus laevis* oocytes, n of transient expressed *N. benthamiana* leaves and n of transgenic tobacco plants. We explained this better in the legend part.

Line 124 and Extended Data Fig. 6: In fact, the pH homeostasis was changed as soon as the green light excitation was on. Although pH deflection is transient as the author pointed out, the pH changes seemed sustained to a similar level as the original version of ChR2 after the large, short-lived deflection. Therefore, it is not appropriate to conclude “Thus, activation of XXM most likely does not affect pH homeostasis.” This is important especially considering the procedures used in pH assay and Ca²⁺ influx. The green light excitation was very short for pH assay (i.e 5 min) as compared to other treatments that elicited Ca²⁺ responses (1h to 24 h). Therefore, authors may not want exclude the possibility that pH did change during longer excitations.

Answer: In the previous version of our manuscript, we demonstrated XXM to trigger short transient pH deflections but steady-state pH homeostasis to reach control levels already ~3 min after stimulus onset during continuous XXM stimulation (formally Extended Data Fig. 6, now Extended Data Fig. 7). Following the suggestion of the reviewer and to test for pH change during longer optogenetic stimulation, we performed the all-optical approach using the pH-biosensor pHuji and extended the XXM activation time to ~1h green light. Similar to the previous result with short excitation, no continuing pH effect can be observed upon longer-term excitation (see updated Extended Data Figure 7).

Extended Data Fig. 13a should include ACR1 and XXM samples for comparisons with PEG and Pst-inoculated samples, like what the authors did in Extended Data Fig.13c.

Answer: We follow the advice of the reviewer and assembled the figure as requested, namely combined the ion leakage measurements of ACR1 and XXM with Extended Data Fig. 13a. The previous Extended Data Fig. 13a is now Extended Data Figure 15a.

Lines 166-167: “ROS production was significantly less in leaves of ACR1 and PEG-treated plants (Extended Data Fig. 13c)”. I didn’t see they are significantly less compared to other samples, such as the Pst-treated samples.

Answer: We now present a statistical analysis on the difference between the groups the reviewer refers to in Extended Data Figure 15c. We have carefully revised the section and substantiated our statements with the newly added histochemical statistics (main text: page 5, lines 164-165).

Fig. 3b,c,d and Fig. 3j, k, l: In all panels for the same hormone, if ever possible, data should be presented in the same y-axis scale for readers to better compare the values.

Answer: We followed the advice of the reviewer and assembled the figure as requested, using the same y-axis scale.

Fig. 3e, f, g, h: The data indicated that Pst-inoculation does not induce JA but light-gated XXM 2.0 induces significantly higher JA production. Authors stated in Line 202: “Levels of JA-Ile and its precursors JA and OPDA were moderately elevated”? Please check Fig. 3e and Extended Data Fig. 17a, b. I didn’t see any elevation because the negative control sample (MgCl₂) also elevated to a similar level.

Answer: We thank the reviewer for the careful reading and assessment of the data. The sentence was corrected accordingly (main text: page 6-7, lines 199-200). The scale of the corresponding figure (Fig. 3e and now Extended Data Figure 19 a, e) was not adjusted to the same y-axis scale as the ones with the optogenetic treatments because we want to provide the reader with the clear information that the spray control (MgCl₂ spray) already had some effect on JA-levels and its precursors, which is also upon the request of referee#1 in comment: 4).

Extended Data Fig. 16 h, i: Please check the y-axis to make sure they are correctly labelled.

Answer: We checked these. They are correct.

Extended Data Fig. 16 and 17 should be reorganized and merged. Line up the panels for the same hormone start with the panel that contains PEG and Pst samples. It will be similar to the Fig.3 for better comparison. Again, the same issue is the y-axis scale for these panels of the same hormone. The issues also occur in Fig. 4a, b. Another way to present subtle changes caused by the large y-axis scale might be to use “inset” within the panel or simply label the number on the data points.

Answer: We reorganized and merged these figures now (see Extended Data Fig. 18 and 19). In Fig. 4a, b we use two y-axis to present the numbers of DEGs for different samples. Fig. 4 or Lines 211-293: This part (80 lines) is lengthy considering the whole results section only has 210 lines (Lines 83-293). Authors may consider shortening the detailed description and adding on the reasons why XXM activation caused 10X more complex transcriptional changes than the Pst.

Answer: We changed the main text (page 7-9, lines 214-288) about the transcriptomics-part due to rearrangements of figures and compressed the descriptions of the results part. Furthermore we added explanations to the XXM specific DEGs compared to *Pst* (main text: page 8, lines 260-269).

Referee #4 (Remarks to the Author):

This a revised manuscript following comments from reviewers. I was not one of the previous reviews, but was guided to address the authors' response letter. This is a very interesting and nicely conducted research that uses synthetic light induced construct to study the transition from a macro world signal to a cell biology molecular level signaling.

The authors have reported adequately their methods and statistics in the text and figures. They have used t-Test and ANOVA (and post-hoc analysis). Per journal instructions, they mentioned p-value for the ANOVA and post hoc tests in the figures, while, to the opinion of this reviewer, it might be more informative to denote different letter to show significant of a group between all the groups and not just the statistical closest group.

Answer: For clarity and in order to be able to display the items clearly to the reader, in some but not all figures, statistical analysis was presented, as requested.

As for the methods, I have noticed the reply to reviewer 3 regarding their query about the correctness of the ROS measurements. Reviewer 3 asked the authors to improve the way they measure ROS, but they declined. Using DAB is outdated and might give inaccurate results (As can be seen in the weird pattern of sample Ret-XXM2 in figure 2e, 1hr sample - that can occur due to de-staining conditions). The other method the authors have chosen is interesting, but not highly used by ROS researchers in living systems (the Nature Comm paper from 2023 is for sand samples). The ROS researchers have recently published a consensus statement (Murphy, M.P., Bayir, H., Belousov, V. et al. Guidelines for measuring reactive oxygen species and oxidative damage in cells and in vivo. Nat Metab 4, 651–662 (2022). <https://doi.org/10.1038/s42255-022-00591>), where they guide the proper ways to measure ROS. Amplex Red is a valid method to measure ROS, with limitations.

Answer: The amperometric technique with quantitative readout we used to detect ROS dynamics is a method that directly measures H₂O₂ with precise temporal resolution, while the method proposed (Amplex Red) is an indirect fluorescence-based method which is prone to artefacts, as stated by reviewer#4. Additionally this staining methods to detect ROS only provides snapshots at a given time. Our direct biophysical measurement has advantage over the indirect method proposed by reviewer#3, and it is used in the animal field routinely. Amplex Red for ROS detection in plants has been introduced in Arabidopsis^{22,23}. The supporting data presented to the reviewer (see Ref Fig 3) shows that Amplex Red can indeed be used to measure ROS in Arabidopsis, but the protocol fails to detect ROS in tobacco leaves upon XXM, but more importantly also with the positive (wounding) controls. Attempts with different buffer solutions to optimize the ROS extraction and Amplex Red recording conditions all failed. Although we are aware of the

possible disadvantages of chemical indicators such as DAB, the figures with this routinely used dye reflect the results of our amperometric measurement method and an additional techniques, which was added to the manuscript (see below).

According to the paper the reviewer mentioned (Murphy et al., *Nat Metab* (2022)), luminol is a valid option to indicate ROS production, but as any chemical indicator (such as AmplexRed) is based on an indirect measure that does have limitations. We now included successful ROS recordings with luminol luminescence measurement, that again pointed towards strong XXM 2.0-triggered ROS production (Extended Data Fig. 15e). Taken together, we have demonstrated solid production of ROS by XXM stimulation using three independent methods.

Ref Fig 3: H₂O₂ extracts with phosphate buffer (PB) or trichloroacetic acid (TCA) from wounded leaves of WT *N. tabacum* and WT Arabidopsis or after 0 and 5 min green light (9 μW/mm²) treated Ret-XXM 2.0 transgene *N. tabacum*. H₂O₂ was detected and quantified by Amplex® Red hydrogen peroxide/peroxidase assay kit (Invitrogen, catalog number: A22188). Error bars = s.e.m., n = 6-8 plants.

The authors also mentioned in this reply the novelty of the connection between direct calcium activation of ROS production by RBOHs. I think that the ROS wave paper have shown it already in 2009 when they used calcium ionophores and recorded ROS responses.

Answer: We have toned down our statement about the proof for in planta Ca²⁺-dependent ROS production as suggested by the reviewer.

Nevertheless, I think the hormones and transcriptomic analysis support the stress signal induction, and with a few additional corrections, **this manuscript is a persuasive and clear research.**

References

- 1 Zhou, Y. *et al.* Optogenetic control of plant growth by a microbial rhodopsin. *Nature Plants* **7**, 144-151, doi:10.1038/s41477-021-00853-w (2021a).
- 2 Huang, S. *et al.* Optogenetic control of the guard cell membrane potential and stomatal movement by the light-gated anion channel *GtACR1*. *Science Advances* **7**, eabg4619, doi:10.1126/sciadv.abg4619 (2021).
- 3 Ochoa-Fernandez, R. *et al.* Optogenetic control of gene expression in plants in the presence of ambient white light. *Nature Methods* **17**, 717-725, doi:10.1038/s41592-020-0868-y (2020).
- 4 Tran, B. Q. & Jung, S. Modulation of chloroplast components and defense responses during programmed cell death in tobacco infected with *Pseudomonas syringae*. *Biochemical and biophysical research communications* **528**, 753-759, doi:10.1016/j.bbrc.2020.05.086 (2020).

- 5 Kenton, P., Mur, L. A. J., Atzorn, R., Wasternack, C. & Draper, J. (—) Jasmonic Acid Accumulation in Tobacco Hypersensitive Response Lesions. *Molecular Plant-Microbe Interactions*® **12**, 74-78, doi:10.1094/mpmi.1999.12.1.74 (1999).
- 6 Li, K. *et al.* An optimized genetically encoded dual reporter for simultaneous ratio imaging of Ca²⁺ and H⁺ reveals new insights into ion signaling in plants. *New Phytol* **230**, 2292-2310, doi:10.1111/nph.17202 (2021).
- 7 Atkinson, M., Keppler, L., Orlandi, E., Baker, C. & Mischke, C. Involvement of Plasma Membrane Calcium Influx in Bacterial Induction of the K⁺/H⁺ and Hypersensitive Responses in Tobacco. *Plant physiology* **92**, 215-221, doi:10.1104/pp.92.1.215 (1990).
- 8 Lee, G., Joo, Y., Kim, S.-G. & Baldwin, I. T. What happens in the pith stays in the pith: tissue-localized defense responses facilitate chemical niche differentiation between two spatially separated herbivores. *The Plant Journal* **92**, 414-425, doi:doi.org/10.1111/tpj.13663 (2017).
- 9 Zhang, G. *et al.* Jasmonate-mediated wound signalling promotes plant regeneration. *Nature Plants* **5**, 491-497, doi:10.1038/s41477-019-0408-x (2019).
- 10 Conn, S. & Gilliham, M. Comparative physiology of elemental distributions in plants. *Annals of Botany* **105**, 1081-1102, doi:10.1093/aob/mcq027 (2010).
- 11 Mühling, K. H. & Läuchli, A. Determination of apoplastic Na⁺ in intact leaves of cotton by in vivo fluorescence ratio-imaging. *Functional plant biology : FPB* **29**, 1491-1499, doi:10.1071/fp02013 (2002).
- 12 Blatt, M. R. A charged existence: A century of transmembrane ion transport in plants. *Plant Physiology*, doi:10.1093/plphys/kiad630 (2024).
- 13 Khare, T., Srivastava, A. K., Suprasanna, P. & Kumar, V. Individual and additive stress impacts of Na⁺ and Cl⁻ on proline metabolism and nitrosative responses in rice. *Plant Physiol Biochem* **152**, 44-52, doi:10.1016/j.plaphy.2020.04.028 (2020).
- 14 Stael, S. *et al.* Plant organellar calcium signalling: an emerging field. *J Exp Bot* **63**, 1525-1542, doi:10.1093/jxb/err394 (2012).
- 15 Nemchinov, L. G., Shabala, L. & Shabala, S. Calcium efflux as a component of the hypersensitive response of *Nicotiana benthamiana* to *Pseudomonas syringae*. *Plant Cell Physiol* **49**, 40-46, doi:10.1093/pcp/pcm163 (2008).
- 16 Dreyer, I. *et al.* Transporter networks can serve plant cells as nutrient sensors and mimic transceptor-like behavior. *iScience* **25**, doi:10.1016/j.isci.2022.104078 (2022).
- 17 Gutermuth, T. *et al.* Tip-localized Ca²⁺-permeable channels control pollen tube growth via kinase-dependent R- and S-type anion channel regulation. *New Phytol* **218**, 1089-1105, doi:10.1111/nph.15067 (2018).
- 18 Graus, D. *et al.* Tobacco leaf tissue rapidly detoxifies direct salt loads without activation of calcium and SOS signaling. *New Phytol* **237**, 217-231, doi:10.1111/nph.18501 (2023).
- 19 Köster, P., DeFalco, T. A. & Zipfel, C. Ca²⁺ signals in plant immunity. *The EMBO Journal* **41**, e110741, doi:10.15252/embj.2022110741 (2022).
- 20 Kang, H.-G. *et al.* Endosome-Associated CRT1 Functions Early in Resistance Gene-Mediated Defense Signaling in Arabidopsis and Tobacco. *The Plant Cell* **22**, 918-936, doi:10.1105/tpc.109.071662 (2010).
- 21 Kutschera, A., Schombel, U., Wröbel, M., Gisch, N. & Ranf, S. Loss of wbpL disrupts O-polysaccharide synthesis and impairs virulence of plant-associated *Pseudomonas* strains. *Molecular plant pathology* **20**, 1535-1549, doi:10.1111/mpp.12864 (2019).
- 22 Chakraborty, S. *et al.* Quantification of hydrogen peroxide in plant tissues using Amplex Red. *Methods* **109**, 105-113, doi:10.1016/j.ymeth.2016.07.016 (2016).
- 23 Fichman, Y., Zandalinas, S. I., Peck, S., Luan, S. & Mittler, R. HPCA1 is required for systemic reactive oxygen species and calcium cell-to-cell signaling and plant acclimation to stress. *The Plant cell* **34**, 4453-4471, doi:10.1093/plcell/koac241 (2022).

[REDACTED]

[REDACTED]

[REDACTED]

Reviewer Reports on the Second Revision:

Referees' comments:

Referee #1 (Remarks to the Author):

Following the previous still open points from this reviewer's site, most were readily tackled. Here two comments to 1) and 4):

1) In response to the issue that the work does not include proof of spatial control, the authors inform that they have separate material showing remote control of stomatal closure (however, based on the figures and brief descriptions available to me, they activate and monitor single cells, but this doesn't necessarily mean that they perform spatial control as it can still be achieved with overall illumination and observing afterwards one cell). They suggest including material on induction of a long-range electric signal, with a preliminary sketch shown here. It would be very relevant to the impact of the work to include material on spatial control, i.e. for instance expanding on the proposed new figure in which they detect long-range electric signalling (of course with all controls and proper experimental set up showing spatial resolution, i.e. that the activity detected elsewhere is induced in the area illuminated).

The new results in protoplasts indeed add on to the description of the system in terms of cellular activation.

4) The results on ref fig 2 should be included in the current manuscript, as well as the explanations provided on the differences observed in terms of JA.

Referee #2 (Remarks to the Author):

The authors have made substantial improvement to the manuscript by diligently addressing nearly all the prior comments. This reviewer believes that XXM-type optogenetic system will be a welcome addition to the toolkit for the plant biology community.

Below are some points for authors to revise and clarify.

(1) Throughout the paper, "Tobacco", "N.benthamiana", and "N. tabacum" have been used. Please replace "tobacco" to either N.benthamiana or N. tabacum in the main text and figure legends. For example, in line 157: "Tobacco inoculation with Pseudomonas bacteria or the flagella epitope flg22 is known to induce robust [Ca²⁺]_{cyt} increases in tobacco leaves". This description may be incorrect without clarifying the plant species and isolates/strains of Pseudomonas.

(2) In the main text and figure legend, please clarify all general terms "Pseudomonas" or "Pst". Are they all Pst DC3000?

(3) In reference 7 and 15 in the response letter, the Ca²⁺ elevations were examined with different combinations of Pseudomonas syringae and Nicotiana species from those used in the current manuscript. The compatibility between microbes and hosts could affect Ca²⁺ responses. In this case, Ca²⁺ elevation in the N.b response to Pst DC3000 in figure 2a (of the response letter) may not be used directly to

compare the response of *N. tabacum* against Pst DC3000 for transcriptional analysis but is helpful to correlate. Therefore, the reviewer suggests that authors include the newly generated data on the Pst DC3000-induced Ca²⁺ influx in *N. bentamiana*, which, although not ideal, will provide some context for the readers. In addition, authors are encouraged to further tone down the parallel between Pst DC3000 spray and XXM activation.

(4) In Fig4: Consider splitting Fig4a, b to make separate y-axials. Consider moving some panels, such as Fig4c-f, to the Extended Data Figures.

(5) For enhancing readability and focus of the paper, the reviewer suggests that the description of transcriptional responses (Fig. 4) to be shortened (Line 214-288).

(6) Data organization: authors may wish to organize the Extended Data into a smaller number of integrated Figures (like the main figures) as they appear quite scattered in the current form.

Referee #4 (Remarks to the Author):

The authors have revised the manuscript based on the comments. I found the current version improved compared to the previous one. The fact that the Amplex Red did not work was surprising. However, the authors have overcome this challenge, in my view, by employing the luminol assay. While the DAB experiment is not perfect, the supporting evidence from the luminol and amperometric measurement is sufficient, in my opinion, to show the ROS differences between the treatments. As before, and even more after the revisions, this is an elegant and clear manuscript with broad interest.

Author Rebuttals to Second Revision:

Referees' comments:

Referee #1 (Remarks to the Author):

Following the previous still open points from this reviewer's site, most were readily tackled. Here two comments to 1) and 4):

1) In response to the issue that the work does not include proof of spatial control, the authors inform that they have separate material showing remote control of stomatal closure (however, based on the figures and brief descriptions available to me, they activate and monitor single cells, but this doesn't necessarily mean that they perform spatial control as it can still be achieved with overall illumination and observing afterwards one cell). They suggest including material on induction of a long-range electric signal, with a preliminary sketch shown here. **It would be very relevant to the impact of the work to include material on spatial control**, i.e. for instance expanding on the proposed new figure in which they detect long-range electric signalling (of course with all controls and proper experimental set up showing spatial resolution, i.e. that the activity detected elsewhere is induced in the area illuminated). The new results in protoplasts indeed add on to the description of the system in terms of cellular activation.

Answer: We are pleased to address the reviewer's suggestion by incorporating the new figure (Extended Data Figure 10), providing a detailed depiction of the precise spatial control of Ret-XXM 2.0 in triggering long-distance electrical signals. In response to the reviewer's recommendation, we have supplemented the experimental details in the methods part of 'Surface potential recording on *N. tabacum* leaves' and legend part particularly emphasizing the localized illumination at a distance from the electrical recording spot. Additionally, as requested by the reviewer, we have included recordings from the proper control plants, Ret-eYFP #1, in these experiments. While our manuscript is not dedicated to the precise spatial control of Ret-XXM 2.0, we hope long-range electric signaling studies can serve as a starting point for future studies in this direction (see lines 279-282).

4) The results on ref fig 2 should be included in the current manuscript, as well as the explanations provided on the differences observed in terms of JA.

Answer: According to reviewer's suggestions, the *Pst*-triggered Ca^{2+} signals from ref fig 2 were now included in Extended Data Fig. 6b, c and discussed in the main text at line 137-138. The observed differences in terms of JA and explanations are further discussed in the main text (see lines 179-181).

Referee #2 (Remarks to the Author):

The authors have made substantial improvement to the manuscript by diligently addressing nearly all the prior comments. This reviewer believes that XXM-type optogenetic system will be a welcome

addition to the toolkit for the plant biology community.

Below are some points for authors to revise and clarify.

(1) Throughout the paper, “Tobacco”, “*N. benthamiana*”, and “*N. tabacum*” have been used. Please replace “tobacco” to either *N. benthamiana* or *N. tabacum* in the main text and figure legends. For example, in line 157: “Tobacco inoculation with *Pseudomonas* bacteria or the flagella epitope flg22 is known to induce robust [Ca²⁺]cyt increases in tobacco leaves”. This description may be incorrect without clarifying the plant species and isolates/strains of *Pseudomonas*.

Answer: Thank you for this important comment. We followed the reviewer’s suggestion and replaced common names with taxonomic names in figure legends, methods and main text for clarity.

(2) In the main text and figure legend, please clarify all general terms “*Pseudomonas*” or “*Pst*”. Are they all *Pst* DC3000?

Answer: In all our experimental work, we focus on *Pseudomonas syringae* pv. *tomato* strain DC3000 (abbreviation: *Pst*) treatment on *N. tabacum*. We have clarified this information now in the main text, methods and legend part.

(3) In reference 7 and 15 in the response letter, the Ca²⁺ elevations were examined with different combinations of *Pseudomonas syringae* and *Nicotiana* species from those used in the current manuscript. The compatibility between microbes and hosts could affect Ca²⁺ responses. In this case, Ca²⁺ elevation in the *N. b* response to *Pst* DC3000 in figure 2a (of the response letter) may not be used directly to compare the response of *N. tabacum* against *Pst* DC3000 for transcriptional analysis but is helpful to correlate. Therefore, the reviewer suggests that authors include the newly generated data on the *Pst* DC3000-induced Ca²⁺ influx in *N. benthamiana*, which, although not ideal, will provide some context for the readers. In addition, authors are encouraged to further tone down the parallel between *Pst* DC3000 spray and XXM activation.

Answer: Actually, in the previous Ref. figure 2a, we detected *Pst*-triggered Ca²⁺ signals in *N. tabacum* and flg22-triggered Ca²⁺ signals in *N. tabacum* and *N. benthamiana*. It seems possible that this detail was overlooked by the reviewer. We agree with reviewer 2 that the previous reference 7 and 15 on Ca²⁺ elevations with different combinations of *Pseudomonas syringae* and *Nicotiana* species cannot be in direct support of our experiments. We therefore replaced the two references with direct experimental evidence of *Pst*-triggered Ca²⁺ signals in *N. tabacum* (Extended Data Fig. 6b, c, line 137-138). The identical *Pseudomonas syringae* (*Pseudomonas syringae* pv. *tomato* strain DC3000) and *Nicotiana* species (*N. tabacum*) in *Pst*-triggered Ca²⁺ signals measurements were used in metabolite and transcriptional analyses.

According to reviewer’s suggestions, we further toned down the parallel between *Pst* spray and XXM activation and discussed the difference between this two methods in the main text part (see line 222-230).

(4) In Fig4: Consider splitting Fig4a, b to make separate y-axials. Consider moving some panels, such as Fig4c-f, to the Extended Data Figures.

Answer: As requested, we split Fig. 4a, b to Fig. 4a-d with separate y-axials now. Accordingly, Fig. 4c-f were moved to the Extended Data Figures part as Extended Data Fig. 8d, e, h, j respectively.

(5) For enhancing readability and focus of the paper, the reviewer suggests that the description of transcriptional responses (Fig. 4) to be shortened (Line 214-288).

Answer: We followed the reviewer's suggestion and shortened this part by about 300 words keeping essential information requested by all referees. We kindly ask the reviewers to check the tracked version as some information was moved to figure legends or methods.

(6) Data organization: authors may wish to organize the Extended Data into a smaller number of integrated Figures (like the main figures) as they appear quite scattered in the current form.

Answer: Thank you. All the Extended Data figures have been re-arranged as suggested by the reviewer. According to the Nature Formatting guide, 10 Extended Data figures were generated and the Extended Data Tables are Supplementary Tables now.

Referee #4 (Remarks to the Author):

The authors have revised the manuscript based on the comments. I found the current version improved compared to the previous one. The fact that the Amplex Red did not work was surprising. However, the authors have overcome this challenge, in my view, by employing the luminol assay. While the DAB experiment is not perfect, the supporting evidence from the luminol and amperometric measurement is sufficient, in my opinion, to show the ROS differences between the treatments.

As before, and even more after the revisions, this is an elegant and clear manuscript with broad interest.

Answer: Thank you.

Reviewer Reports on the Third Revision:

Referees' comments:

Referee #1 (Remarks to the Author):

The authors have answered all issues raised.

Author Rebuttals to Third Revision:

Referees' comments:

Referee #1 (Remarks to the Author):

The authors have answered all issues raised.

Answer: Thanks.